# Global influence of mantle temperature and plate thickness on intraplate volcanism

P. W. Ball [1,2 ✉], N. J. White[1 ✉], J. Maclennan[1] & S. N. Stephenson[1,3]

The thermochemical structure of lithospheric and asthenospheric mantle exert primary controls on surface topography and volcanic activity. Volcanic rock compositions and mantle seismic velocities provide indirect observations of this structure. Here, we compile and analyze a global database of the distribution and composition of Neogene-Quaternary intraplate volcanic rocks. By integrating this database with seismic tomographic models, we show that intraplate volcanism is concentrated in regions characterized by slow upper mantle shear-wave velocities and by thin lithosphere (i.e. <100 km). We observe a negative correlation between shear-wave velocities at depths of 125–175 km and melt fractions inferred from volcanic rock compositions. Furthermore, mantle temperature and lithospheric thickness estimates obtained by geochemical modeling broadly agree with values determined from tomographic models that have been converted into temperature. Intraplate volcanism often occurs in regions where uplifted (but undeformed) marine sedimentary rocks are exposed. Regional elevation of these rocks can be generated by a combination of hotter asthenosphere and lithospheric thinning. Therefore, the distribution and composition of intraplate volcanic rocks through geologic time will help to probe past mantle conditions and surface processes.

[1] Bullard Laboratories, Department of Earth Sciences, University of Cambridge, Madingley Rise, Cambridge, UK. [2] Research School of Earth Sciences, Australian National University, Canberra, ACT, Australia. [3] Department of Earth Sciences, University of Oxford, Oxford, UK. ✉email: patrick.ball@anu.edu.au; njw10@cam.ac.uk

Mantle convection drives plate tectonics, redistributes heat, cycles chemical species, and generates dynamic topography at the Earth's surface. Constraining spatio-temporal changes in the thermochemical structure of the mantle will help to refine our understanding of these interlinked phenomena. This significant challenge requires careful integration of diverse observations. Two well-established methodologies for probing the thermochemical state of the mantle are igneous geochemistry and seismic tomography. It is particularly useful to analyze the relationship between igneous geochemistry and upper mantle structure away from complications associated with active plate boundaries in order to understand the interaction between mantle composition, temperature and melting through geologic time.

An important difficulty is that composition of partial melts within the mantle and shear-wave velocity of mantle rocks are both sensitive to the combined effects of temperature and chemical composition. The uppermost mantle is subdivided into a mechanically strong lithosphere and a weak underlying asthenosphere that is ~100–200 km thick. For a given mantle source composition, the depth and degree of melting are principally controlled by a combination of asthenospheric temperature and lithospheric thickness[1]. The extent of melting increases with increasing potential temperature, $T_p$, and decreasing pressure so that elevated asthenospheric temperature and/or thinner lithosphere produce greater volumes of melt. Given suitable assumptions, the location, volume and composition of volcanic rocks can be used to analyze the thermal structure of the uppermost mantle[2–4]. Nevertheless, it is important to bear in mind that the distribution and composition of volcanic rocks are also significantly influenced by mantle composition, by the geometry of mantle flow, and by the interaction between melts and surrounding rocks as they ascend to the surface[5–11].

Global seismic tomographic models demonstrate that shear-wave velocity anomalies, $\Delta V_s$, occur throughout the upper mantle. Although it is agreed that shear waves propagate slower through warm mantle and faster through cold mantle, it is also clear that $V_s$ varies in a non-linear fashion with pressure and temperature[12,13]. Furthermore, shear-wave speed is also sensitive to mantle composition and to the presence of interstitial melt[14,15]. By analyzing the distribution and composition of volcanic rocks in conjunction with shear-wave velocity anomalies, it is possible to determine temperature variations within the upper mantle. A related approach has been successfully used along mid-oceanic ridges where lithospheric thickness can be assumed to be negligible[16,17].

Here, we investigate a more general problem by analyzing intraplate regions, where both asthenospheric temperature and lithospheric thickness vary. Our principal aim is to quantity the relationship between the composition of volcanic rocks and mantle seismic velocities with a view to constraining the size, extent and surface expression of putative thermal anomalies. We are less concerned with the more difficult problem of how these anomalies are generated in the first place. First, we examine correlations between the distribution and composition of Neogene-Quaternary intraplate volcanism, $\Delta V_s$, and lithospheric thickness. Secondly, we estimate potential temperature and lithospheric thickness using a combination of geochemical and seismologic techniques. Finally, we scrutinize the link between intraplate volcanism and the distribution of emergent marine sedimentary deposits, which are a tangible manifestation of dynamic topographic uplift. Although our primary focus is on relatively youthful volcanic rocks, we are hopeful that our quantitative framework will provide helpful tools for interrogating the Phanerozoic record of these processes.

## Results

**Spatial distribution of intraplate magmatism.** We have compiled a global database that consists of >20,000 geochemical analyses of Neogene-Quaternary intraplate volcanic rocks (Database 1; Fig. 1; and Supplementary Tables 1 and 2). Since lithospheric plates translate across the globe, and since sub-plate mantle circulation evolves as a function of time and space, we deliberately restrict our study to samples that are <10 Ma and located <400 km whence they were erupted, based upon present-day plate speeds[18]. Global surface-wave tomographic models have a horizontal resolution of 200–600 km, which means that these restrictions ensure co-location of intraplate volcanic rocks and pertinent observations of sub-surface structure[19–22]. The majority of analyses are of mafic samples that are located far from active plate boundaries, although we have included analyses from locations adjacent to several extensional provinces and subduction zones that exhibit intraplate-like compositions (e.g., western North America, Anatolia, East African Rift). We have carried out a literature review to identify and remove samples pertaining to subduction zone processes (see Supplementary Information). Database 1 shows that most intraplate volcanism is concentrated within bands located on continental lithosphere away from cratonic regions where lithosphere >200 km thick can occur. The four most extensive bands reach along the length of western North America, along the eastern seaboard of Australia, throughout eastern China and southeast Asia, and across Europe through Anatolia and Arabia down to southern Africa. The smaller number of database entries from the oceanic realm compared with the continents reflects sampling bias toward sub-aerial locations.

If the distribution of volcanic rocks is compared with the pattern of upper mantle shear-wave velocities, it is clear that intraplate volcanism is concentrated within regions where negative shear-wave velocity anomalies occur at depths of 150 ± 25 km. Figure 1a shows a striking visual association for the SL2013sv tomographic model (see "Methods"[21]). This global vertical shear-wave velocity model exploits body and surface waves that include both fundamental and higher modes with periods of 11–450 s. It is important to emphasize that similar results have been obtained for a range of other tomographic models (Supplementary Fig. 1). This global relationship is consistent with regional studies, which show that $\Delta V_s$ correlates with geochemical indicators of melt fraction within intraplate settings[23–25]. Intraplate volcanism is almost completely absent in regions where $\Delta V_s$ is positive. There is a similarly compelling spatial relationship between the distribution of intraplate volcanic rocks and lithospheric thickness (Fig. 1b). To construct this map, shear-wave velocities from the SL2013sv model are converted into temperature by exploiting a global calibration between $V_s$ and $T$, based upon the oceanic plate cooling model[26]. Here, we utilize the revised and modified $V_s$-to-$T$ parameterization described by ref. [27], which is based upon an empirical anelastic parameterization (see "Methods"[13]). Lithospheric thickness is estimated by tracking the depth to the 1175 °C isothermal surface[27]. This isothermal surface is chosen since it provides a good fit to the peak change in the orientation of azimuthal anisotropy, which is characteristic of the transition between rigid lithosphere and convecting asthenosphere within oceanic regions[28,29]. In the continental realm, it is clear that intraplate volcanic rocks are concentrated within regions where the lithosphere is <100 km thick.

These visual associations can be quantified by formally examining relationships between, $\Sigma A_I$, the cumulative areal distribution of binned intraplate volcanic samples, $\Delta V_s$, and lithospheric thickness (see "Methods"). Eighty-nine percent of

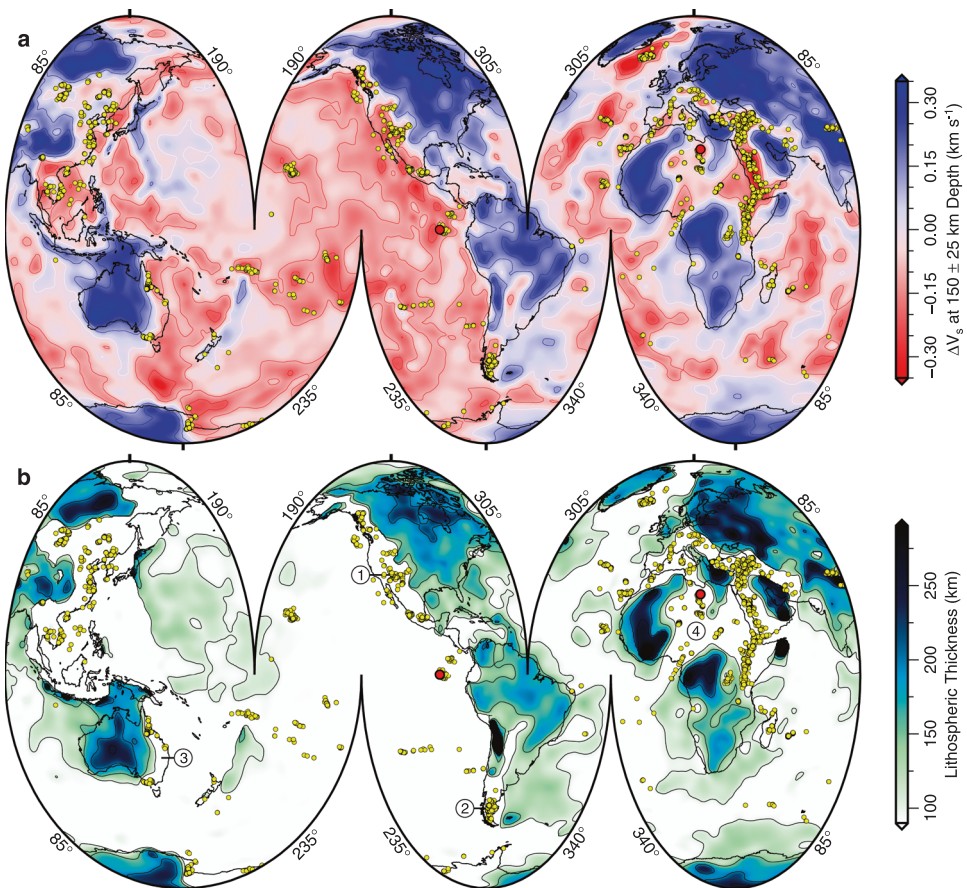

**Fig. 1 Neogene-Quaternary intraplate volcanism. a** Segmented Mollweide projection of globe showing average value of shear-wave velocity anomaly, $\Delta V_s$, taken from **SL2013sv** tomographic model at depth of $150 \pm 25$ km[21]. Red/white/blue contours = positive/zero/negative values of $\Delta V_s$ plotted at intervals of 0.1 km s$^{-1}$; yellow circles = loci of intraplate magmatic samples <10 Ma and located <400 km from locus of eruption based upon present-day plate speeds (see Supplementary Materials for Database 1). Two red circles = loci of Galápagos and Haruj provinces used for geochemical modeling in Fig. 4. **b** Spatial variation of lithospheric thickness calculated using intersection of 1175 °C isothermal surface from **SL2013sv** tomographic model that is converted into temperature[27]. Black contours = thickness values at intervals of 50 km. Encircled numbers highlight loci where gently dipping Late Cretaceous-Cenozoic marine strata crop out at elevation: 1 = western North America; 2 = southernmost South America; 3 = eastern Australia; and 4 = north Africa[55].

$\Sigma A_I$ occurs in regions underlain by negative values of $\Delta V_s$ whose areal distribution only constitutes 61% of global area (Fig. 2a). Ninety-five percent of $\Sigma A_I$ occurs in regions underlain by continental lithosphere <100 km thick whose areal distribution constitutes 62% of global area (Fig. 2b). These spatial associations are consistent for a range of tomographic and lithospheric thickness models (Supplementary Fig. 1). A Kolmogorov–Smirnov test is used to gauge the likelihood of these relationships being merely coincidental ("Methods"[30]). The probability that co-location of intraplate samples, negative values of $\Delta V_s$, and anomalously thin continental lithosphere arises from chance is <1 in $10^{100}$.

**Relationships between tomography and geochemistry.** The ratio of La and Sm concentrations in mafic igneous rocks is often used to track mantle melting[31–33]. Since La is less compatible within the mantle source compared with Sm, the La/Sm content of a melt decreases as melt fraction increases. The relationship between La/Sm and $\Delta V_s$ is a useful way to explore the role that sub-plate thermal anomalies play in generating intraplate volcanism. In order to assess this relationship, we sub-divide our global database of La/Sm and $\Delta V_s$ measurements into 1° bins and determine the Pearson product-moment correlation coefficient, $R$, between these measurements (Fig. 2c; "Methods"). Since 96%

of areal bins occur where lithospheric thickness is <100 km, we use the average value of $\Delta V_s$ at $150 \pm 25$ km depth to represent asthenospheric mantle. Mineral loss or accumulation can significantly alter the composition of a lava sample away from that of the original mantle melt. Removal or addition of mineral cargo will decrease or increase MgO concentration of the melt, respectively. Thus, MgO content is considered to be a useful indicator of either process. Here, we have excised all samples that have MgO contents <9 wt% and >14.5 wt% from Database 1 to ensure that we only retain samples that are close to the composition of primitive mantle melts. For this filtered version of Database 1, only bins with >5 samples that have been analyzed for La and Sm concentrations are exploited in order to mitigate the influence of local compositional heterogeneity.

There is a positive correlation between La/Sm ratios and $\Delta V_s$ values for the **SL2013sv** tomographic model at a depth of $150 \pm 25$ km ($R = 0.65 \pm 0.07$; Fig. 2c). Thus, igneous rocks with lower La/Sm ratios, which are indicative of higher melt fractions, coincide with regions with lower values of $\Delta V_s$ at a depth of $150 \pm 25$ km, which are indicative of higher temperatures at this depth. The value of this correlation does not significantly change if different methods of binning and filtering are employed, if locations adjacent to mid-oceanic ridges are excluded, or if regions where plate subduction has recently occurred are excised (Supplementary Figs. 2–4). In Fig. 2d, correlations between La/Sm

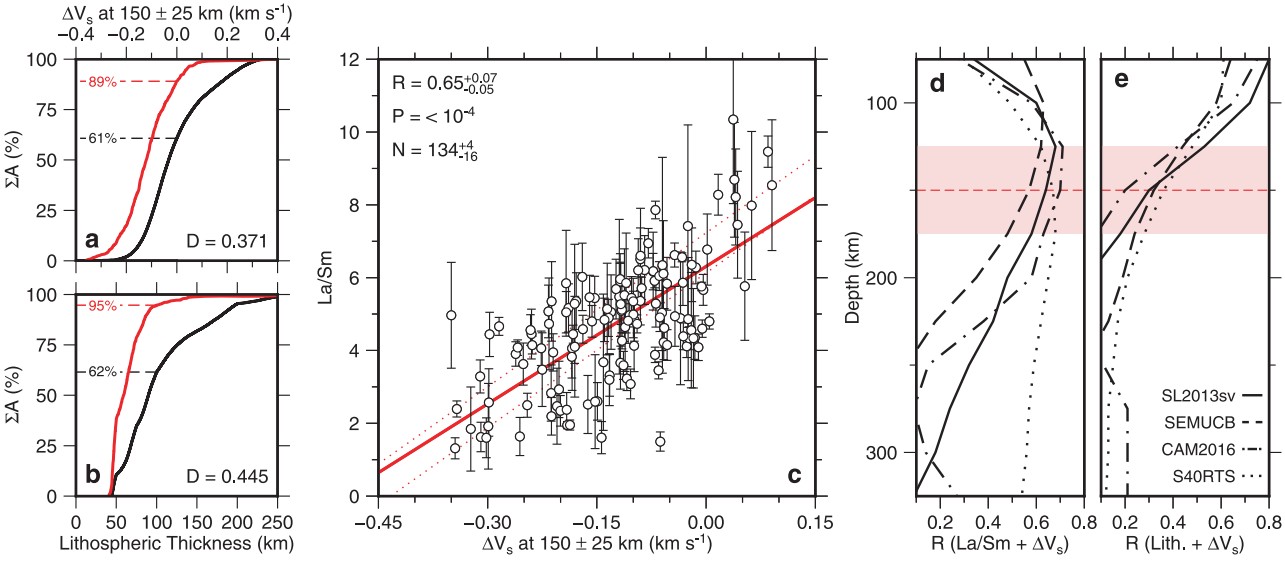

**Fig. 2 Spatial and geochemical correlations. a** Percentage cumulative area, $\Sigma A$, plotted as function of average value of $\Delta V_s$ from SL2013sv tomographic model at depths of $150 \pm 25$ km where globe is subdivided into 1° bins weighted according to $\cos \phi$ where $\phi$ = latitude in degrees; black curve = cumulative areal distribution of $\Delta V_s$; red line = cumulative areal distribution of binned intraplate volcanic samples; black/red numbered dashed lines = percentages of global surface and of intraplate volcanism with $\Delta V_s < 0$ km s$^{-1}$; D = value of Kolmogorov–Smirnov statistic. Probability value of Kolmogorov–Smirnov test is $p = 10^{-107}$ ("Methods"). **b** $\Sigma A$ plotted as function of lithospheric thickness. Black/red lines = cumulative areal distribution of lithospheric thickness and of binned intraplate volcanic samples, respectively; black/red numbered dashed lines = percentages of global surface and of intraplate volcanism with lithosphere <100 km thick; D as before. Probability is $p = 2 \times 10^{-154}$. **c** La/Sm plotted as function of average value of $\Delta V_s$ at depths of $150 \pm 25$ km from SL2013sv tomographic model. Circles and error bars = average value $\pm \sigma$ for each 1° bin; red line = line that best fits values weighted according to $\cos \phi$; pair of dotted red lines = uncertainty envelope for suite of best-fit lines where Database 1 is binned using 99 different configurations spaced at 0.1° intervals; R = correlation coefficient and its uncertainty; P = population correlation coefficient; N = number and range of bins. Note that Database 1 is screened to include only samples where $14.5 \geq$ MgO wt% $\geq 9$, <10 Ma, and number of samples per bin >5. **d** Value of R, calculated between La/Sm and $\Delta V_s$, plotted as function of depth for four different tomographic models. Solid/dash-dot/dashed/dotted lines = SL2013sv/CAM2016Vsv/SEMUCB-WM1/S40RTS tomographic models[19,21,22,34]; R = 0.28 is minimum value of R distinguishable from zero at significance level of =0.001. Red dashed line with shaded rectangle = $150 \pm 25$ km for reference. **e** Value of R, calculated between lithospheric thickness and $\Delta V_s$, plotted as function of depth as before.

and $\Delta V_s$ are shown as a function of depth for the SL2013sv, CAM2016Vsv, SEMUCB-WM1 and S40RTS global tomographic models[19,21,22,34]. In each case, the value of R is greatest between 100 and 200 km depth. For three of these models, the value of R sharply decreases at depths of >250 km. Since both La/Sm and $\Delta V_s$ are expected to decrease as mantle temperature increases, these observations suggest that intraplate volcanism is sensitive to temperature variations within the uppermost mantle.

Notwithstanding this correlation, melt chemistry and upper mantle shear-wave velocity can also be influenced by changes in lithospheric thickness, by the composition of the mantle source region, and by depth of the base of the melt column, which is controlled by a combination of mantle temperature and composition[11,14,32,35]. Although we define the lithosphere-asthenosphere boundary to be the 1175 °C isothermal surface, the base of the actual thermal boundary layer probably extends to greater depths. Global surface-wave tomographic models have a vertical resolution of 25–50 km[19,21,22,34]. Variations in thickness of the thermal boundary layer coupled with seismic resolution may, therefore, influence values of $\Delta V_s$ estimated within the asthenospheric mantle. Consequently, any correlation between La/Sm and $\Delta V_s$ could also be influenced by lithospheric thickness variations. At depths where $\Delta V_s$ values are affected by lithospheric thickness, a positive correlation between La/Sm and $\Delta V_s$ is expected. In Fig. 2e, correlations between lithospheric thickness and $\Delta V_s$ are shown for the suite of tomographic models as a function of depth. Beneath intraplate volcanic provinces, lithospheric thickness correlates significantly with $\Delta V_s$ at depths ≤100 km. At depths ≥150 km, this correlation rapidly becomes insignificant (i.e., where R < 0.28, which is the minimal

correlation distinguishable from zero at a significance level of 0.001 for a sample size of 134; see "Methods"). Clear correlations between La/Sm and $\Delta V_s$ (i.e., R = 0.5–0.7) are observed at depths of 150–200 km where the influence of lithospheric thickness changes is deemed to be negligible (Fig. 2d). Thus, it is reasonable to assume that correlations between $\Delta V_s$ and La/Sm at these depths are principally controlled by temperature variations within the asthenospheric mantle.

Composition of the mantle source region affects the values of both La/Sm and $\Delta V_s$ since melting occurs to a lesser degree within a depleted source region, depleted mantle has lower initial La/Sm values than fertile mantle, and shear waves propagate faster through depleted mantle than through fertile mantle[14,36]. To determine how the thermochemical structure of the mantle controls melt composition, we carried out principal component analysis on a suite of eight incompatible element concentrations taken from our filtered and binned version of Database 1 (see "Methods"). We then calculated correlations for each of these principal components with respect to geochemical and geophysical indicators of mantle temperature and depletion. The first principal component, $P_1$, accounts for $79 \pm 1\%$ of the variance within our filtered and binned version of Database 1. The weightings of $P_1$ are ~0.4 for all elements with the exception of Yb (Fig. 3a). As melt fraction increases, incompatible element concentrations within the melt decrease. Since the elemental data within this analysis is both mean-centered and variance-scaled, the range of values for each element between the lowest and highest melt fractions should be approximately equal. Therefore, an equal weighting over a wide range of incompatible elements for $P_1$ suggests that $P_1$ is primarily sensitive to melt fraction variation.

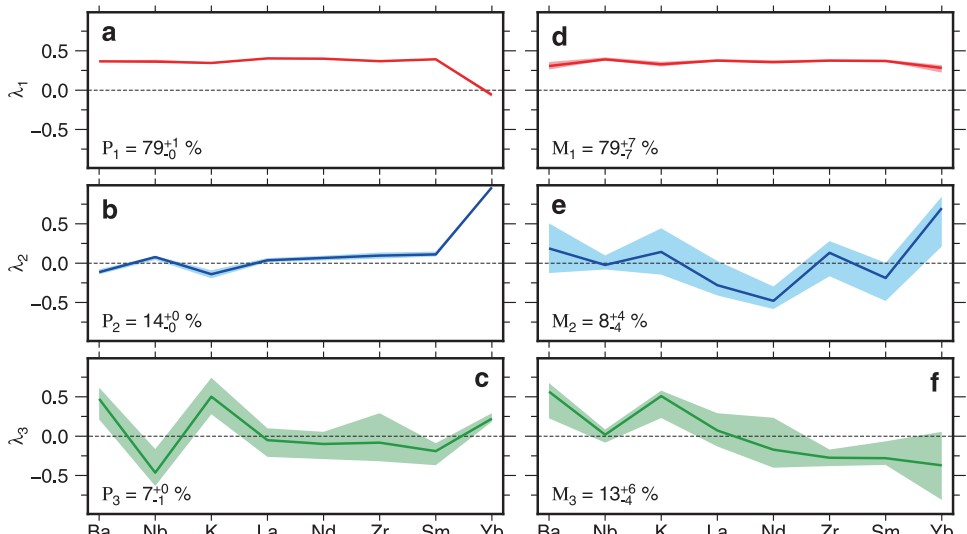

**Fig. 3 Principal component analysis of incompatible element concentrations. a** Average elemental weightings, $\lambda_1$, for first principal component, $P_1$. Database 1 is screened and binned as described in caption of Fig. 2 before being mean-centered and variance-scaled. Colored line with shaded band = average value of $\lambda_1$ for each element with range of weighting for 99 binning configurations. In this case, band is narrower than width of line. Proportion of variance described by $P_1$ is given in bottom left-hand corner. **b** Same for $P_2$. **c** Same for $P_3$. **d** Average elemental weightings for first principal component, $M_1$, generated using synthetic model that was run 99 times for appropriate range of $T_p$, lithospheric thickness, and mantle composition. Colored line/shaded band = average value/range of $\lambda_1$ for each element. **e** Same for $M_2$. **f** Same for $M_3$.

As melt fraction increases, incompatible elemental concentrations, and thus $P_1$, will decrease within the melt. Alternatively, since these elemental concentrations can vary as a function of source depletion, $P_1$ may instead be primarily sensitive to mantle composition. To investigate these contrasting hypotheses, we generate correlations for $P_1$ with respect to La/Sm, $\Delta V_s$ and $\varepsilon$Nd. Note that the $\varepsilon$Nd value of igneous rocks is a commonly used proxy for mantle depletion since depleted mantle has a higher value of $\varepsilon$Nd than fertile mantle[6,36]. If variations in $P_1$, La/Sm, $\Delta V_s$ and $\varepsilon$Nd were primarily controlled by mantle depletion, we would expect $P_1$ to positively correlate with La/Sm, but to negatively correlate with $\Delta V_s$ and $\varepsilon$Nd. However, $P_1$ positively correlates with both La/Sm and $\Delta V_s$, and it negatively correlates with $\varepsilon$Nd ($R = 0.87$, $0.59$ and $-0.50$, respectively; Supplementary Fig. 5a–c). If $\Delta V_s$ variations were primarily dependent upon mantle composition then the most depleted mantle would be associated with the fastest shear-wave speeds (i.e., $\varepsilon$Nd and $\Delta V_s$ would positively correlate). Since the opposite relationship is observed, it is less likely that the correlation between La/Sm and $\Delta V_s$ is controlled by source compositional variations. Instead, we conclude that $P_1$ is sensitive to melt fraction variations. This result is anticipated since temperature changes are known to have a greater effect upon $V_s$ than mantle composition[12,14]. We infer that upper mantle temperature variations are a dominant control on the positive relationship between melt fraction (i.e., $P_1$ and La/Sm values) and $\Delta V_s$. Note that, while fractional crystallization can have a modest influence on the value of La/Sm, $P_1$ does not correlate with MgO concentrations and the observed range of La/Sm values is too great to be controlled by fractional crystallization alone (Supplementary Fig. 6).

Melt fraction, and therefore both La/Sm and $P_1$, vary as a function of both asthenospheric temperature and lithospheric thickness. We can disentangle the relative contributions of these quantities by using the second principal component, $P_2$, which accounts for 14% of the variance and is dominated by variations in Yb. Compatibility of Yb within the mantle increases at depths greater than 60–80 km where garnet becomes stable at the expense of spinel[32,37]. Since Yb is far more compatible in garnet-bearing mantle relative to spinel-bearing mantle, Yb

concentrations within a melt are more sensitive to the depth at which melting occurs than to changes in melt fraction. Therefore, we can use $P_2$ as a proxy for comparing the respective contributions of melting within the spinel and garnet stability fields. The contribution of melting within the garnet stability field relative to the spinel field is greater beneath thicker lithosphere or in regions where elevated asthenospheric temperature acts to deepen the onset of melting. Consequently, $P_2$ can act as a proxy for lithospheric thickness, provided that asthenospheric $T_p$ remains constant. Alternatively, $P_2$ can act as a proxy for asthenospheric $T_p$, provided that lithospheric thickness is fixed. $\Delta V_s$ at depths of $150 \pm 25$ km is sensitive to asthenospheric $T_p$ variations, but insensitive to lithospheric thickness (Fig. 2e). We do not observe a correlation between $P_2$ and $\Delta V_s$ at $150 \pm 25$ km depths ($R = -0.07$; Supplementary Fig. 5e). This absence of correlation implies that asthenospheric temperature alone does not control the depth range over which melting occurs and that lithospheric thickness variation must play an important role. Consequently, these changes of lithospheric thickness will influence the observed values of La/Sm and, therefore, the inferred final melt fractions. However, if lithospheric thickness variations were the dominant control on melt fraction, we would not observe a correlation between La/Sm and $\Delta V_s$ at depths of $150 \pm 25$ km (Fig. 2d). We believe that this analysis demonstrates that asthenospheric temperature variations are the primary control on values of La/Sm but that lithospheric thickness changes exert an important secondary effect. Note that this conclusion partially contradicts previous studies, which identify a dominant lithospheric thickness signal by comparing melt chemistry with plate age in oceanic regions (e.g., refs. [11,38]). These studies were primarily concerned with major element measurements, which we do not consider here. Such opposing views are reconcilable since lithospheric thickness may have a more significant impact upon major element composition than $T_p$ if melts stall and thermally re-equilibrate at or near the base of the lithosphere[25].

The third principal component, $P_3$, accounts for $7 \pm 1$% of the variance and is primarily sensitive to variations in Ba, Nb, and K. Subduction zone melting is commonly enriched in Ba and K but

depleted in Nb. Partial melting of metasomatized lithospheric mantle can exhibit anomalously high, or anomalously low, normalized concentrations of K and Ba relative to Nb[39,40]. The low variance of $P_3$ suggests that there are minor contributions either from prior subduction zone magmatism, from melting of metasomatized lithospheric mantle, or from both. $P_3$ has low variance and does not correlate either with La/Sm or with $\Delta V_s$ at depths of $150 \pm 25$ km (Supplementary Fig. 5). Thus, the degree of contamination by lithospheric melts is evidently not the primary control for incompatible element concentrations in intraplate mafic igneous rocks.

These results are corroborated by analysis of synthetic principal components that are calculated for a suite of geochemical models. One-hundred and fifty synthetic trace element distributions were calculated by randomly varying values of $T_p$, lithospheric thicknesses, and mantle compositions within fixed limits (i.e., 1250–1450 °C, 35–75 km, and primitive/depleted mantle, respectively; see "Methods"). This procedure is repeated 99 times. The first synthetic principal component, $M_1$, is similar to $P_1$ both in terms of elemental weightings and variance (Fig. 3d). $M_1$ strongly correlates with $T_p$ and does not significantly correlate either with lithospheric thickness or with mantle depletion ($R = -0.85$ to $-0.72$, $R = 0.05$ to $0.42$, $R = -0.27$ to $0.21$, respectively; Supplementary Fig. 7). It is reasonable to conclude that both $M_1$ and $P_1$ are primarily sensitive to changes in $T_p$. $M_2$ accounts for $8 \pm 4\%$ of the variance and, like $P_2$, it is dominated by variations in Yb (Fig. 3e). For the majority of model runs, $M_2$ correlates with lithospheric thickness but does not correlate either with $T_p$ or with mantle depletion ($R = -0.81$ to $-0.26$, $R = -0.37$ to $0.24$, $-0.22$ to $0.18$, respectively; Supplementary Fig. 7). Correlation between $M_2$ and lithospheric thickness may indicate that $P_2$ is also primarily sensitive to lithospheric thickness variations. $M_3$ has a similar distribution of weightings as $P_3$ but not as strong weighting for Ba, K, and Nb. As expected, $M_3$ does not consistently correlate with $T_p$, lithospheric thickness or depletion. These observations add strength to the idea that $P_3$ represents lithospheric contamination, which is not accounted for by this synthetic model. Note that conclusions based on these synthetic models depend upon the assumption that asthenospheric $T_p$, lithospheric thickness, and mantle depletion are uncorrelated with respect to each other on Earth.

**Calculating asthenospheric temperatures.** The global distribution of intraplate magmatism together with a positive correlation between La/Sm and $\Delta V_s$ at a depth of $150 \pm 25$ km suggest that intraplate magmatism is principally associated with elevated asthenospheric temperatures beneath thin lithosphere. This inference can be quantitatively investigated by analyzing relationships between the chemical composition of volcanic rocks, shear-wave velocity anomalies and asthenospheric potential temperature, $T_p$. An important goal is to compare geochemical and geophysical estimates of $T_p$.

First, the chemical composition of intraplate volcanic rocks is used to constrain the depth and degree of mantle melting. Here, we exploit a geochemical inverse modeling strategy, which minimizes the misfit between observed and calculated concentrations of rare earth elements (REE) by systematically varying melt fraction as a function of depth ("Methods"[2,41]). In this polybaric model, REE concentrations are calculated by integrating instantaneous melt compositions along isentropic melting paths. These melting paths are determined by a combination of potential temperature of the mantle source region, $T_p$, and lithospheric thickness, $a$. The top of the melting column is, by definition, the base of the lithosphere. Beneath the lithosphere, melt fraction as a function of depth is controlled by a combination of $T_p$ and $a$

chosen melting parameterization. Here, we exploit the hydrous lherzolitic melting model described by Katz et al.[35]. Note that some model parameters have been updated to honor experimental constraints that have subsequently been obtained ("Methods"[42]). This melting model requires a $T_p$ value of 1312 °C in order to generate the average crustal thickness at a mid-oceanic ridge (i.e., 6.9 km[29]). We assume that this value represents the potential temperature of ambient asthenospheric mantle. For each volcanic province, we systematically vary $T_p$ and $a$ from 1250 to 1550 °C and from 30 to 80 km, respectively. For each combination of $T_p$ and $a$, the misfit between observed and calculated compositions is measured. In this way, the values of $T_p$ and $a$, which yield the best fit, are identified. Note that the quality of fit between observed and calculated REE concentrations depends upon the relative contributions of melting within the spinel- and garnet-bearing peridotite stability fields. Combinations of values of $T_p$ and $a$ are deemed acceptable provided that the residual misfit between observed and calculated REE concentrations is not >1.5 times the smallest residual misfit value for any given model run. Volcanic provinces that exhibit a greater range of REE concentrations can generally be fitted within acceptable limits using a larger number of $T_p$ and $a$ combinations.

Figure 4 shows inverse modeling results for the Galápagos volcanic islands located upon the Nazca plate and for the Haruj volcanic province located within north Africa. In each case, an optimal distribution of melt fraction as a function of depth is calculated by a grid search in which only $T_p$ and $a$ are varied. Mantle depletion and water content are set by the value of εNd ("Methods"). This straightforward approach enables trade-off between values of $T_p$ and $a$ to be clearly identified. For the Galápagos archipelago, we obtain an excellent fit between observed and calculated REE concentrations for $T_p = 1366 \pm 15$ °C and $a = 47 \pm 5$ km, which is within ~30 °C and ~5 km of previous estimates (Fig. 4a, b[24]). The misfit function shows that there is minimal trade-off between $T_p$ and $a$ and that the value of $a$ could be smaller but not significantly larger (Fig. 4c). For the Haruj province, we obtain a satisfactory fit for $T_p = 1367 \pm 28$ °C and $a = 56 \pm 2$ km, in agreement with previous estimates (Fig. 4d, e[43]). The misfit function shows that there is a negative trade-off between $T_p$ and $a$, which means that a slightly thicker lithosphere with hotter asthenospheric mantle could fit the observations equally well (Fig. 4f). In both examples, a combination of elevated asthenospheric temperature and anomalously thin lithosphere is required, which is principally a consequence of the constraints imposed by low REE concentrations and by the depth of the spinel-garnet phase transition, respectively.

Inverse modeling has been carried out for intraplate volcanic analyses from our filtered and binned version of Database 1 (Supplementary Fig. 9). In this way, we calculate values of $T_p$ and $a$ for every 1° bin in locations where intraplate volcanism occurs. These values can be directly compared with asthenospheric temperatures and with plate thicknesses obtained from tomographic models that have been converted into temperature. Here, we compare geochemical estimates of $T_p$ with values determined at $150 \pm 25$ km depths for the **SL2013sv** model ("Methods"[21,27]). Note that the geochemical inverse model and the $V_s$-$T$ conversion scheme assume hydrous and anhydrous mantle sources, respectively. Consequently, a slightly higher value of ambient potential temperature, $T_p = 1333$ °C, would be required to generate the average oceanic crustal thickness for the $V_s$-$T$ scheme[27,29]. Globally, we observe a positive correlation of $R = 0.54$ between geochemical and tomographic estimates of $\Delta T_p$, which is the temperature anomaly with respect to ambient asthenospheric mantle (Fig. 5a).

We acknowledge that limitations of both geochemical and tomographic methodologies could give rise to $\Delta T_p$ discrepancies

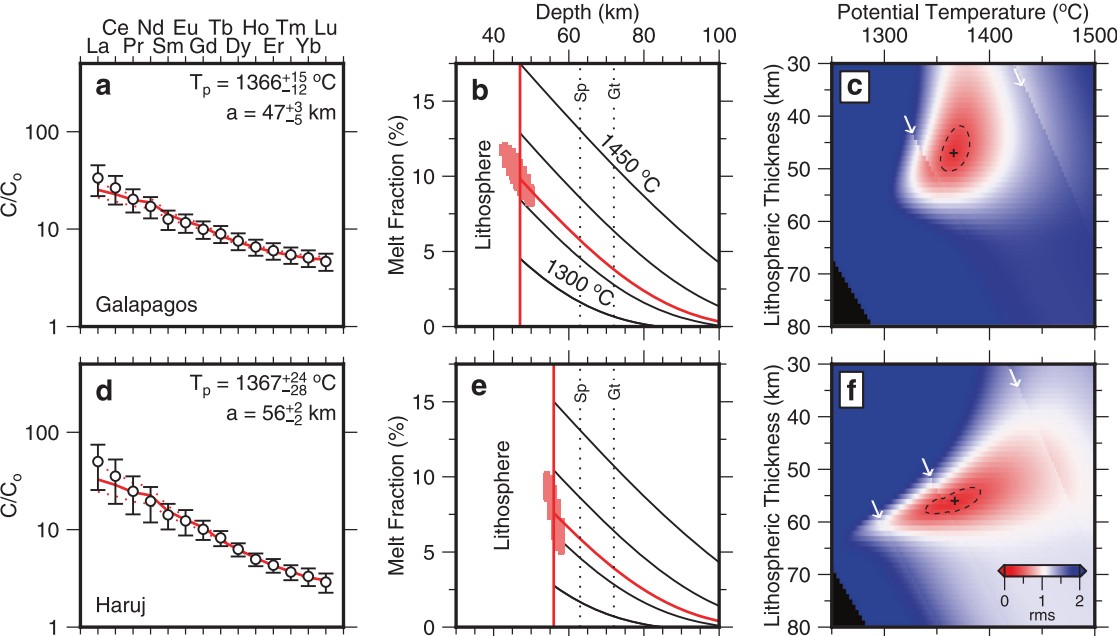

**Fig. 4 Inverse modeling of rare earth element (REE) concentrations. a** Galápagos Islands, Ecuador (0 ± 0.5°N, 92 ± 0.5° W). Open circles with vertical bars = average REE concentrations ± σ normalized with respect to source composition where εNd = 6.21; red line = REE concentrations calculated by inverse modeling where residual rms value at global minimum is 0.31; pair of dotted red lines = calculated REE concentrations where residual rms value is 1.5× that at global minimum. Optimal values and associated uncertainties of potential temperature, $T_p$, and of lithospheric thickness, $a$, displayed at top right-hand side. **b** Melt fraction plotted as function of depth. Curved red line = melt fraction calculated by fitting observed REE concentrations in panel **a**; vertical red line = calculated top of melting column; pink polygon = range of uncertainty for $T_p$ and $a$ at top of melting column calculated from suite of models where residual rms value is 1.5× that at global minimum; black lines = set of isentropic curves calculated using parametrization of ref. [42] corrected for appropriate water content and labeled by values of potential temperature; pair vertical dotted lines at 63 and 72 km = limits of spinel-garnet transition zone[37,43]. **c** Misfit between observed and calculated REE concentrations plotted as function of $T_p$ and $a$ where color scale of rms value calculated using Eq. (22) is given in bottom right-hand corner of panel **f**; black cross = locus of global minimum; dashed black line indicates 1.5 × rms value at global minimum; white arrows = loci of phase transitions. **d**–**f** Same for Al-Haruj province, Libya (28 ± 0.5°N, 18 ± 0.5°E) where εNd = 5.18 and rms value at locus of global minimum = 0.28.

(e.g., refs. [14,26,43,44]). Global tomographic models are considerably damped and smoothed. Ray path coverage of the upper mantle is variable, spatial resolution is limited to ~200–600 km, and lateral smearing of fast velocities can sometimes occur adjacent to thick cratonic lithosphere. These spatial resolution issues could account for the small number of intraplate volcanic regions that fall on regions with positive values of $\Delta V_s$ at depths of 150 ± 25 km. Calibration of the $V_s$–$T$ parametrization is predicated upon the validity of the plate cooling model since extrapolation of elastic and anelastic behavior determined from laboratory experiments might not be directly applicable to damped tomographic models[26]. The presence of melt within the mantle may reduce $V_s$ as a result of poroelastic effects that are not accounted for within this parameterization[15]. However, the amount of melt retained within the mantle is probably very small (i.e., ~0.1%), especially at a depth of 150 km, which in most regions is beneath the peridotite solidus[35,45]. Since the reduction of $V_s$ caused by poroelastic effects should scale as a function of melt fraction, these effects can be considered to be negligible[46]. The parameterization does not account for compositional variations, which means that temperature estimates for continental regions, especially depleted cratonic cores, may involve additional uncertainties. In continental regions with thin lithosphere (i.e., ≲75 km), additional uncertainties can arise as a result of inaccuracies in the parameterization of crustal structure. These inaccuracies can cause "bleeding" of slow crustal velocities into the uppermost mantle that are interpreted as hotter temperatures, which yield underestimates of lithospheric thickness[26]. Nevertheless, temperature profiles calculated using this tomographic approach provide acceptable fits to continental geothermal profiles derived from xenolith thermobarometric observations for locations with a range of lithospheric thicknesses (50–200 km[27]). The observed negative correlation between εNd and $\Delta V_s$ at depths of 150 ± 25 km also indicates that, in regions of thin lithosphere, shear-wave velocities are controlled by thermal anomalies rather than by compositional variations associated with depletion and enrichment (Supplementary Fig. 5). Within the oceanic realm, where lithospheric structure and composition are broadly uniform, the uncertainty of tomographically derived lithospheric thickness estimates is probably ±30 km[47].

Geochemical inverse modeling of intraplate volcanic rocks necessarily depends upon a simplified treatment of mantle source composition, upon the depth of the spinel-garnet phase transition, upon experimentally determined partition coefficients, and upon details of the melting model[2]. Changes in values of input parameters inevitably affect absolute values of $T_p$ and $a$. For example, increasing the depth to the spinel-garnet transition by 5 km, decreasing water content of the mantle by 0.01 wt%, or varying mantle composition from depleted to primitive mantle affects calculated values of $T_p$ and $a$ by +15 °C and +5 km, by +5–10 °C and +2 km, and by +20 °C and −3 km, respectively[43]. Furthermore, the inverse model assumes a uniform lherzolitic source and does not consider harzburgitic and pyroxenitic lithologies. For example, the presence of harzburgite and pyroxenite within the mantle can lead to over- and underestimates of $T_p$, respectively, if a purely peridotitic model is implemented[48]. These sources of uncertainty could be responsible for low temperature values recorded in some provinces and for the minor, but

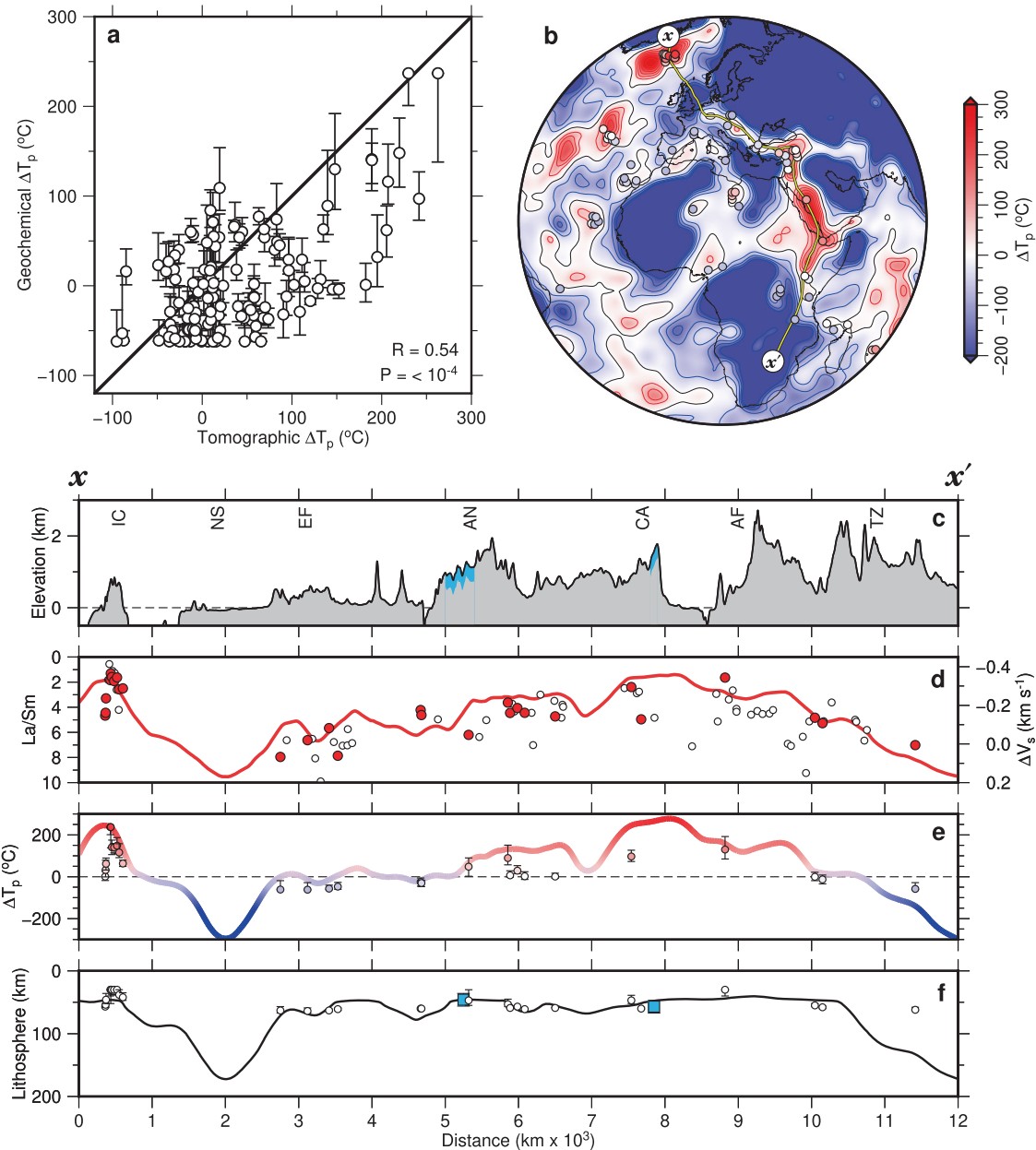

**Fig. 5 Relationships between calculated temperatures, lithospheric thickness variation, and dynamic topography. a** Diagram showing relationship between $\Delta T_p$ determined by geochemical inverse modeling and that determined by calibration of **SL2013sv** tomographic model where ambient mantle temperatures are assumed to be 1312 °C and 1333 °C, respectively. Open circles with vertical and horizontal error bars = calculated values of temperature for global distribution of intraplate volcanism ± 1.5× minimum misfit; black line 1:1 relationship; $R$ = correlation coefficient; $P$ = population correlation coefficient; $R = 0.28$ is minimum value of $R$ distinguishable from zero at significance level of = 0.001. **b** Map showing spatial variation of average value of $\Delta T_p$ at depths of 150 ± 25 km calculated from **SL2013sv** tomographic model. Red/white/blue contours = positive/zero/negative values of $\Delta T_p$ at intervals of 50 °C; colored circles = geochemical values of $\Delta T_p$; yellow line between $x$ and $x'$ = locus of transects shown in panels **c–f**. **c** Transect from $x$ to $x'$ that intersects Iceland (IC), North Sea (NS), Eifel (EF), Anatolia (AN), Central Arabia (CA), Afar (AF) and Tanzania (TZ). Black line with gray polygon = topographic profile; dashed line = reference sea level; blue polygons = loci of uplifted Cenozoic marine strata[58,59]. **d** Red line = average values of $\Delta V_s$ at depths of 150 ± 25 km along transect calculated from **SL2013sv** tomographic model; open circles = observed values of La/Sm averaged into 1° bins that lie within <200 km of transect; red circles = bins with > 5 values of La/Sm. **e** Colored ribbon = average values of $\Delta T_p$ at depths of 150 ± 25 km calculated from **SL2013sv** tomographic model where color scale is shown at right-hand side of panel **b**; colored circles with vertical bars = values of $\Delta T_p$ calculated for screened and binned version of Database 1 ± 1.5× minimum misfit. Dashed black line at $\Delta T_p = 0$ is for visual guidance. **f** Black line = lithospheric thickness variation along transect calculated from **SL2013sv** tomographic model; circles with vertical bars = estimates of lithospheric thickness determined by geochemical modeling ± 1.5× minimum misfit; blue squares = present-day lithospheric thickness estimated from elevation of marine rocks by assuming combination of lithospheric mantle thinning, where original thickness is 120–150 km, and sub-plate thermal anomaly with thickness of 100 km whose magnitude is taken from panel **e**.

systematic, offset between geochemical and tomographic temperature estimates (Fig. 5a). The correlation coefficient between geochemical and seismic temperature estimates is less than the correlation coefficient between La/Sm and $\Delta V_s$ ($R = 0.65$ and $R = 0.54$, respectively). Although we have shown that asthenospheric temperature variations control the correlation between La/Sm and $\Delta V_s$, lithospheric thickness changes represent an important secondary geochemical signal. Since our geochemical potential temperatures are calculated by accounting for lithospheric thickness changes, the decrease in correlation coefficient that we observe may result from removal of this secondary signal. Despite the inherent uncertainties associated with both modeling approaches, the positive correlation between these independent estimates affirms the underpinning assumptions and highlights the potential rewards of combining temperatures estimated from geochemical analyses of intraplate volcanic rocks with temperatures calculated by calibrating tomographic models.

In order to highlight the consistency between geochemical and tomographic estimates of mantle potential temperature and lithospheric thickness, we present a hemispherical transect that summarizes regional variations of La/Sm, $\Delta V_s$ at a depth of $150 \pm 25$ km, $\Delta T_p$ at a depth of $150 \pm 25$ km, and $a$ across Europe and Africa (Fig. 5b). This transect starts at the Icelandic plume, crosses Central Europe and Arabia, traverses the East African Rift, and finishes on the South African craton (Fig. 5c). The lowest global values of La/Sm and $\Delta V_s$ are associated with the Icelandic plume (Fig. 5d). Further south, there is a gradual decline in both of these values between the Eifel region and Central Arabia. South of the Afar region, La/Sm and $\Delta V_s$ values increase again, although there is some scatter. As expected, there are concomitant increases and decreases in geochemical and tomographic estimates of $\Delta T_p$ (Fig. 5e). Geochemical and tomographic estimates of lithospheric thickness also broadly match along this transect (Fig. 5f). Although it is not guaranteed that a seismically defined lithosphere-asthenosphere boundary should coincide with the depth at which melting ceases, it is likely that the difference between these depths is minimal compared to the uncertainties inherent in both techniques. Calculated lithospheric thicknesses of ~50–70 km occur beneath European and Arabian intraplate volcanic provinces. Given that crustal thicknesses in these regions are ~30–40 km, our combined geochemical and tomographic values suggest that the lithospheric mantle is remarkably thin beneath these provinces[49]. Note that in regions where the lithosphere is >125 km thick, a correlation between tomographically derived estimates of lithospheric thickness and the values of $\Delta T_p$ at a depth of $150 \pm 25$ km is expected since these values of $T_p$ reside within the lithospheric mantle (e.g., North Sea, Tanzania).

## Discussion

Our global analysis of volcanic provinces and shear-wave velocity anomalies suggests that a combination of asthenospheric temperature anomalies and lithospheric thickness variations helps to determine the spatial distribution of intraplate magmatism. This inference is manifest by a range of geochemical and geophysical observations. First, the bulk of Neogene-Quaternary intraplate volcanism coincides with slow shear-wave velocity anomalies and thin lithosphere. Secondly, there is a positive correlation between La/Sm values of intraplate volcanic rocks and the average magnitude of these shear-wave velocity anomalies within an asthenospheric channel centered on $150 \pm 25$ km. Thirdly, potential temperatures and lithospheric thicknesses calculated by geochemical inverse modeling of REE concentrations generally match those values obtained from a $V_s$-$T$ calibration of the SL2013sv tomographic model. Significantly, our results complement those obtained by a global study of the mid-oceanic ridge

system, which shows that shear-wave velocity anomalies correlate with major element compositions of basaltic rocks and with axial ridge depths[17]. Note that, in addition to the influence of temperature anomalies, these correlations could be partly modified by mantle compositional variation or by crystal fractionation beneath mid-oceanic ridges[50,51].

Intraplate volcanism is spatially associated with thin lithosphere and its melt composition can be used to estimate asthenospheric temperature. Both thinning of undepleted lithospheric mantle and elevated asthenospheric temperature are significant means for generating regional uplift on horizontal length scales of order $10^3$ km and on timescales of order 10 Ma. Therefore, our results have direct implications for the spatial and temporal evolution of dynamic topography, which is generated and maintained by mantle convective processes[52,53]. We define dynamic topography to embrace long-wavelength topography generated by mantle flow, isostatic responses to thermochemical processes within the convecting mantle, as well as regional changes in thickness of the lithospheric mantle[54,55]. If we assume an equilibrated lithospheric thickness of $a_0 = 120$–$150$ km at mean sea level, the amount of regional uplift is given by

$$U = \frac{\alpha T_1}{1 - \alpha T_1}\left(\frac{a_1^2}{2a_0} + \frac{a_0}{2} - a_1 + \frac{\Delta T}{T_1}h\right), \qquad (1)$$

where $a_1$ is the present-day lithospheric thickness, $\alpha = 3 \times 10^{-5}\,°C^{-1}$ is thermal expansivity, $\Delta T$ is an asthenospheric temperature anomaly of thickness $h$, and $T_1 = 1390\,°C$ is ambient asthenospheric temperature (i.e., the temperature at 150 km depth provided $T_p = 1330\,°C$, mantle rock density $= 3300\,kg\,m^{-3}$, and specific heat capacity $= 1187\,J\,kg^{-1}\,K^{-1}$ (refs. [42,56,57]). If lithospheric thickness is reduced to, say, $a_1 = 60$ km, if $\Delta T = 50\,°C$, and if $h = 100$ km, $U = 0.81$–$1.34$ km for the disequilibrated case where the effects of pressure upon density are ignored. Variations in $\Delta T$ will change $U$ by ~$3\,m\,°C^{-1}$. Thus, a significant corollary of our proposed mechanism for generation of intraplate volcanism is rapid large-scale uplift. This prediction is easily tested by examining the relationship between topographic elevation and emergent (but undeformed) marine sedimentary rocks that crop out in regions where intraplate volcanism predominates, where asthenospheric temperatures are anomalously high, and where continental lithosphere is anomalously thin. Dramatic examples include western North America, southernmost South America, eastern Australia, and North Africa where marine deposits of negligible dip indicate that hundreds of meters of wholesale rapid uplift occurred during Cenozoic times (Fig. 1b[55]). In western Arabia and in eastern Anatolia, Cenozoic marine rocks occur at Jabal Umm Himar and within the Sivas Basin at present-day elevations of 1.2 and 1.5 km, respectively (Fig. 5c, f[58,59]). Regional uplift of these undeformed marine deposits can be achieved by a combination of emplacement of an asthenospheric temperature anomaly and lithospheric mantle thinning[60,61].

Dynamic topography undoubtedly makes a profound contribution to relative sea-level changes, ancient oceanic circulation, fluvial drainage patterns, and sedimentary deposition[62–67]. Here, we have shown that intraplate magmatism, positive asthenospheric temperature anomalies and thinned lithosphere are closely related during Neogene-Quaternary times. Removal or thinning of lithospheric mantle produces regional uplift but subsidence will eventually occur as the lithosphere conductively cools and re-thickens[68,69]. These processes generate vertical motions of the order of hundreds of meters at the Earth's surface that are superimposed on motions generated by mantle convection. There is considerable debate about the way in which mantle flow contributes to dynamic topography, and about the relative contributions of the upper and lower mantle[70]. By combining the distribution and

composition of intraplate volcanism with calibrated tomographic models, we have shown that asthenospheric temperature variation and lithospheric thickness changes can make a significant contribution to dynamic topography. Since intraplate magmatism occurs throughout the stratigraphic record, we suggest that a combined analysis of igneous rocks and stratigraphy could help to elucidate the history of mantle convective and dynamic topographic phenomena throughout the Phanerozoic Era.

## Methods

**Tomographic model**. For this study, we primarily exploited the SL2013sv tomographic model[21]. This global upper mantle model uses body and surface waves that include both fundamental and higher modes with periods of 11–450 s. A total of ~750,000 seismograms were analyzed by ref. [21] and misfits were calculated by comparing observed and calculated waveforms (≤18th overtone). Significantly, the SL2013sv model inverts for crustal structure during the optimization process whereas other global models mostly, but not always, use a defined a priori crustal architecture. The SL2013sv model has a vertical resolution of 25–50 km and chequerboard tests demonstrate that features with diameters of ~600 km can be clearly resolved at lithospheric depths. In areas of greater ray-path coverage (e.g., North America, Europe) finer scale features should be resolvable. Here, this model is shown relative to AK-135 reference model recomputed for a reference period of 50s and slightly modified following a preliminary optimization[71]. This reference model also incorporates laterally varying crustal structure that is initially based upon CRUST2.0 and is continually updated during optimization[21,49]. Descriptions of the CAM2016Vsv, SEMUCB-WM1 and S40RTS tomographic models are provided in Supplementary Information[19,22,34].

**Shear-wave velocity to temperature conversion**. We exploit a $V_s$-to-$T$ conversion scheme to estimate temperature variations of the upper mantle from shear-wave velocities. The temperature model exploited in our study is created by applying the empirical approach of ref. [27] to the SL2013sv tomographic model[21]. This scheme was originally developed by ref. [26] and it has been updated to exploit an empirical formula, which describes $V_s$ as a function of pressure and temperature[13]. Several of the parameters used in this empirical formula were experimentally constrained[13]. However, seven material constants are required to be independently calibrated for each different tomographic model. These constants are: $\mu_U^0$, $\frac{\partial \mu_U}{\partial T}$, $\frac{\partial \mu_U}{\partial P}$, $\nu_r$, $E_a$, $V_a$ and $\frac{\partial T_s}{\partial z}$, which represent the unrelaxed shear modulus at surface pressures and temperatures, how the shear modulus changes as a function of temperature, how the shear modulus changes as a function of pressure, shear viscosity for a reference pressure, temperature and grain size (1200 °C, 1.5 GPa, 1 mm), activation energy, activation volume and solidus temperature as a function of depth, respectively.

All seven empirically determined parameters are permitted to vary within physically plausible limits (e.g., refs. [13,26,72]). The resultant $V_s(T, P)$ parameterization is calibrated by minimizing four misfit functions between observed and calculated values in respect of four different sets of constraints (i.e., $H_1$–$H_4$). For oceanic lithosphere, $V_s$ varies as a function of depth and plate age[12]. Oceanic plate profiles parallel to plate spreading (i.e., flowlines) taken from the SL2013sv tomographic model are averaged and $V_s$ as a function of depth and plate age is determined. These stacked tomographic profiles are compared to values of $V_s$ estimated by applying the $V_s$-to-$T$ parameterization to an oceanic plate cooling model[29]. $H_1$ is then calculated by computing the rms misfit between observed and calculated $V_s$ profiles at depths of 75 and 100 km[27]. Oceanic plate cooling models can only be used to minimize the difference between observed and calculated values of $V_s$ at depths ≤125 km[29]. Therefore, a second misfit function, $H_2$, is constructed at depths beneath the upper-thermal boundary layer (i.e., from >225 to 400 km). $V_s$ values as a function of depth for the SL2013sv tomographic model are averaged across oceanic regions and compared with values of $V_s$ estimated from a 1333 °C isentrope calculated for peridotite using material constants from ref. [42]. Both the oceanic plate cooling model and the temperature profile for convecting mantle are assumed to have a $T_p$ of 1333 °C[27,29,42]. Mantle with a $T_p$ value of 1333 °C is considered to be at ambient temperatures within the final model. These empirically derived parameters are used to provide a prediction of shear-wave attenuation, $Q^{-1}$ (ref. [73]). The calculated value of $Q^{-1}$ can be compared to an estimate of seismic attenuation at depths >150 km beneath old oceanic floor (>100 Ma[74]). The misfit between observed and predicted $Q^{-1}$, $H_3$, is calculated for these locations. Finally, the misfit between an assumed value of shear viscosity, $\log_{10}[\nu_{ref}] = 3 \times 10^{20}$ Pa s, and the mean value of $\log_{10}[\nu]$ determined beneath the thermal boundary layer, $H_4$, is calculated. The combined misfit, $H$, between observed and calculated values of $V_s$, is quantified by summing weighted versions of these four rms misfit functions, $H_1$–$H_4$, where

$$H = \sum_{i=1}^{4} w_i H_i \Big/ \sum_{i=1}^{4} w_i. \tag{2}$$

$w_1$–$w_4$ are weighting coefficients with values of 10, 1, 2 and 2, respectively[27]. See refs. [27] and[47] for a detailed explanation of this $V_s$-to-$T$ parameterization.

**Kolmogorov–Smirnov statistical test**. A two-sample Kolmogorov–Smirnov test is used to calculate how likely it is that two cumulative distribution functions, CDFs, are drawn from the same reference distribution[30]. The probability, $P$, is approximated by

$$P = \exp\left(\frac{-2nmD^2}{n + m}\right), \tag{3}$$

where $D$ is the Kolmogorov–Smirnov Test Statistic (i.e., the maximum magnitude difference between two CDFs). $D$ can vary between 0 and 1. $m$ and $n$ are the number of samples in each CDF, which in this case are areal distribution of data (i.e., shear-wave velocity or lithospheric thickness) over the Earth's surface and the subset of that distribution, which contains recent intraplate volcanism, respectively.

Database 1 is used to define locations of recent intraplate volcanism. Here, Database 1 is filtered to remove samples that are >10 Ma and samples that are >400 km from their predicted site of eruption. Note that we do not filter for MgO content. For ease of analysis, we have subdivided the Earth's surface into 1° × 1° bins where $p$ is the total number of bins. The subset of bins containing intraplate volcanic rocks is termed $q$. To avoid biasing this analysis by oversampling at high latitudes, the number of bins in each distribution, $p$ and $q$, are weighted by the latitude of each bin (i.e., a bin closer to the equator covers a larger geographic area). $m$ and $n$ are calculated by weighting each bin, $p_i$ and $q_i$, by $\cos\phi_i$ where $\phi_i$ is the latitude of the bin such that

$$m \approx \sum_{i=1}^{p} \frac{\pi \cos\phi_i}{2}, \qquad n \approx \sum_{i=1}^{q} \frac{\pi \cos\phi_i}{2} \tag{4}$$

where $\frac{\pi}{2}$ is the global average bin weighting. Thus, one bin at the equator in the distribution $p$ will count for ≈1.6 bins while a bin at 80° north will count as ≈0.3 bins. The probability that 1° bins containing intraplate volcanic samples are randomly distributed so that the great majority lie above regions of negative $\Delta V_s$ is <1 in $10^{100}$. The probability that bins containing intraplate volcanic samples are randomly distributed such that they lie above thin (<100 km thick) lithosphere is <1 in $10^{150}$.

**Correlation coefficient**. The quality of correlation between two datasets, for example between La/Sm and $\Delta V_s$, can be determined using the Pearson product-moment correlation coefficient, $R$. In this study, we compare global databases subdivided into 1° × 1° bins. Since the geographic area of a given bin depends upon its latitude, the influence of each bin, $i$, on the value of $R$ is once again assigned a weighting, $w_i = \cos\phi_i$, so that bins closer to the poles that cover smaller geographical areas have less effect on the value of $R$. The weighted Pearson product-moment correlation coefficient is defined as

$$R_{xy} = \frac{\Sigma w_i (x_i - \bar{x}_w)(y_i - \bar{y}_w)}{\sqrt{\Sigma w_i (x_i - \bar{x}_w)^2 \Sigma w_i (y_i - \bar{y}_w)^2}}, \tag{5}$$

where

$$\bar{x}_w = \Sigma w_i x_i / \Sigma w_i, \qquad \bar{y}_w = \Sigma w_i y_i / \Sigma w_i. \tag{6}$$

The symbols $x_i$ and $y_i$ denote the $x$ and $y$ values of each bin. The intercept, $\beta_0$, and the slope, $\beta_1$, which define the linear best-fit to observed values are calculated using

$$\beta_0 = \bar{y}_w = \beta_1 \bar{x}_w, \qquad \beta_1 = \frac{\Sigma w_i (x_i - \bar{x}_w)(y_i - \bar{y}_w)}{\Sigma w_i (x_i - \bar{x}_w)^2}. \tag{7}$$

To determine the statistical significance of these correlations, we can use either the population correlation coefficient or a look-up table of $t$-test critical values, which define the lowest value of $R$ that is distinguishable from 0 for a given sample size. For these significance-limit calculations, we apply a weighting of 1 to all bins.

**Principal component analysis**. Before principal component analysis is carried out, Database 1 is filtered so that only samples with 9 < MgO wt% <14.5, which are <10 Ma, which are <400 km from their predicted site of eruption, and which contain all eight incompatible trace elements, are included. These measurements are binned in the way described in the main text and any bins that have ≤5 samples are removed. Each bin is then normalized to average composition for each element within the filtered and binned version of Database 1. Principal component analysis is carried out using the statsmodels.multivariate.pca routine from the Python software package where the following settings are selected: number of calculated components = 3; standardize = "True"; and method = "NIPALS".

To construct a synthetic dataset, elemental concentrations are calculated using the INVMEL model for a range of $T_p$ values, lithospheric thicknesses, and mantle compositions. Random distributions of $T_p$, lithospheric thickness and $\varepsilon$Nd are generated using Gaussian distributions, each of which is defined by a mean and standard deviation of 1350 °C and 40 °C, 55 km and 8 km, and 5 and 2, respectively. Outer limits for $T_p$, lithospheric thickness and $\varepsilon$Nd are set at 1250–1450 °C, 35–75 km, and 0–10, respectively. Finally, resultant $T_p$, lithospheric thickness and $\varepsilon$Nd values are rounded to the nearest 5 °C, 1 km and 0.25, respectively. In each case, 150 values are generated and these values are combined to describe a series of input parameters for 150 INVMEL models. Concentrations of Ba, Nb, K, La, Nd, Zr, Sm and Yb calculated for these model runs are then modeled using principal component analysis. This procedure is repeated 99 times.

**Geochemical estimates of $T_p$.** The INVMEL-v12 geochemical model is used to calculate rare earth element (REE) concentrations generated by mantle melting for different values of $T_p$ and lithospheric thickness[41]. Concentration of each REE within an instantaneous melt, $c_l$, and concentration within the residue, $c_s$, are related to each other by two equations, which must be simultaneously solved. These equations are

$$\frac{dc_s}{dX} = \frac{c_s - c_l}{1 - X} \quad \text{and} \quad c_l = \frac{c_s(1 - X)}{\bar{D}(1 - X_0) - \bar{P}(X - X_0)}, \quad (8)$$

where $X$ and $X_0$ the total melt fraction at the current and previous time step, respectively; $\bar{D}$ is the bulk distribution coefficient for any given element within the solid assemblage, and $\bar{P}$ is the bulk distribution coefficient for the melting assemblage[75]. $\bar{P}$ and $\bar{D}$ vary as a function of depth since both the mineral assemblage present and individual mineral distribution coefficients are depth-dependent. To calculate $c_l$ and $c_s$ at each depth, the expressions in Eq. (8) are numerically integrated using a fourth-order Runga-Kutta scheme from the base of the melting interval, where $X = 0$, to the top of the melting interval, where $X = X'$. The average melt composition for all melt extracted from a rock at a single depth, $C$, is related to $c_l$ by

$$C = \frac{1}{X'} \int_0^{X'} c_l(X) dX. \quad (9)$$

The mean composition of all of the melt extracted from the melting interval 0–$h$ is given by

$$\mathscr{C} = \int_0^h \frac{X'C}{1 - X'} dz \Big/ \int_0^h \frac{X'}{1 - X'} dz. \quad (10)$$

$X'(z)$ is calculated from the equations of ref. [35] updated using revised parameter values from ref. [42]. $X'$ is a function of potential temperature, $T_p$, lithospheric thickness, $a$, the weight fraction of water present within the source region, $X_{H_2O}^{bulk}$, and the modal clinopyroxene by mass of the peridotite source, $M_{cpx}$. $X_{H_2O}^{bulk}$ is approximated from the concentration of Ce within the source region and $X_{Ce}^{bulk}$ is constrained by assuming that $X_{H_2O}^{bulk}/X_{Ce}^{bulk} = 200$[76]. $M_{cpx} = 0.17$ is fixed for all models.

Composition of the mantle source region used in this modeling procedure varies as a function of the average value of $\varepsilon$Nd calculated for each province. It is calculated by linearly mixing between a primitive mantle source, PM, and a depleted MORB mantle, DMM, in order to match the observed value of $\varepsilon$Nd (Supplementary Table 5[77]). Note that varying REE element concentrations within the source region as a function of $\varepsilon$Nd also affects the value of $X_{H_2O}^{bulk}$ for the source since we assume $X_{H_2O}^{bulk}/X_{Ce}^{bulk} = 200$. Exploiting the melting parameterization of ref. [42] and assuming an $\varepsilon$Nd value of 10, a value of $T_p = 1312 \pm 28$ °C is required to generate $6.9 \pm 2.2$ km of oceanic crust at a mid-oceanic ridge, assuming a triangular melt geometry[29]. We thus assume that 1312 °C is the ambient value of $T_p$. Within the mantle, the aluminous phase present varies as a function of depth between plagioclase, spinel and garnet. The mineral proportions used within each stability field are given in Supplementary Table 6. The plagioclase-spinel transition zone is set at 25–35 km and the spinel-garnet transition zone is set at 63–72 km[2,37].

The bulk distribution coefficient of the melting assemblage, $\bar{P}$, is calculated to determine the concentration of REEs within the melt phase. The **INVMEL** model incorporates two melting regimes: one regime where all minerals listed in Supplementary Table 6 are present; and a second regime where the more fusible minerals are exhausted, leaving only olivine and orthopyroxene. The proportion of each mineral present within the source region by weight is given by $F_n$, where $n = 1, 2, 3, 4, 5$ and $6$ refer to olivine, orthopyroxene, clinopyroxene, plagioclase, spinel and garnet, respectively. The point at which clinopyroxene and aluminous phases (i.e., plagioclase, spinel and garnet) become exhausted is referred to as $X_1$. $X_1$ is assumed to be located where $X = 0.18$. The proportion of clinopyroxene and aluminous phase present is assumed to vary linearly between the initial proportion of each mineral present within the source, $F_n = F_n^0$, and when the minerals are exhausted $F_n = 0$, between $X = 0$ and $X = X_1$. The proportion of olivine to orthopyroxene remains fixed and the concentration of these minerals within the source region is varied so that

$$\sum_{n=1}^N p_n = 1, \quad (11)$$

$$\text{if } n \geq 3, \quad p_n = \frac{F_n X}{X_1}, \quad (12)$$

$$\text{if } n < 3, \quad p_n = \frac{F_n(1 - \sum_{n=3}^N p_n)}{F_1 + F_2}, \quad (13)$$

$$\text{if } X < X_1, \quad \bar{D} = \sum_{n=1}^N F_n D_n \quad \text{and} \quad \bar{P} = \sum_{n=1}^N p_n D_n, \quad (14)$$

$$\text{if } X \geq X_1, \quad \bar{D} = \frac{F_1 D_1 + F_2 D_2}{F_1 + F_2} \quad \text{and} \quad \bar{P} = p_1 D_1 + p_2 D_2, \quad (15)$$

where $N$ is the total number of mineral phases.

To calculate the bulk distribution coefficient for a given element within the solid assemblage, $\bar{D}$, the partition coefficients for each mineral, $D_i$, must be parameterized. Values of $D_i$ for each REE in olivine, orthopyroxene and spinel are listed in Supplementary Table 8. In the mantle, partition coefficients necessarily vary as a function of pressure, temperature and mineral chemistry as sites within mineral lattices expand and contract. The equation for calculating the partitioning of an element with a charge $v+$ and a radius $r_i$ entering into site $M$ of a crystalline lattice is governed by

$$D_i = D_{0(M)}^{v+} \times \exp\left(\frac{-4\pi N_A E_M^{v+}\left(\frac{1}{2} r_{0(M)}^{v+}(r_i - r_{0(M)}^{v+})^2 + \frac{1}{3}(r_i - r_{0(M)}^{v+})^3\right)}{RT}\right), \quad (16)$$

where $N_A$ is Avogadro's number, $R$ is the gas constant, $E_M^{v+}$ is the elastic response of the site $M$, $r_{0(M)}^{v+}$ is the radius of the site and $D_{0(M)}^{v+}$ is the partition coefficient for an element with a charge $v+$ and a radius $r_{i(M)}^{v+}$ [78]. The constants required to calculate $D_i$ for REEs in clinopyroxene, plagioclase and garnet as a function of pressure, temperature and chemistry are listed in Supplementary Tables 8 and 9. Partitioning of REEs into garnet is governed by the fraction of pyrope present, PYR, and can be calculated from

$$\text{PYR} = \frac{\text{MgO}_{mol}\left(1 - \frac{\text{Cr}_2\text{O}_{3mol}}{\text{Cr}_2\text{O}_{3mol} + \text{Al}_2\text{O}_{3mol}}\right)}{\text{FeO}_{mol} + \text{MgO}_{mol} + \text{CaO}_{mol}}, \quad (17)$$

where $\text{MgO}_{mol}$, $\text{Cr}_2\text{O}_{4mol}$, $\text{Al}_2\text{O}_{3mol}$, $\text{FeO}_{mol}$, $\text{MgO}_{mol}$, $\text{CaO}_{mol}$ are the fraction by weight of each oxide within garnet divided by their respective molecular weights (Supplementary Table 7[79]). Partitioning of REEs into clinopyroxene depends upon the proportions of Ca and Al within the clinopyroxene phase, $x_{Ca}^{M2}$ and $x_{Al}^{M1}$, respectively[78]. These proportions are calculated using

$$x_i = i_{mol} \times C_i \times \frac{1}{6} \sum_i^I i_{mol} O_i, \quad (18)$$

$$x_{Ca}^{M2} = x_{Ca}, \quad (19)$$

$$x_{Al}^{M1} = x_{TiO} + \frac{1}{2}(x_{Al_2O} - 2x_{TiO} - x_{Na_2O} - x_{K_2O}), \quad (20)$$

where $x_i$, $i_{mol}$, $C_i$ and $O_i$ are the proportions of oxide $i$ within clinopyroxene, fraction by weight of oxide $i$ within clinopyroxene divided by its molecular weight, the number of cations in the oxide $i$, the number of oxygen atoms in the oxide $i$ listed in Supplementary Table 7, respectively.

**Grid search procedure.** Before comparing calculated REE concentrations, $\mathscr{C}_m$, with observed REE concentrations, $\mathscr{C}_o$, the observed concentrations are corrected for olivine fractional crystallization. For each sample, the proportion of olivine that must be added into the melt, $\chi_{ol}$, to ensure that the melt is in equilibrium with olivine of 90% forsterite content, is calculated. The equations of ref. [4] are used, assuming $Fe^{3+}/\sum Fe = 0.15$[43]. The average $\chi_{ol}$ value for all samples within a given volcanic province is used and no REEs are assumed to be present within the cumulate olivine. Corrected observed REE concentrations, $\mathscr{C}_c$, are therefore given by

$$\mathscr{C}_c = \mathscr{C}_o(1 - \chi_{ol}). \quad (21)$$

To identify that model that best fits the corrected REE concentrations, $\mathscr{C}_c$ is compared with $\mathscr{C}_m$ calculated for each pair of $T_p$ and $a$ values at 1 °C and 1 km intervals, respectively. $T_p$ is varied between 1250–1550 °C and $a$ is varied between 30 and 100 km. Certain samples do not have recorded concentrations for all 14 REEs. To ensure average REE concentrations are consistent, if <50% of samples record concentrations for a given element, it is omitted from the misfit calculation. Following this screening process, if the number of REEs that a province contains, $N$, is <7 then $T_p$ and $a$ are not recorded for that particular province. The root mean squared (rms) misfit, $H$, between $\mathscr{C}_c$ and $\mathscr{C}_m$ at each $T_p$ and $a$ pair is given by

$$H(T_p, a) = \sqrt{\frac{1}{N}\left(\sum_{n=1}^N \frac{(\mathscr{C}_c - \mathscr{C}_m)^2}{\sigma_c^2}\right)}, \quad (22)$$

where $\sigma_c$ is the standard deviation of $\mathscr{C}_c$ for each province. The best-fitting pair of $T_p$ and $a$ values is taken to be the global minimum for $H$. If the value of $H$ at the global minimum is >1 then $T_p$ and $a$ are not recorded for that province. Acceptable models are those where the value of $H$ is <1.5 × the global minimum. The errors for each $T_p$ and $a$ estimate show this range of acceptable values.

## Data availability
All geochemical data analyzed during this study are included in this published article and are included and appropriately referenced within Supplementary Data File 1 and Supplementary Tables 1 and 2. Tomographic models, $V_s$-to-$T$ models, and lithospheric thickness maps analyzed in this study can be accessed through the references provided (e.g., refs. [19–22,27,80]).

## Code availability
The INVMEL software package is available on request from D. McKenzie. All other codes used can be provided by the authors upon request.

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

## Acknowledgements

P.W.B. acknowledges support by Shell Research. We are grateful to K. Czarnota, I. Frame, L. Kennan, M.J. Hoggard, M. Klöcking, D. Lyness, F. McNab, C.P.B. O'Malley, F. Richards, G. Roberts, O. Shorttle, and K. Warners for their help. D. McKenzie generously provided software for the INVMEL forward model. Y. Niu provided an insightful review. Figures were prepared using Generic Mapping Tools software. Database 1 was compiled with the aid of the GEOROC database (www.georoc.edu). Cambridge Earth Sciences contribution number esc.6010.

## Author contributions

All authors jointly contributed to the conceptualization and design of this research project. P.W.B. collated the geochemical database, performed the geochemical modeling, carried out the statistical analysis, and drafted the figures. P.W.B. and N.J.W. wrote the paper with assistance and advice from J.M. and S.N.S.

## Competing interests

The authors declare no competing interests.
