## [Peer Review File · Nature Communications]

Reviewers' comments:

Reviewer #1 (Remarks to the Author):

Ball, White & MacLennan: Nature Communications 2020 submission The authors show that the distribution of Neogene and Quaternary intraplate volcanic rocks is correlated with seismic tomographic models having low Vs between 100 and 200 km, and with thin lithosphere (i.e. < 70 km). They infer that temperature anomalies are responsible for the positive correlation between ΔV_s and La/Sm in the basalts.

Their overarching conclusion is stated at the end in Section 4: "Our global analysis of volcanic provinces and shear-wave velocity anomalies suggests that a combination of sub-plate asthenospheric temperature anomalies and thin lithosphere controls the spatial distribution of intraplate magmatism."

This is a major claim that may hold in some cases. But do the authors really think that the locations of Hawaii and Iceland, to name two, are really controlled by asthenospheric temperature anomalies in the 100-200 km range? Are Hawaii and Iceland melting from hot and shallow asthenosphere? Is there no role for deep mantle plumes that are anchored at the base of the mantle, LLSVPs? The author's conclusion is in stark contrast with that reached by Scott French and Barbara Romanowicz (2015; Nature) that Hawaii's present position is controlled by the northern edge of the Pacific LLSVP; here they showed the Hawaiian plume as a broad region of low seismic velocity that is continuously connected from the surface to the LLSVP. The conclusions of Ball et al. also contradict an important role for the Pacific LLSVP on the locations of Austral Cook, Easter, Galapagos, Marquesas, Pitcairn, Samoa, Society Islands (e.g., Montelli et al., 2004; Torsvik et al., 2006; Hassan et al., 2015). Importantly, Williams, Mukhopadhyay, Rudolph, Romanowicz (2019) show that many of these volcanoes above the Pacific LLSVP have elevated primitive $^3\text{He}/^4\text{He}$; this is evidence against the model that they were formed by partial melting of shallow sub-lithospheric asthenospheric mantle.

On the other side of the world, many of the volcanoes considered by the authors have distributions that are controlled by the African LLSVP (e.g., Montelli et al., 2004; Torsvik et al., 2006; Hassan et al., 2015; Williams et al., 2019). These include those from the East African Rift (Afar, Kenya, Tanganyika), Azores, Cameroon, Canary Islands, Cape Verde, Comoros, Crozet, Hoggar, St. Helena, Tristan da Cunha, and Iceland. Importantly, EAR volcanism is continuously connected to the African LLSVP at the base of the mantle via a broad low velocity seismic anomaly called the African Superplume (e.g., Mulibo and Nyblade, GRL 2013; GRL; Hansen, Nyblade, Benoit, EPSL 2012).

The authors have inverted REE concentrations in Galapagos lavas to infer mantle potential temperature and lithosphere thickness and they show their results in Figure 4. Their inferred lithospheric thickness is 48 ± 3 km, and they cite agreement with Gibson and Geist (2010). More recent work gives 91 ± 8 km beneath the southeastern archipelago and 72 ± 5 km beneath surrounding regions, based on receiver function studies (Byrnes et al., 2015).

Recommendation to the Editor

For this paper to go forward in Nature, Ball et al. need to explain why their model of uppermost asthenosphere mantle melting is a better way of interpreting the spatial distribution of intraplate volcanism than deep mantle plumes connected to LLSVPs.

Reviewer #2 (Remarks to the Author):

Dear Editor,

I read the manuscript by Ball et al. with great interest. I consider this is an exciting piece of work, the kind of work that I wanted to do by myself for some long time, but I could not because I am only a passive user of seismic tomography. Among other things, the significant correlation between La/Sm of basalts and δV_s anomaly (Fig. 2c) is a brilliant finding of far reaching significance. I have reservations on the interpretation of this correlation, but given its significance, I recommend the manuscript be published with revision. In revision, I suggest authors consider my comments given at the margins of this word document, which is summarized below. Fix some typographic errors, and errors in some statements and reasoning.

Regards,

Yaoling Niu (you may please reveal my identity to the authors)

Page 2: Commented [YN1]	Yaoling Niu	2020/6/28
--------------------	------------------

Be explicit what this positive correlation physically means. That is, higher speed correlates with lower extent of melting?

Page 2: Commented [YN2]	Yaoling Niu	2020/6/28
--------------------	------------------

Be explicit in terms of physical meaning.

Page 2: Commented [YN3]	Yaoling Niu	2020/6/28
--------------------	------------------

See

[1] Niu et al. (2011, JPet; <http://community.dur.ac.uk/yaoling.niu/MyReprints-pdf/2011NiuEtAl-JPetrology.pdf>)

[2] Niu & Green (2018, ESR; <http://community.dur.ac.uk/yaoling.niu/MyReprints-pdf/2018Niu&Green-ESR-LAB.pdf>)

Page 2: Commented [YN4]	Yaoling Niu	2020/6/28
--------------------	------------------

This is reasonable, but caution is still necessary because 10 Myrs is a rather long time during which many things may have changed. For example, active volcanism only takes place on the Big Island of Hawaii, but no volcanism on any of the Hawaiian Islands to the north only a few Myrs older.

The co-location will be less a problem with slow plate motion speed.

Page 2: Commented [YN5]	Yaoling Niu	2020/6/28
--------------------	------------------

Yes, this review article states some basic geological concepts: Niu (JAES, 2020; <http://community.dur.ac.uk/yaoling.niu/MyReprints-pdf/2020Niu-JAES-SS-VolumeBreakup.pdf>)

Page 2: Commented [YN6]	Yaoling Niu	2020/6/28
--------------------	------------------

Eastern China!

Page 2: Commented [YN7]	Yaoling Niu	2020/6/28
--------------------	------------------

It is a good one, but not striking as stated. It is not obvious in the cases of New Zealand, Mongolia.

Also, is it appropriate to treat the Red Sea as “intra-plate” setting? It has mature seafloor and separates the

Arabic plate from the African plate with a full separation rate of > 20 mm/yr.

Page 2: Commented [YN8] Yaoling Niu 2020/6/28

What is the vertical resolution? I gather the correlation would be better if the depth interval is restricted to 80-150 km?

Page 3: Commented [YN9] Yaoling Niu 2020/6/28

Yes, almost!

Page 3: Commented [YN10] Yaoling Niu 2020/6/28

Yes, agree!

Page 3: Commented [YN11] Yaoling Niu 2020/6/28

Unclear!

Page 3: Commented [YN12] Yaoling Niu 2020/6/28

True! But surprisingly in most cases (if we deal with global data), the averages of smaller numbers of samples are as significant despite the relatively large variability (or s.d.).

Page 3: Commented [YN13] Yaoling Niu 2020/6/28

This is expected without having to invoke temperature variation. The correlation is significant at > 99% confidence level while a factor of up to 3 La/Sm variation at a given $\square V_s$ also exists. Both are likely real!

So, the first order correlation between the erupted basalts may actually reflect (or inherited from) the amount of “trapped” interstitial melt in the mantle source region (deeper than the dry solidus, but shallower than the volatile-bearing solidus); the average depth of 150 km with highest R’s makes sense.

The greater melt retention (or high melt porosity, melt fraction), (1) the lower La/Sm in such melt and (2) the lower $\square V_s$ of the bulk mantle. $\square V_s$ is directly proportional to the shear modulus, which is much more sensitive to the amount of the melt porosity than temperature variation!

So, the large La/Sm variation at a given $\square V_s$ may (1) reflect average source compositional variation from one location to another (variably depleted or refertilized source lithologies, volatiles as well as major elements and modal mineralogy), which may also (2) affect the properties of extent of melt retention (melt porosity).

Page 3: Commented [YN14] Yaoling Niu 2020/6/28

Please put legend in (d) which model is which.

Page 3: Commented [YN15] Yaoling Niu 2020/6/28

The lid effect is far more significance! See <http://community.dur.ac.uk/yaoling.niu/MyReprints-pdf/2011NiuEtAl-JPetrology.pdf>

Page 4: Commented [YN16] Yaoling Niu 2020/6/28

Why? With the presence of volatiles and liquid/fluid phases (melt), the V_s would be reduced because of reduced shear modulus. Depleted source would be depleted in volatiles and would be less likely to form a melt phase at any given condition. If you are talking about major elements, then the depleted source would have high Mg# and low Al₂O₃ (thus less garnet). As the result, the bulk rock would have reduced density, which will facilitate V_s propagation (density is on the denominator). Hence, shear waves are expected to propagate faster, not slower, through the depleted mantle.

Page 4: Commented [YN17] Yaoling Niu 2020/6/28

This is possible, but the presence of a melt phase (tiny fraction) = melt retention = melt porosity, would have

significantly greater effect.

Page 4: Commented [YN18] **Yaoling Niu** **2020/6/28**

This would be debatable!

Page 4: Commented [YN19] **Yaoling Niu** **2020/6/28**

This is not obvious from Fig. 6.

Page 4: Commented [YN20] **Yaoling Niu** **2020/6/28**

This is not true! For example, see this paper <http://community.dur.ac.uk/yaoling.niu/MyReprints-pdf/2010HuangEtAl-JPet.pdf>

Up to 5.23 wt% K₂O and 1116-1566 ppm Ba.

Page 4: Commented [YN21] **Yaoling Niu** **2020/6/28**

Probably true! BUT, if the depth of melting is not important, what would be the effect of temperature that you argue for? This needs rigorous elaboration.

Metasomatism that we often invoke to use refers to the source history prior to the recent major melting event.

Page 4: Commented [YN22] **Yaoling Niu** **2020/6/28**

The temperature effect is neither obvious nor implied by the data; preferred interpretation?

I do not object the effect of mantle temperature variation, but the reasoning seems to have something missing!

Page 4: Commented [YN23] **Yaoling Niu** **2020/6/28**

No! Not on the Pacific plate, but Nazca plate subducting beneath South America.

Page 5: Commented [YN24] **Yaoling Niu** **2020/6/28**

You need to tell what REE sources values are used. Is a uniform mantle REE source composition assumed?

Page 5: Commented [YN25] **Yaoling Niu** **2020/6/28**

Why different T_p's are assumed? It is better explained here.

Page 5: Commented [YN26] **Yaoling Niu** **2020/6/28**

This is actually NOT expected. Why should there be an inverse correlation between mantle T_p and lithosphere thickness? There is no reason.

Mantle T_p is the surface projection of mantle temperature along the adiabat, which related to the mantle convection, whereas lithosphere thickness would have long geological history that neither controls mantle convection nor is a consequence of present-day mantle convection. This needs explanation in terms of understandable physics.

Page 5: Commented [YN27] **Yaoling Niu** **2020/6/28**

There may be some artefacts? Thin lithosphere means decompression to a shallow level, i.e., $F \propto P_o - P_f$ (Niu

et al, 2001; Niu & Green, 2018), i.e., the lid effect:

<http://community.dur.ac.uk/yaoling.niu/MyReprints-pdf/2011NiuEtAl-JPetrology.pdf>

<http://community.dur.ac.uk/yaoling.niu/MyReprints-pdf/2018Niu&Green-ESR-LAB.pdf>

So, thin lithosphere lid is associated with higher extent of melting and relatively lower incompatible elements

(also lower La/Sm) in the basalts, whereas thick lid is associated with lower extent of melting and relatively higher incompatible elements (also higher La/Sm).

Page 5: Commented [YN28]

Yaoling Niu

2020/6/28

This is interesting! Objectively, the Na90 argument is incorrect. See Niu (2016):

<http://community.dur.ac.uk/yaoling.niu/MyReprints-pdf/2017Niu-JPet-GlobalMORBMajor.pdf>

This is not a matter of personal preference or choice, but my objective scientific analysis.

Page 5: Commented [YN29]

Yaoling Niu

2020/6/28

Explain this *O*.

Page 5: Commented [YN30]

Yaoling Niu

2020/6/28

Both New Zealand and Mongolia are such exceptions?

Page 6: Commented [YN31]

Yaoling Niu

2020/6/28

See above comment of the unexpected inverse correlation of *T_p* and lithosphere thickness.

Page 6: Commented [YN32]

Yaoling Niu

2020/6/28

What sort of marine data?

Page 6: Commented [YN33]

Yaoling Niu

2020/6/28

Hard to follow! You are comparing marine data with thin continental lithosphere.

Page 6: Commented [YN34]

Yaoling Niu

2020/6/28

What may have caused the lithosphere thinning in these places? Here is my version of continental lithospherethinning, including some places also mentioned here.

<http://community.dur.ac.uk/yaoling.niu/MyReprints-pdf/2014Niu-GTM.pdf> (See Figures 10-15), which is summarized here: <http://www.mantleplumes.org/Hydration.html>

Page 6: Commented [YN35]

Yaoling Niu

2020/6/28

No! Basal removal or lithosphere thinning will NOT cause uplift, but if anything, will lead to subsidence due to Airy isostasy. See page 172 of this short paper: <http://community.dur.ac.uk/yaoling.niu/MyReprints-pdf/2019Niu-CSB-India-Asia.pdf>

Reviewer #3 (Remarks to the Author):

This paper describes an interesting combined analysis of seismic tomography and REE trends. It is encouraging that the two approaches yield correlated estimates of asthenospheric temperatures. However, the paper is poorly organised and the presentation and discussion of results leaves many aspects insufficiently discussed and explored, making it difficult to be convinced of the (significance of the) results.

(1) The manuscript is relatively poorly written and organised. For most of the methods the reader is referred to other papers, which makes it difficult to assess the robustness of the results. Issues raised in the introduction do not link well to those addressed in the analyses. And the conclusions provide some method limitations and discuss dynamic topography rather than wrapping up the main results from the analyses. In addition, there is a very large supplement with a lot of results that are further results of the analyses rather than included to support the content of the main text. I am not sure that Nature Communications offers the best format to publish this work.

(2) It does not become very clear what the main new results are. Some of the results are rather trivial (intraplate volcanism tends to occur where plates are thin), other results are only partially discussed (what is the relation between potential temperature and plate thickness). There are a number of statistical tests, but many are not too clear or convincing: (i) Kolmogorov Smirnov tests seem quite a lot of effort for illustrating that, as one would expect from a textbook understanding of adiabats and solidi, intraplate volcanism concentrates where plates are thin, (ii) what does the principal component analysis add?, (iii) how significant is the correlation between geochemical and seismic temperatures and what about the systematic offset between the two? Although it is suggested that sublithospheric temperatures exert a main control on the occurrence of intraplate magmatism, I did not find the case against a dominant control by lithospheric thickness convincing.

(3) The introduction does not provide a very clear motivation for the study. The first paragraph of the introduction mentions the integration of observations and geodynamic modelling, but this is not something this paper actually does. A next paragraph mentions: "Recent advances in topographic modelling enable shear wave velocities to be mapped in detail throughout the upper mantle", but most of the tomographic models used are not all that recent. Mainly the motivation is one of opportunity, seismic and geochemical estimates of mantle potential T under continents can only be combined where intraplate volcanism occurs. The final conclusion appears to be one on dynamic topography, a topic only introduced in the last few paragraphs of the paper.

(4) All aspects of the methods need more explanation in the main text. The actual method section only discussed statistical methods and the REE modelling.

(4.1) For the geochemical database, what suite of geochemical indicators is used to remove samples? Supplementary Table 3 only mentions MgO. Is the database to be released with the paper? Will the database contain additional information about how samples have been chosen for inclusion or removal?

(4.2) A discussion is missing of whether sampling bias could affect the interpretation of the distribution of intraplate volcanism. I.e., what is the uncertainty and potential bias on the cumulative areal extent that is used for later correlations?

(4.3) The shear velocity models are used for correlation with the distribution of intraplate volcanism without any introduction of the Vs models, their sensitivity, resolution (laterally and with depth) and limitations. There should also be some discussion of what the reference value of Vs means as much significance is attached to where anomalies are positive and negative. Some of these aspects are mentioned in the final section (on geodynamic implications) but without discussing how these might influence the results.

(4.4) The conversion to thermal structure needs more explanation than currently given, including mentioning uncertainties and possible biases. To what extent are the seismic velocity models able to independently resolve lithospheric thickness and asthenospheric temperature? And how correlated are (continental) lithospheric thickness and asthenospheric? Very correlated if one goes by Fig. 5. This is surprising as one might have expected that under continents, asthenospheric temperature is not the only control on thickness, but that compositional structure of the lithosphere, tectonic history and distribution of heat producing elements might also have an effect.

(4.5) Actually it is may not be appropriate to call the temperature inferred from velocities between 100-200 km depth below continents T_p . Definitely where the continents are thicker than 100 km, these are lithospheric temperatures or an average of lithospheric and asthenospheric temperatures. With the choice for base of the lithosphere as the 1175°C isotherm, even if the estimated thickness is less than 100 km, some of (transient) thermal boundary layer, which extends to temperatures above 1175°C, will be below this depth and hence these temperatures will average into the “sublithospheric” temperature. Furthermore, the minimum vertical resolution achieved with the seismic models is generally about 50 km, adding to the uncertainties in the seismic temperature estimates. These arguments together with what appears to be a strong correlation between thickness and dVs in 100-200 km depth range indicate that the velocities in this depth range may in part represent variations in lithospheric thickness.

(4.6) Fig . 4 gives an example, for two locations, for the potential trade-off between depth and temperature in the results from the REE data. It would help if the reader were given a more general insight in this important trade-off, and under which circumstances the two might be independently resolvable and when not (maybe with an additional figure).

(4.7) It would also be good if it could be illustrated that by combining the geochemical and seismic constraints it is possible to resolve thickness- T_p trade-offs.

(5) Comments on the results

(5.1) It is good to see that the correlation between La/Sm and dVs is investigated before a comparison between the temperature estimates from the two types of observables is made. But could you also provide an indication of how significant the correlation is?

(5.2) It is nice that there is also a correlation between T_p from seismic and geochemical analyses. However, again the question is how significant is an R of 0.55? And in addition, most of the seismic T_p estimates are higher than the geochemical estimates. This warrants discussion.

(5.3) If however as argued above the variations in dVs in the 100-200 km depth range is, at least in part, an expression of variable lithospheric thickness than the correlation between La/Sm and dVs is an argument for a strong control by lithospheric thickness rather than by sublithospheric temperatures on intraplate volcanism. This again relates to the question of how independently resolvable the two effects are by La/Sm trends.

(5.4) The conclusion from the work appears to be that both thin lithosphere and elevated asthenospheric temperatures contribute to intraplate volcanism. However, the distribution of asthenospheric temperatures is not really discussed. As queried above, are the seismic estimates actually estimates of T_p ? A strong correlation between T_p and lithospheric thickness is not necessarily expected. Hotter asthenosphere could thin the lithosphere, but thinner lithosphere also allows melting to occur for cooler mantle temperatures.

(6) The discussion of dynamic topography comes as a bit of a diversion. The rest of the paper had not mentioned dynamic topography, and the distribution of temperature and thickness obtained was not thoroughly discussed before being used in this next step of interpretation.

Minor points

(i) The abstract lists as a main aim to link intraplate volcanism to mantle flow. However, the analyses do not constrain mantle flow, only temperature. Lateral variations in temperature, which may relate to flow are not systematically discussed.

(ii) “While V_s is sensitive to pressure and temperature variations, it also depends on ... anelasticity...”. This is poorly worded. The variations of V_s with anelasticity contribute to the sensitivity of V_s to temperature and pressure (and possibly also composition, melts and other fluids).

(iii) “Figure 1a demonstrates that there is a striking visual correlation between intraplate volcanism and average shear wave velocity anomalies” This sentence needs clarification. As written it implies that intraplate volcanism occurs where there are V_s anomalies. But V_s anomalies cover most of the globe, while intraplate volcanisms does not.

(iv) Fig. 1 has quite binary colour scales that draw the attention to a specific contour that correlates reasonably with the boundary between areas with and without continental intraplate volcanism.

(v) Fig. 2d could use a legend

(vi) Why are the reference potential temperatures for seismic and geochemical estimates not consistent? This needs some motivation in the main text. Also these reference values need to be introduced before it is argued that in the two examples in Fig. 4 the T_p inferred is elevated.

Global Influence of Mantle Temperature and Plate Thickness on Intraplate Magmatism

P. W. Ball, N. J. White, J. Maclennan and S. N. Stephenson

Response to Reviewers' Comments

Reviewer 1

Reviewer 1 has written a relatively short one-page review which starts by noting that we “show that the distribution of Neogene and Quaternary intraplate volcanic rocks is correlated with seismic tomographic models” and that we “infer that temperature anomalies are responsible for the positive correlation”. They assert that “this is a major claim that may hold in some cases”. Thus the reviewer implicitly acknowledges that the topic of contribution is a significant one of global reach but they have concerns about whether or not our conclusions can actually be globally applied. Furthermore, this reviewer raises three significant issues, the first of which is the principal and substantive concern of this first review. (a) The reviewer queries whether we “really think that the location of Hawaii and Iceland... are really controlled by asthenospheric temperature anomalies in the 100–200 km range”. Or as restated, “is there no role for deep mantle plumes?” (b) The reviewer points out, in a related concern, that “many of the volcanoes... have distributions that are controlled by the African LLSVP”. This comment centers around the extent to which our results mesh with observed associations between mantle plumes and the edges of large low velocity provinces within the lowermost mantle. (c) The third concern specifically relates to the Galápagos Archipelago where Reviewer 1 points out that our results presented in Figure 4 are in conflict with the results of a receiver function study published by Byrne et al. (2015). Reviewer 1 concludes by stating that we need to explain why our “model of uppermost asthenospheric melting is a better way of interpreting... intraplate volcanism than deep plumes connected to LLSVPs”.

We are grateful to Reviewer 1 for pointing out and emphasizing these significant wider global connections and concerns. We take these general and specific concerns very seriously indeed and we have addressed them as carefully as possible as described below. In summary, we do not believe that our principal global results are actually in conflict with these widely published and widely held views that the conduits of many mantle plumes are rooted in LLSVPs within the lower mantle. Indeed, we view our results as being supportive and complementary in this regard. It is unfortunate that we did not make this complementary relationship clearer and we have now done so in the revised Ms. We have also carefully addressed the specific concern regarding the results that we present in Figure 4 regarding the Galápagos Archipelago.

Response to specific comments

(1) “Do the authors really think that the locations of Hawaii and Iceland, to name two, are really controlled by asthenospheric temperature anomalies in the 100–200 km range? Are Hawaii and Iceland melting from hot and shallow asthenosphere? Is there no role for deep mantle plumes that are anchored at the base of the mantle, LLSVPs? The author’s conclusion is in stark contrast with that reached by Scott French and Barbara Romanowicz (2015; *Nature*) that Hawaii’s present position is controlled by the northern edge of the Pacific LLSVP; here they showed the Hawaiian plume as a broad region of low seismic velocity that is continuously connected from the surface to the LLSVP. The conclusions of Ball et al. also contradict an important role for the Pacific LLSVP on the locations of Austral Cook, Easter, Galápagos, Marquesas,

Pitcairn, Samoa, Society Islands (e.g., Montelli et al., 2004; Torsvik et al., 2006; Hassan et al., 2015).”

We do not believe that our global analysis and results, which specifically address the uppermost mantle, contradict in any way the deep-seated mantle plume model outlined by Reviewer 1. In our view, both sets of views are complementary in that Reviewer 1 is concerned with the origin and control of deep mantle plumes whereas we are concerned with the specific way in which such plumes, once they arrive at the base of the lithospheric plate, generate basaltic magmatism and regional vertical motions. As stated by Reviewer 1 in their description of the Hawaiian plume, mantle plumes are commonly observed within tomographic models as broad regions of anomalously low seismic velocity within the upper mantle. Our conclusion that intraplate volcanism coincides with low seismic velocities in the upper mantle encompasses low-velocity regions generated by mantle plumes. For example, a significant correlation between the distribution of intraplate volcanism and shear-wave velocity at 100–200 km depths is shown in Figure 2d of the main text using the SEMUCB tomographic model. The SEMUCB model is used by French and Romanowicz (2015) to describe the presence of mantle plumes nucleating along LLSVPs. Therefore, we believe our results are complementary with, rather than in “stark contrast” of, the results of French and Romanowicz (2015).

Whilst we understand that many volcanic regions, such as Hawaii and Iceland, are generated above mantle plumes, in other regions such as Anatolia or Eastern Australia, the presence of a single identifiable mantle plume is less clear and other mechanisms, such as lithospheric delamination and edge-driven convection, are often invoked (Göğüş et al., 2017; Rawlinson et al., 2017). Since we have conducted a global analysis, which encompasses all intraplate volcanic provinces, any discussion of mantle plumes would not be applicable to the whole database. Therefore, we intentionally avoided a lengthy discussion of mantle plumes and their specific origins within our manuscript since to do so would discount a significant proportion of our database.

In conclusion, we do not think that our near-surface results in any way conflict with the deep-seated origin of mantle plumes.

(2) “Importantly, Williams, Mukhopadhyay, Rudolph, Romanowicz (2019) show that many of these volcanoes above the Pacific LLSVP have elevated primitive $^3\text{He}/^4\text{He}$; this is evidence against the model that they were formed by partial melting of shallow sub-lithospheric asthenospheric mantle.”

We agree with the first statement but we are unable to agree with the inference regarding partial melting since a deep lower mantle origin is compatible with subsequent shallow melting. It is well known that ^3He is both incompatible during mantle melting and cannot be generated within the Earth. High $^3\text{He}/^4\text{He}$ ratios are therefore associated with melting of primordial mantle material. This observation is often used to infer that the mantle material is sourced from the lowermost mantle. However, no melting will actually occur within a mantle plume until the solidus is crossed at depths < 200 km (Katz et al., 2003). Whilst the mantle material may indeed be derived from a primitive source within the lowermost mantle, it must be physically transported upward through the plume conduit to shallow depths before melting can occur. Thus, in our view, and in the view of the majority of igneous petrologists, high $^3\text{He}/^4\text{He}$ ratios cannot be cited as evidence that these melts were not formed by partial melting at the base of the lithosphere.

(3) On the other side of the world, many of the volcanoes considered by the authors have distributions that are controlled by the African LLSVP (e.g., Montelli et al., 2004; Torsvik et al., 2006; Hassan et al., 2015; Williams et al., 2019). These include those from the East African Rift (Afar, Kenya, Tanganyika), Azores, Cameroon, Canary

Islands, Cape Verde, Comoros, Crozet, Hoggar, St. Helena, Tristan da Cunha, and Iceland. Importantly, EAR volcanism is continuously connected to the African LLSVP at the base of the mantle via a broad low velocity seismic anomaly called the African Superplume (e.g., Mulibo and Nyblade, GRL 2013; GRL; Hansen, Nyblade, Benoit, EPSL 2012).”

We do not disagree with these important statements about the ultimate origin of plumes. We also agree (and several of us have published upon) the importance of the African LLSVP and the African Superplume and its role in generating shallower mantle temperature anomalies. Here, our analysis is concerned with understanding which processes control the distribution of melting within the uppermost mantle. Our principal conclusion is that the distribution and composition of intraplate volcanic rocks are controlled by temperature variations in the uppermost mantle. Mantle plumes are a key mechanism for transporting heat to the uppermost mantle and, as a result, mantle melting is often associated with mantle plumes. However, no melting will occur within a mantle plume until the solidus is exceeded at shallow depths (Katz et al., 2003). Therefore, the presence and whole-rock composition of mantle-derived melts can only provide information about mantle conditions at or above the solidus. It cannot be used to infer whether melting occurred within a plume which nucleated at the core-mantle boundary, the edge of an LLSVP, or a mantle discontinuity, or if melting occurred without the influence of a mantle plume. It is important to emphasize that we are making no assertions about the origin of deep mantle plumes which we in no way dispute—it is simply that the concerns of our Ms lie within the way in which melt is generating when the solidus is crossed at shallow depths.

(4) “The authors have inverted REE concentrations in Galápagos lavas to infer mantle potential temperature and lithosphere thickness and they show their results in Figure 4. Their inferred lithospheric thickness is 48 ± 3 km, and they cite agreement with Gibson and Geist (2010). More recent work gives 91 ± 8 km beneath the southeastern archipelago and 72 ± 5 km beneath surrounding regions, based on receiver function studies (Byrnes et al., 2015)”

The Reviewer is mistaken in their interpretation of the results presented by Byrnes et al. (2015), which are actually in complete agreement with our own results. Byrnes et al. (2015) interpreted the seismic discontinuity (or G discontinuity) at 91 ± 8 km and 72 ± 5 km depth to represent the maximum extent of anhydrous melting rather than the base of the lithosphere. We refer Reviewer 1 to the last line of the abstract of Byrnes et al. (2015) which states— “We conclude that the G discontinuity beneath the archipelago does not mark the boundary between rigid lithosphere and convecting asthenosphere.” We conclude that the results of this independent analysis of a very different dataset provide corroborative evidence that completely supports our own results of Figure 4.

Receiver function analysis is a technique that allows seismologists to map internal boundaries with sharp seismic impedance contrasts within the Earth. For example, receiver functions analysis is often used to estimate crustal thicknesses since there is a large difference in material properties between the crust and lithospheric mantle. The mantle on either side of the lithosphere-asthenosphere boundary is made of the same material and a diffuse rather than a sharp change in seismic velocity is expected along this boundary (Stixrude and Lithgow-Bertelloni, 2005; Priestley and McKenzie, 2006). Since the change in seismic velocity at the thermally-defined lithosphere-asthenosphere boundary is small, the ability of receiver function analysis to resolve the depth of this boundary is controversial. Analysis of oceanic bathymetry reveals that oceanic lithospheric thickness increases over time in a predictable manner (Parsons and Sclater, 1977). The Galápagos archipelago sits on top of oceanic lithosphere < 10 Ma (Bird, 2003). Using any of the well-established plate cooling

models, 10 Ma oceanic lithosphere has a thickness of ~ 50 km (Parsons and Sclater, 1977; McKenzie et al., 2005; Richards et al., 2018). Our lithospheric thickness estimate of 48 ± 3 km is completely consistent with these plate cooling estimates. Given that the lithosphere beneath Galápagos has been cooling above a hot mantle plume, a lithospheric thickness of 50 km can be considered an upper estimate. We believe these receiver function studies are imaging boundaries, such as a change in anisotropy or composition, within the mantle which are deeper than the lithosphere-asthenosphere boundary. This interpretation is corroborated by the results presented in Byrnes et al. (2015). We have chosen not to include this reference within our manuscript.

(5) “For this paper to go forward in Nature, Ball et al. need to explain why their model of uppermost asthenosphere mantle melting is a better way of interpreting the spatial distribution of intraplate volcanism than deep mantle plumes connected to LLSVPs.”

We completely agree with Reviewer 1 that many intraplate volcanic provinces are generated by mantle plumes. However, we do not believe that all intraplate volcanic provinces are necessarily generated by melting within mantle plumes or that all mantle plumes are necessarily connected to LLSVPs. We do not believe that other authors would make such claims for the same reasons. For example, our global compilation of intraplate volcanism includes samples from Eastern China, Mongolia, Russia, Western United States and Antarctica which are in some cases many thousands of kilometers from the edges of either LLSVP. Our analysis shows that intraplate volcanism is associated with regions of hot asthenospheric mantle beneath thin lithosphere, we have chosen not to speculate on the heat source for these hot regions as it is unlikely to be exactly the same mechanism globally.

Reviewer 2 - Yaoling Niu

Professor Niu has written a generous-minded review in which he expresses great enthusiasm and high regard for the global analysis that we have carried out. A key point is that he recognizes the novelty, significance and strength of the Manuscript which we think is a great vote of confidence. He considers our manuscript to be “an exciting piece of work, the kind of work that I wanted to do by myself for some long time”. Although Professor Niu “has some reservations about the interpretation” of the “significant correlation between La/Sm and ΔV_s ”, he considered it to be a “brilliant finding of far-reaching significance”. Therefore, Professor Niu recommended our manuscript be published with revision. We thank Professor Niu very much for his complimentary and carefully considered review. We have endeavoured to fully address in a very serious way all of his careful comments, especially those concerning the influence of lithospheric thickness variations.

Response to specific comments

(1) “Shear-wave velocities between 100 and 200 km depth positively correlate with elemental ratios that are regarded as proxies for melting – Be explicit what this positive correlation physically means. That is, higher speed correlates with lower extent of melting?”

We fully agree. We have now added an extra sentence to the abstract which explicitly states that higher predicted melt fractions coincide with regions of slower upper-mantle shear-wave velocities.

(2) “Our results are independently corroborated by marine stratigraphic observations indicative of rapid regional uplift – Be explicit in terms of physical meaning”

Our meaning was obscure. We agree with Reviewer 2 that the logic behind this statement is opaque and we have therefore changed “Our results are independently corroborated by marine stratigraphic observations indicative of rapid regional uplift” to “Finally, intraplate volcanism is often prevalent where elevated, but undeformed, marine sedimentary rocks show that Cenozoic regional uplift occurred”. Although this statement could be viewed as being about a different topic, namely dynamic topography, it is in fact of considerable importance to the wider linkage and integration of the topic.

(3) “In addition to temperature, the distribution and composition of volcanic rocks are significantly influenced by mantle composition, by lithospheric thickness – See Niu et al., (2011) and Niu et al., (2018).”

We agree and we have added Niu et al. (2011) as a reference for the importance of lithospheric thickness within this statement.

(4) “Since lithospheric plates translate across the globe and since sub-plate mantle circulation evolves as a function of time and space, we restrict our study to samples that erupted less than ten million years ago – This is reasonable, but caution is still necessary because 10 Myrs is a rather long time during which many things may have changed. For example, active volcanism only takes place on the Big Island of Hawaii, but no volcanism on any of the Hawaiian Islands to the north only a few Myrs older. The co-location will be less a problem with slow plate motion speed.”

We fully agree with Reviewer 2 that the filtering process in our initial submission only accounts for temporal variations in mantle structure. Over 10 Ma, fast-moving plates such as the Pacific or Australian plates can move > 1000 km (Argus et al., 2011). Therefore, we have modified our strategy and we have added an additional filtering process that only permits inclusion of samples which have moved < 400 km since eruption, based on present-day plate speeds (Rebuttal Figure 1a). We chose a value of 400 km since it is approximately the spatial resolution of global tomographic models used in this study (200–600 km; Schaeffer and Lebedev, 2013; Ho et al., 2016; French and Romanowicz, 2014; Ritsema et al., 2011). This additional filter removes the oldest samples from the Australian, Cocos, Juan de Fuca, and Pacific plates (Rebuttal Figure 1b). Whilst we have set the upper limit for distance moved to 400 km, varying this upper limit between 100–1000 km makes negligible difference to the observed correlation (Rebuttal Figure 2d). We have now corrected the main text to include discussion of this additional filter.

(5) “bands positioned on continental lithosphere away from thicker cratons – Yes, see Niu (2020)”

We are glad the reviewer and their previous scientific contributions agree with our observations. However, since this statement is a description of our database rather than an interpretation we feel that an additional reference at this point within the text is unnecessary.

(6) “Western China – Eastern China!”

We thank the reviewer for spotting this typographic error and we have updated the manuscript accordingly.

(7) “Figure 1a demonstrates that there is a striking visual correlation between intraplate volcanism and average shear wave velocity anomalies – It is a good one, but not striking as stated. It is not obvious in the cases of New Zealand, Mongolia.”

Figure 1: Filtering the database by plate speed. (a) Segmented Mollweide projection of globe showing present-day plate speeds (Argus et al., 2011). Green circles = loci of intraplate magmatic samples < 10 Ma, which have moved < 400 km since eruption (see Supplementary Materials Tables 1 and 2 for compiled database). (b) Green circles = loci of intraplate magmatic samples < 10 Ma which have moved > 400 km since eruption.

Figure 2: **Correlation between La/Sm and ΔV_s for different filtering parameters.** (a) Value of R calculated between La/Sm and ΔV_s as function of minimum number of samples in 1° bin; colored lines = 4 different global tomographic models (SL2013sv, CAM2016Vsv, SEMUCB-WM1 and S40RTS; Schaeffer and Lebedev, 2013; Ho et al., 2016; French and Romanowicz, 2014; Ritsema et al., 2011). Database is subdivided into 1° bins weighted by $\cos \phi$; dotted black line = chosen value for correlation shown in Figure 2c of the main text; dashed gray line = value of R that can be resolved from zero at significance level = 0.001 given number of bins. (b) Value of R as function of minimum content of MgO (wt%) for each sample. (c) Value of R as function of maximum age of each sample. (d) Value of R as function of maximum distance from point of eruption based on present-day plate speeds (Argus et al., 2011). (e) Value of R as function of bin size. (f) value of R as function of depth.

We partly agree. Whilst we concur with Reviewer 2 that the visual association is not completely perfect, we believe that since $\sim 90\%$ of intraplate volcanism is above regions of negative ΔV_s use of the word striking is warranted. It is a dramatic and important result of some significance.

(8) “Also, is it appropriate to treat the Red Sea as “intra-plate” setting? It has mature seafloor and separates the Arabic plate from the African plate with a full separation rate of > 20 mm/yr.”

We partly agree on this technical point. Our dataset includes eight Red Sea samples, however, these data are collected from seamounts within the Red Sea which are thought to be derived from the Afar mantle plume (Volker et al., 1997). We have included analysis within the Supplementary Materials where all samples adjacent to active spreading centres are removed, a significant correlation between La/Sm and ΔV_s at 150 ± 25 km depth is still observed ($R = 0.6$; Supplementary Figure 4).

(9) “average shear wave velocity anomalies, ΔV_s , between 100 and 200 km depth – What is the vertical resolution? I gather the correlation would be better if the depth interval is restricted to 80–150 km?”

It is important to emphasize that our global results are not altered or impacted upon if shear-wave velocity anomalies are averaged over a range of possible depth intervals or, indeed, not averaged at all. We have carried out exhaustive testing to double-check the robustness of our inferences with respect to this variation. Within global tomographic models, crustal structure is often poorly resolved or imposed *a priori*, and so real and modeled crustal structure can significantly vary. Crustal thinning can occur in areas where crustal thickness has been underpredicted, which can sometimes give rise to spuriously low velocities at mantle depths. Since crustal thickness is generally ~ 0 –70 km and since the spatial resolution of tomographic models is ~ 25 –50 km, these erroneous slow velocities can sometimes extend to ~ 100 km depth (Priestley and McKenzie, 2013). Therefore, we have deliberately chosen not to put emphasis on results generated from tomographic models at depths < 100 km.

The vertical resolution of all the tomographic models included within this study is ~ 25 –50 km (Ritsema et al., 2011; French and Romanowicz, 2014; Schaeffer and Lebedev, 2013; Ho et al., 2016). The optimal correlation between ΔV_s and La/Sm is achieved when averaging between 100–150 km. This result is an important one. However, we have now altered our study to use 150 ± 25 km depth in order to completely avoid variations in lithospheric thickness affecting shear wave velocities observed beneath intraplate volcanic provinces. This change does not alter our conclusions. We have now included a full description of the SL2013sv model as well as its vertical and spatial resolution within a revised Methods section. A description of the other tomographic models mentioned within this study can now be found within the Supplementary Materials.

(10) “Intraplate volcanism is almost completely absent in regions where ΔV_s is positive – Yes, almost!”

We are glad the reviewer agrees with our statement, which is a significant one.

(11) “. In the continental realm, it is clear that the distribution of intraplate volcanism is closely associated with lithosphere that is thinner than 100 km – Yes, agree!”

We are glad the reviewer agrees with our statement, which is a significant one.

(12) “Only bins with > 5 values of La/Sm – Unclear!”

We agree that this statement is opaque. We have expanded “Only bins with > 5 values of La/Sm” to “Only bins with > 5 samples analysed for La and Sm concentrations” to provide extra clarity.

(13) “averaging smaller numbers of analyses may not be representative of geochemical variability – True! But surprisingly in most cases (if we deal with global data), the averages of smaller numbers of samples are as significant despite the relatively large variability (or s.d.)”

We are glad the reviewer agrees with our statement.

(14) “This relationship between La/Sm and ΔV_s indicates that the chemistry of intraplate volcanism is modulated by asthenospheric temperature since smaller values of La/Sm are associated with negative V_s anomalies. – This is expected without having to invoke temperature variation. The correlation is significant at > 99% confidence level while a factor of up to 3 La/Sm variation at a given V_s also exists. Both are likely real! So, the first order correlation between the erupted basalts may actually reflect (or inherited from) the amount of “trapped” interstitial melt in the mantle source region (deeper than the dry solidus, but shallower than the volatile-bearing solidus); the average depth of 150 km with highest R_s makes sense. The greater melt retention (or high melt porosity, melt fraction), (1) the lower La/Sm in such melt and (2) the lower V_s of the bulk mantle. V_s is directly proportional to the shear modulus, which is much more sensitive to the amount of the melt porosity than temperature variation! So, the large La/Sm variation at a given V_s may (1) reflect average source compositional variation from one location to another (variably depleted or refertilized source lithologies, volatiles as well as major elements and modal mineralogy), which may also (2) affect the properties of extent of melt retention (melt porosity).”

We partly agree with some, but not all, of these statements. In essence, Reviewer 2 has suggested that the correlation between La/Sm and ΔV_s may not represent temperature variations within the mantle, and instead may result from variable melt retention within the mantle, or from compositional variations within the mantle. First, we will address the suggestion that variable melt retention could generate the correlation between V_s and La/Sm. The amount of melt present during partial melting is generally regarded as being small ($\sim 0.1\%$) since buoyancy-driven melt extraction is regarded as efficient based upon isotopic systematics (McKenzie, 2000). As Reviewer 2 suggests, the presence of melt within the mantle will reduce V_s through poroelastic effects (Hirschmann, 2010). However, reduction of V_s due to poroelastic effects scales as a function of melt fraction and when the melt fraction within the mantle is small ($\sim 0.1\%$) the reduction in V_s is very small (Takei, 2017).

Recent experimental studies have shown that the reduction in V_s below the dry peridotite solidus, which was previously attributed to the presence of trapped melt, can instead be explained by anelastic effects (McCarthy et al., 2011; Yamauchi and Takei, 2016). This development of understanding is a significant one. At high homologous temperatures, the mantle ceases to behave elastically and there is a rapid decrease in seismic velocity. These anelastic effects are included within the V_s -to- T parameterization we have chosen to use (Priestley and McKenzie, 2013; Yamauchi and Takei, 2016; Hoggard et al., 2020). We do not consider melt retention as a possible explanation for the correlation between La/Sm and ΔV_s , especially at depths well below the dry peridotite solidus (150 km). We refer Reviewer 2 to Takei (2017), which is an excellent review paper on the effects of partial melting and anelasticity on seismic velocity.

Secondly, we will address the suggestion that compositional variations control the correlation between ΔV_s and La/Sm. Depleted sources have reduced incompatible element and volatile concentrations and so melting of a more depleted mantle at an equivalent potential temperature will yield higher La/Sm concentrations. Shear waves propagate faster through depleted mantle and so

a positive correlation between La/Sm and ΔV_s is expected to result from compositional variations. Since a positive correlation is observed between La/Sm and ΔV_s within our data, it is reasonable for Reviewer 2 to suggest that this correlation may be controlled by mantle composition. However, results from comparing geochemical indicators of mantle composition, such as ϵNd and V_s , suggest that the opposite is in fact true (Rebuttal Figure 6d,g). We observe a negative correlation between ϵNd and ΔV_s , and so the most depleted mantle is associated with the slowest shear-wave velocities. Therefore, the correlation between La/Sm and ΔV_s cannot be accounted for by compositional variations alone. We have now made this point clear by appealing to the principal component analysis section of our revised manuscript. Finally, within our principal component section, we conclude that P_3 represents variable enrichment by remobilizing/melting metasomatized regions of the lithosphere or enrichment of the mantle source through subduction processes. Since P_3 does not correlate with ΔV_s , we conclude that these processes are not more prevalent in areas of lower ΔV_s .

(15) “Figure 2d – Please put legend in (d) which model is which”

We agree and have now added a legend to Figure 2.

(16) “However, temperature changes have a much greater effect upon V_s than does mantle composition – The lid effect is far more significance! See Niu et al., (2011)”

We partly agree. Niu et al. (2011) use major elemental concentrations to show that composition varies as a function of oceanic lithospheric thickness. They state that regions with a thick lithospheric lid have low melt fractions and strong garnet signatures whilst regions with thin lids have high melt fractions and weak garnet signatures. In their analysis, lithospheric thickness was assumed to vary purely as a function of plate age, and all global samples from a single lithospheric thickness are averaged together regardless of the number of samples in each location.

Our analysis concerns not only geochemical data but also, critically, geophysical data. By including tomographic constraints, it is possible to account for the natural variation in plate thickness that exists at a single age and then compare geochemical proxies for melt fraction and garnet signature (i.e. principal components 1 and 2, respectively) to ΔV_s , a proxy for mantle temperature. Additionally, since our data is spatially binned, we account for sampling bias between volcanic provinces and include a greater number of individual bins within our analysis. Finally, Niu et al. (2011) is limited to the oceanic realm whereas in our study we also include continental regions. We find that ΔV_s correlates with melt fraction. However, we find that the strength of the garnet signature does not correlate with either melt fraction or with ΔV_s . We agree that lithospheric thickness exerts a first order control on the presence of melting (i.e. no melting occurs under very thick lithosphere regardless of asthenospheric temperature; Figures 1b and 2b of the main text). By combining geochemical and tomographic observations, our analysis indicates that in locations where melting occurs, asthenospheric mantle temperature variations exert the primary control on melt composition. Whilst we agree with Reviewer 2 that lithospheric thickness variations influence the extent of melting, and therefore La/Sm values, we believe our analysis shows that this is a secondary effect.

Nonetheless, we believe our conclusions, as they were stated within the original submission, were overly simplistic. To address this point by Reviewer 2, we have expanded our principal component analysis section and we have now added a discussion about the influence of lithospheric thickness variations on ΔV_s . We have also stated that the reduction in correlation between Figures 2c and 5a in the main text may relate to the removal of this secondary signal.

(17) “shear waves propagate more slowly through depleted mantle – Why? With the presence of volatiles and liquid/fluid phases (melt), the V_s would be reduced because of

reduced shear modulus. Depleted source would be depleted in volatiles and would be less likely to form a melt phase at any given condition. If you are talking about major elements, then the depleted source would have high Mg# and low Al_2O_3 (thus less garnet). As the result, the bulk rock would have reduced density, which will facilitate V_s propagation (density is on the denominator). Hence, shear waves are expected to propagate faster, not slower, through the depleted mantle.”

We agree that mantle depletion leads to an increase in shear-wave velocity (see Lee, 2003; Schutt and Leshner, 2006). We have now amended this error within the manuscript.

(18) “However, temperature changes have a much greater effect upon V_s than does mantle composition – This is possible, but the presence of a melt phase (tiny fraction) = melt retention = melt porosity, would have significantly greater effect.”

We disagree with this statement by Reviewer 2 and refer them to our explanation outlined in our reply to Comment (14).

(19) “Thus we infer that sub-lithospheric temperature variations are the dominant control on the positive relationship between melt fraction (i.e. La/Sm values) and ΔV_s . – This would be debatable!”

We partly disagree. Whilst we believe the above statement to be correct, we admit that it is presumptuous and comes before the argument has been fully formed since we have not addressed the effect of lithospheric thickness variations on melt composition at this point in the manuscript. We have now changed “sub-lithospheric” to “upper mantle” so that the statement encompasses both sub-lithospheric temperature and lithospheric thickness variations. In this way, we have improved our line of argument.

(20) “P2 is almost completely dominated by variations in Yb alone – This is not obvious from Fig. 6.”

By mentioning Figure 6, we believe that Reviewer 2 is referring to Figure 6 of the Supplementary Materials. We do not cite Figure 6 as evidence for the statement highlighted by the reviewer and we are therefore confused by this comment. However, both Figure 3 in the main text and Figure 6 in the Supplementary Material clearly show that for principal component 2, P_2 , Yb is the only element weighted more or less than ± 0.2 and therefore clearly dominates variations in P_2 ($\text{Yb} > 0.7$). A few sentences earlier we state “Note that, while fractional crystallization can increase La/Sm values, the range of La/Sm values observed is too great to be controlled by fractional crystallisation alone (Supplementary Figure 6)”. If Reviewer 2 is instead referring to this sentence, then the absence of a correlation between MgO concentrations and P_1 demonstrates that the variation in La/Sm cannot be controlled by fractional crystallisation alone. This principal is explained in lines 103–106 of the original Supplementary Materials.

(21) “Subduction zone melting is commonly enriched in Ba and K but depleted in Nb, whilst partial melting of metasomatized lithospheric mantle often exhibits anomalously low concentrations of K and Ba (Späth et al., 2001). – This is not true! For example, see this paper - Huang et al. (2010). Up to 5.23 wt% K_2O and 1116–1566 ppm Ba.”

Amphibole and phlogopite are absent within anhydrous peridotite but can be present within metasomatized mantle. Large ion lithophile elements, such as Ba or K, are compatible in amphibole and phlogopite (McKenzie and O’Nions, 1995). If amphibole- or phlogopite-bearing mantle partially melts, and these phases are not exhausted during melting, then more Ba and K will be retained within the mantle than during melting of hydrated peridotite (Späth et al., 2001). Therefore, low

normalized concentrations of Ba or K relative to high field strength elements such as Nb can often result from melting of metasomatized lithosphere (Späth et al., 2001). For a thorough analysis, see Figure 8 of Pilet et al. (2011), which shows trace element compositions predicted for a wide variety of metasomatized lithospheric melting models (see Figure 8d,j for Ba/Nb). Whilst we believe our original statement to be correct, we agree with Reviewer 2 that the opposite (i.e. high Ba and K resulting from melting of metasomatized mantle) can also occur. To improve this section and make our statement more broadly applicable, we have changed “often exhibits anomalously low concentrations of K and Ba” to “can exhibit anomalously high, or anomalously low, normalized concentrations of K and Ba relative to Nb”. We have also added Huang et al. (2010) as a reference at the end of the sentence.

(22) “Thus depth of melting and metasomatism are evidently not the primary controls of incompatible element concentrations in intraplate mafic igneous rocks – Probably true! BUT, if the depth of melting is not important, what would be the effect of temperature that you argue for? This needs rigorous elaboration. Metasomatism that we often invoke to use refers to the source history prior to the recent major melting event”

We regret the clumsy wording within this sentence. When the temperature of the mantle is increased, the depth at which melting initiates is deepened. Therefore, a change in the depth range over which melting occurs as a direct result of changing temperature. By “depth of melting” we meant the relative contributions of melting within the spinel- and garnet-peridotite fields since P_2 , which is sensitive to the presence of garnet within the source region, has low variance. We have therefore changed the sentence to state “the relative contributions of melting within the spinel and garnet stability fields” rather than “depth of melting”. By metasomatism we do not mean the degree to which present-day melts metasomatize the mantle. Rather, we mean the contribution of melts from metasomatized mantle sources within any given melt. Therefore, we have changed “metasomatism” to “the degree of contamination by lithospheric melts”.

(23) “The global distribution of intraplate magmatism combined with the positive correlation between La/Sm and V_s both imply that asthenospheric temperature variation plays a role in melt generation. – The temperature effect is neither obvious nor implied by the data; preferred interpretation? I do not object the effect of mantle temperature variation, but the reasoning seems to have something missing!”

We disagree. Within Figure 2, we show a correlation between La/Sm and ΔV_s at 150 ± 25 km depth. ΔV_s at 150 ± 25 km does not correlate with lithospheric thickness variations and is therefore directly related to asthenospheric temperature variations. Whilst compositional variations can affect both ΔV_s and La/Sm, our principal component analysis demonstrates that compositional variations do not control the correlation between these data. Therefore, the correlation we observe between La/Sm and ΔV_s at 150 ± 25 km depth implies that asthenospheric temperature variation does indeed play a role in melt generation.

(24) “Galápagos volcanic islands located on the Pacific plate – No! Not on the Pacific plate, but Nazca plate subducting beneath South America.”

We thank the reviewer for identifying this error and have updated the manuscript accordingly.

(25) “Geochemical modelling – You need to tell what REE sources values are used. Is a uniform mantle REE source composition assumed?”

In the Methods section, we state that “The composition of the mantle source region used in this modeling procedure varies as a function of the average value of ϵNd calculated for each province.

Its composition is calculated by linearly mixing between a primitive mantle source, PM, and a depleted MORB mantle, DMM, in order to match the observed value of ϵNd (Supplementary Table 5)". Supplementary Table 5 contains the rare earth element concentrations for primitive and depleted mantle.

(26) "Ambient mantle potential temperatures are assumed to be 1312 °C and 1331 °C for geochemical and tomographic estimates of ΔT_p , respectively – Why different T_p s are assumed? It is better explained here."

We agree and we have now explained the difference between the two reference temperatures within the main text.

(27) "As expected, there are concomitant increases and decreases in calculated values of potential temperature anomalies, ΔT_p , and lithospheric thickness, a – This is actually NOT expected. Why should there be an inverse correlation between mantle T_p and lithosphere thickness? There is no reason. Mantle T_p is the surface projection of mantle temperature along the adiabat, which related to the mantle convection, whereas lithosphere thickness would have long geological history that neither controls mantle convection nor is a consequence of present-day mantle convection. This needs explanation in terms of understandable physics."

We partly disagree mostly because there seems to have been a misunderstanding concerning the observation that we were trying to convey. By the statement above, we meant that there are concomitant increases and decreases between geochemical and tomographic potential temperature anomalies, and concomitant increases and decreases between geochemical and tomographic lithospheric thickness estimates. We did not mean to imply a correlation *per se* between potential temperature, T_p , at 100–200 km depth and lithospheric thickness. As can be seen in Figure 2e of the main text, there is no significant correlation at 150 ± 25 km depths between ΔV_s and lithospheric thickness estimates beneath intraplate provinces. We have therefore changed the above statement to "As expected, there are concomitant increases and decreases in geochemical and tomographic ΔT_p estimates (Figure 5e). Geochemical and tomographic estimates of lithospheric thickness also broadly agree along the transect (Figure 5f)".

Whilst this comment by Reviewer 2 stems from our poorly constructed sentence, we feel that it is important to explain that a global correlation between T_p at 150 ± 25 km and lithospheric thickness is expected. T_p is tool for comparing the spatial heat content of the mantle that accounts for compression. The use of T_p cannot be taken as an indication that we are discussing the convecting mantle only (for a full explanation of T_p we have included an extract from McKenzie and Bickle, 1988, which can be found at the end of this rebutted comment). In regions where the thermal boundary layer extends to depths > 125 km, the base of the lithosphere will be included in our calculation of average T_p at 150 ± 25 km depths. Therefore, in regions where the thermal boundary layer is > 125 km thick, we expect the coldest T_p estimates to be associated with the thickest lithosphere. Since the thermal boundary layer is < 125 km thick beneath the majority of intraplate provinces we do not see a strong correlation between tomographic T_p and lithospheric thickness within our results (Figure 2e in main text). To address this confusion, we have added a statement to the end of the description of Figure 5 within the main text— "Note that in regions where the lithosphere is > 125 km thick, a correlation between ΔT_p and lithospheric thickness at depth of 150 ± 25 km is expected".

McKenzie and Bickle (1988) – "It is commonly necessary to compare the heat content of material at different depths. Such a comparison is straightforward if the material is incompressible, since the difference in heat content is then simply proportional to the difference in temperature. But

this simple result fails when the material is compressible. Those who ascend mountains are well aware of the adiabatic temperature gradient which exists because air is compressible. The heat content of two air masses is only proportional to the temperature difference between them if they have been brought to the same pressure, which must be done reversibly to conserve entropy. In meteorology and oceanography, this problem has been well understood for many years. Rather than using the entropy of a mass of fluid to define its heat content, it is common practice in these subjects to define a new temperature, called the potential temperature, which is the temperature the fluid mass would have (hence the term ‘potential’) if it were compressed or expanded to some constant reference pressure.”

(28) “Thirdly, inverse modeling demonstrates that the composition of erupted basalts are best matched by assuming isentropic melting of warm asthenosphere beneath thin lithosphere, which is corroborated by independently calculating asthenospheric temperatures and lithospheric thicknesses from shear wave tomographic models. – There may be some artefacts? Thin lithosphere means decompression to a shallow level, i.e., $F \propto P_o - P_f$ (Niu et al, 2001; Niu & Green, 2018), i.e., the lid effect: Niu et al., (2011) and Niu & Green, (2018). So, thin lithosphere lid is associated with higher extent of melting and relatively lower incompatible elements (also lower La/Sm) in the basalts, whereas thick lid is associated with lower extent of melting and relatively higher incompatible elements (also higher La/Sm).”

We agree with Reviewer 2 that lithospheric thickness variations can impact upon the melt fractions, and therefore incompatible element concentrations, generated. However, our inverse modeling approach encompasses a larger suite of rare earth elements, not just La and Sm, and it solves for both potential temperature and lithospheric thickness (see Figure 4). By solving for lithospheric thickness, we have acknowledged and accounted for the possible artifact generated by the lid effect. We believe the reduction in correlation coefficient between Figure 2c and Figure 5a in the main text may represent the removal of the artifact provided by the lid effect on the correlation between composition and temperature. We have added an additional statement to our discussion detailing this interpretation.

To address this comment, we have expanded the principal component analysis section within the main text to include synthetic principal analysis results outlined in Comment 26 of our reply to Reviewer 3. We have also included additional analysis showing that ΔV_s represents asthenospheric rather than lithospheric variations. We have also expanded our discussion comparing geochemical and tomographic temperature estimates.

(29) “global study of the mid-oceanic ridge system, which showed that shear-wave velocity anomalies correlate with Na_{90} , a geochemical proxy for ridge temperature – This is interesting! Objectively, the Na_{90} argument is incorrect (see Niu, 2016). This is not a matter of personal preference or choice, but my objective scientific analysis”

We partly agree. Niu (2016) shows that Na_{72} , a similar proxy to Na_{90} , correlates with ridge depth, this finding is in agreement with the first-order observations of Dalton et al. (2014). Where these studies diverge is in interpreting the cause of this correlation. Niu (2016) state that the correlation between Na_{72} and ridge-depth is compositionally controlled. They argue that a dense fertile mantle source beneath a mid-oceanic ridge will lead to high Na_{72} and a deep ridge compared to a depleted source. Instead, Dalton et al. (2014) argue that the correlation between Na_{90} , ridge-depth, and V_s is thermally controlled. Cold mantle beneath a mid-oceanic ridge will lead to high Na_{90} , a deep ridge and relatively fast V_s anomalies compared to hot mantle.

Given the strong correlation between ridge depth and ΔV_s , if ridge depth is controlled by mantle

compositional variations as suggested by Niu (2016), shear waves must propagate slower through depleted mantle. However, many studies argue that the opposite is true (Lee, 2003; Schutt and Leshner, 2006). Furthermore, thermal models of the upper-mantle estimated from tomographic models can reasonably reproduce ridge depth variations without incorporating mantle compositional variations (Richards et al., 2020). We therefore do not believe the effect of mantle temperature on this correlation can be completely discounted, especially in areas where mantle plumes interact with mid-oceanic ridges. However, to inform the reader that this topic is one of active debate, we have altered this section of the manuscript from “Significantly, our results complement those obtained by a global study of the mid-oceanic ridge system, which showed that shear-wave velocity anomalies correlate with Na_{90} , a geochemical proxy for ridge temperature (Dalton et al., 2014).” to “Significantly, our results complement those obtained by a global study of the mid-oceanic ridge system, which shows that shear-wave velocity anomalies correlate with major element compositions and with axial ridge depths (Dalton et al., 2014). It has been pointed out that these correlations could also be influenced by mantle compositional changes beneath mid-oceanic ridges (Niu, 2016).” We hope that the reviewer is satisfied with this more nuanced statement.

(30) “ $O(10\text{--}100)$ km – Explain this O .”

In many scientific disciplines O is an accepted abbreviation of “of order”. We concede that O is not generally used in the geochemical and geophysical literature and have therefore replaced O with the long-form equivalent.

(31) “These spatial resolution issues may explain the small number of intraplate bins that fall on areas of positive ΔV_s – Both New Zealand and Mongolia are such exceptions?”

The North Island of New Zealand is above a subducting slab. Lateral smearing of fast velocities from this subducting slab may be responsible for the fast velocities at 100–200 km depth across New Zealand. Mongolia is not one of these exceptions since the majority of Mongolia’s intraplate volcanism coincides with negative ΔV_s at 100–200 km depths.

(32) “Nevertheless, we are struck by the simplicity and self-consistency of our global results which imply that the distribution and chemistry of intraplate magmatism are controlled by mantle dynamics. – See above comment of the unexpected inverse correlation of T_p and lithosphere thickness”

We stand by this statement since we believe we have adequately explained that we did not mean to imply a correlation between T_p and lithospheric thickness within the discussion (see Comment 26).

(33) “emergent marine strata – what sort of marine data?”

By “marine strata” we mean sedimentary rocks deposited under marine conditions. If these rocks are now found at elevations well above sea level, it is a clear indication that uplift has occurred. To make this point clearer to the reader, we have changed “strata” to “sedimentary rocks” or “deposits”.

(34) “ emergent marine strata in regions where intraplate volcanism predominates and where continental lithosphere is anomalously thin – Hard to follow! You are comparing marine data with thin continental lithosphere.”

By marine strata, we mean sedimentary rocks that were deposited within marine conditions. A significant proportion of marine sediments are deposited on the continental shelf above continental

rather than oceanic lithosphere. Therefore, since we have now changed marine strata to marine sedimentary rocks we hope this section is now clearer for the reader. The linkage is an important one.

(35) “What may have caused the lithosphere thinning in these places? Here is my version of continental lithosphere thinning, including some places also mentioned in Niu (2015; See Figures 10-15), which is summarized here: <http://www.mantleplumes.org/Hydration.html>”

There have been numerous such proposals, most of which are too difficult to test at the moment. Whilst we are interested in the mechanisms behind lithospheric thinning, we believe discussion of these mechanisms is beyond the scope of our study.

(36) “Removal or thinning of lithospheric mantle produces regional uplift – No! Basal removal or lithosphere thinning will NOT cause uplift, but if anything, will lead to subsidence due to Airy isostasy (Niu, 2019).”

We completely disagree with this statement by Reviewer 2. Unfortunately, the citation referenced by the reviewer is a conference abstract so we cannot fully assess the argument presented. However, we have expanded our argument and we explain why thinning of the lithospheric mantle causes uplift in the revised Manuscript. We believe the reviewer is referring to tectonic thinning of the lithosphere, where both the crust and lithospheric mantle are thinned together. When the whole lithosphere is thinned, subsidence will occur as the density increase from crustal thinning outweighs the density decrease from lithospheric mantle thinning (McKenzie, 1978). However, we specifically state “lithospheric mantle thinning” rather than “lithospheric thinning” in our manuscript. In areas without significant depletion, the lithospheric mantle is denser than the asthenospheric mantle. Therefore, thinning and thickening of the lithospheric mantle generates uplift and subsidence, respectively. The relationship between oceanic bathymetry and plate age is a common example (Parsons and Sclater, 1977). As the oceanic plate cools, lithospheric mantle beneath it thickens and the plate subsides. There is plenty of precedent for the concept that lithospheric mantle thinning generates uplift (e.g., Bird, 1979, which has > 1000 citations) and we provide a simple mathematical equation within the manuscript. We therefore believe our current statement to be correct and appropriate.

Reviewer 3

Reviewer 3 encouragingly states that this manuscript “describes an interesting combined analysis of seismic tomography and REE trends”. They also state that “it is encouraging that the two approaches yield correlated estimates of asthenospheric temperatures”. These statements are positive endorsements of our global approach and of the results that we have obtained. However, this reviewer feels that our manuscript is “poorly organised” and that the results were “insufficiently discussed and explored”. We are grateful to Reviewer 3 for their extremely thorough, thought-provoking and very helpful review. They have made a significant number of detailed comments and criticisms which we have dealt with as carefully as possible during extensive revision of the manuscript. Although we disagree that the submitted manuscript was poorly organized, we have taken this comment seriously and tried to improve the clarity and readability of the revised manuscript.

Response to specific comments

(1) “The manuscript is relatively poorly written and organised. For most of the methods the reader is referred to other papers, which makes it difficult to assess the robustness of the results.”

We disagree with this statement. It is hard to be completely objective but we believe that our manuscript is well written and organized. Nevertheless, we take all reviewer discussion seriously and we have paid especial attention to the standard of writing, legibility and organization during the revision process. In terms of methodological referral, we did carefully and extensively describe the geochemical melting model and statistical methods that underpin our thesis, despite the fact that both suites of techniques are well-described in the literature. However, we agree with Reviewer 3 that the methods presented within the manuscript could be more comprehensively and thoroughly presented. Consequently, we have added two substantial new parts to the methods section. The first describes details of the SL2013sv tomographic model. The second presents the procedure for converting shear wave velocities into estimates of temperature. We emphasize that both of these descriptions could be obtained from the original recently published literature but we are conscious of the need to present a thorough and careful case so that the robustness of our results can be recognized. Finally, we have also gone to considerable lengths to reference all the data and observations that we exploit and we have provided references for previous case studies where similar methodologies have been applied. We hope that these additions satisfy the requirements of the referee within the constraints available to us.

(2) “Issues raised in the introduction do not link well to those addressed in the analyses.”

We largely disagree with this statement. In Paragraph 1 of the introduction, we explain the necessity of trying to understand how the mantle has changed through time. We state that there are two important indirect observations of mantle structure: seismic tomography and igneous geochemistry. In Paragraph 2, we outline the uncertainties for each set of observations when they are used in isolation. In Paragraph 3, we clearly state that our objective is to closely combine these observations in order to understand mantle structure. It is this combination of two disparate fields that we believe to be the strength of our approach.

In the analytical portion of our manuscript, we directly compare geochemical and tomographic observations. First, we compare the spatial distribution of intraplate magmatism against seismic tomographic models. Secondly, we detail and interpret correlations between shear-wave velocity and the geochemical composition of intraplate volcanic rocks. Thirdly, we use these datasets to estimate mantle structure. Therefore, we believe our introduction links well with the analyses presented. Nevertheless, we have striven during the revision process for greater clarity of purpose and we hope that this reviewer will be more satisfied with our level of presentation.

(3) “The conclusions provide some method limitations and discuss dynamic topography rather than wrapping up the main results from the analyses.”

We understand the reviewer’s request for a summarizing conclusion. We do, in fact, summarize our findings through the manuscript, recapping at the start of each section, for example at the start of Section 4. Nonetheless, we believe that it is important to provide an explanation of the geodynamic implications of this work for the wider community since we anticipate that it will be of use to a wide range of geoscientists. In putting together this manuscript, we chose to closely follow the *Nature* style guide. We refer Reviewer 3 to the “How to Write a Paper” section of the *Nature* website:

www.nature.com/nature-research/for-authors/write#how-to-write-a-scientific-paper.

The following extract is written in the “Elements of Style” section – “Finally, a word about concluding paragraphs. It is commonly advised that a paper should begin by stating what will be said, continue by saying what is to be said, and then conclude by summarizing what has been said. This is bad advice that recommends lazy composition. Conclusions are not mandatory, and those that merely summarize the preceding results and discussion are unnecessary (and, for publication in *Nature Physics*, will be edited out). Rather, the concluding paragraphs should offer something new to the reader.” Regarding our discussion of dynamic topography, we think that it is important to link our geochemical and seismologic results with dynamic topographic observation which provide a further means for testing and closing the loop. Thus we do not regard dynamic topography as just an afterthought: it is integral to our thesis.

(4) “In addition, there is a very large supplement with a lot of results that are further results of the analyses rather than included to support the content of the main text. I am not sure that Nature Communications offers the best format to publish this work.”

We disagree. Our manuscript is a comprehensive global analysis and we believe that it is both important and normal practise to include all analytical datasets and results in the Supplementary Material. Here, size is not a particular concern as far as the journal is concerned. In our global analysis, it is obviously not possible to show all of the data controls. For example, whilst we include two geochemical modeling results in Figure 4 of the main text, we do not want readers to think we have “cherry picked” these results. For full and open transparency, we therefore include all modeling results within the Supplementary Materials. Because of the global scale of our study, these additional results take up 13 pages. Similarly, we believe it is important to include tables and figures outlining our global database as well as a complete reference list. These tables, figures and reference list take up > 100 pages within the Supplementary Materials. Given the size of these data and additional figures, there is no journal format available where this material would be included within the main text, nor should there be. Therefore, we do not believe the size of our Supplementary Materials should be taken as an indication that this study belongs in a longer journal format. In fact, we believe that the analysis and conclusions of our study can be explained clearly and concisely in a manner that is ideal for this short-format journal article. Furthermore, *Nature Communications* limits the number of figures within the main text to 10. We currently have 5 figures within the main text and if the reviewer considers any of our additional analyses essential for the story they can easily be included. We do not think that it is necessary to expand the number of figures in this way since all of our data and analyses are available in an easily accessible manner.

(5) “It does not become very clear what the main new results are. Some of the results are rather trivial (intraplate volcanism tends to occur where plates are thin), other results are only partially discussed (what is the relation between potential temperature and plate thickness).”

We do not think that our results are trivial and we believe that the quantitative connections that we have made between geochemical analysis/modeling and calibrated seismic tomography are novel, significant and of global reach. We believe that the main results of our paper are as follows:

- We have compiled a global intraplate database of > 20,000 igneous samples, each sample is tied to a geographic location and age range. This new and important database will be made publicly available for use by the community.
- We show a clear global association between locations of Neogene-Quaternary intraplate volcanism, negative shear-wave velocity anomalies and lithospheric thickness. While this correlation may be expected, it is not trivial since we do not know of any previous studies which

show this relationship using a global database. Therefore, we strongly feel that this is a significant result of our study. Even if such an association is expected by some, it is an excellent test of the accuracy and resolution of tomographic and lithospheric thickness models.

- We show a clear correlation between ΔV_s at 100–200 km depths and La/Sm, a proxy for melt fraction. This result is very significant and new since it shows that for high-MgO volcanic rocks, a simple proxy such as La/Sm can be a direct reflection of upper-mantle thermal conditions. We feel that prior to this study, many geochemists would argue that mantle composition and lithospheric contamination would play a greater role in controlling La/Sm within high-MgO volcanic rocks than we observe. In that sense, our results are an important step forward.
- We use both geochemical and tomographic techniques to estimate asthenospheric temperature and lithospheric thickness beneath intraplate volcanic provinces. We find an encouraging positive correlation between the two. Many geochemical studies would estimate melt fraction, temperature and lithospheric thickness beneath a single volcanic province as a final result. Here we provide estimates for > 100 global locations.
- Thinning of the lithosphere and warm asthenospheric temperatures can generate dynamic uplift of Earth’s surface. In the final section, we show that many regions with active intraplate volcanism have experienced significant uplift during Neogene and Quaternary times. Therefore, the presence of intraplate volcanism within the stratigraphic record may serve as an important indicator of both mantle and surface conditions at the time of eruption. This part of the manuscript is neither an afterthought nor an irrelevancy: the linkage to independent stratigraphic observations is a key test of our global analysis that brings crucial closure.

We think these significant results are clearly outlined throughout the manuscript, especially within the summary paragraph at the beginning of Section 4. We note that there is a correlation between potential temperature and plate thickness but this correlation is not necessarily the central result of our study. When lithospheric thicknesses are > 100 km, a correlation between potential temperature at 100–200 km depth and plate thickness is perhaps expected on fluid dynamical grounds (i.e. thermal boundary layer considerations; see also replies to Comments 18–20).

(6) “There are a number of statistical tests, but many are not too clear or convincing: (i) Kolmogorov Smirnov tests seem quite a lot of effort for illustrating that, as one would expect from a textbook understanding of adiabats and solidi, intraplate volcanism concentrates where plates are thin”

We mostly disagree. Whilst we agree with the reviewer that the association between intraplate volcanism and thin lithosphere is partly expected and straightforward to explain, we do not know of any previous studies which have shown that this association holds true on a global scale by quantitatively analyzing a comprehensive database. Therefore, we believe it is an key observation that should be included in the manuscript. We also believe that it is necessary to quantify the spatial relationship between intraplate volcanism, shear-wave velocity and lithospheric thickness. In order to quantify this relationship, a Kolmogorov-Smirnov test is the most appropriate statistical technique. Regarding other statistical tests used throughout the manuscript, we have now included population correlation coefficients and we have stated the limits of statistical significance within the main text.

(7) “(ii) what does the principal component analysis add?”

We believe that principal component analysis is an important sense check. Apart from temperature variations, it is widely recognized that mantle composition and lithospheric thickness can also cause variations in La/Sm and ΔV_s . Principal component analysis enables us to deconvolve the main trends within the trace element data and to attribute these trends to physical properties within the mantle. In this way, we use principal component analysis to show that the composition of intraplate basaltic rocks are dominated by a principal component which can be related to melt fraction. By comparing this principal component to ϵNd variations, we have carefully demonstrated that compositional variations of the mantle does not control the correlation between ΔV_s and La/Sm (Rebuttal Figure 6). We also use principal component analysis in the main text to show that lithospheric thickness changes can impact measured La/Sm values.

In revising the manuscript, we have enhanced the principal component analysis section by including synthetic results. By comparing the results of observed and synthetic principal component analyses, we show that this primary component is controlled by temperature rather than lithospheric thickness variations (Rebuttal Figure 7a,b). Finally, analysis of principal component 3 demonstrates that contamination from melting within the lithosphere does not significantly impact of the correlation between La/Sm and ΔV_s . In this way, we have underlined and strengthened the principal component analysis which we believe to be an important toolkit for bolstering our results.

(8) “(iii) how significant is the correlation between geochemical and seismic temperatures and what about the systematic offset between the two?”

The fact that geochemical and seismic temperatures are in broad agreement with each other is of great significance— it is important to bear in mind that we are bringing two very disparate sets of observations together and yet a self-consistent global story has emerged. Nevertheless, the reviewer is correct to highlight the small systematic offset, the nature of which we do not fully understand and which is a topic for future research. To highlight the significance of the correlation and to draw attention to the offset, we have now added the population correlation coefficient to Figure 5a. We have also revised the discussion of the uncertainties inherent to both the geochemical and tomographic T_p estimate within Section 3. We believe this expanded discussion covers various different possible causes of this offset although we would emphasize that we do not have a complete explanation.

(9) “Although it is suggested that sublithospheric temperatures exert a main control on the occurrence of intraplate magmatism, I did not find the case against a dominant control by lithospheric thickness convincing.”

We are rather puzzled by this criticism by Reviewer 3 but we take it very seriously since refers to a key conclusion of the manuscript. As it happens, we do not state within the manuscript that the occurrence of volcanism is controlled by sub-lithospheric temperatures. We completely agree with Reviewer 3 that a primary control on the occurrence of intraplate magmatism is likely to be lithospheric thickness, partly because intraplate magmatism is not present in regions where the lithosphere is > 100 km thick (Figure 2b in main text). This negative association is an important one. However, whilst the occurrence of magmatism may be primarily controlled by lithospheric thickness, we believe our study shows that the composition of intraplate basaltic rocks is also moderated in a significant way by sub-lithospheric temperature variations. To bolster this argument, we have now included a discussion of the effects of lithospheric thickness variations on ΔV_s , we have expanded our principal component analysis section, and we have revised our discussion of tomographic and geochemical T_p estimates. In summary, it is very likely, on fluid dynamic considerations of thermal boundary layer development, that asthenospheric temperature variation and lithospheric thickness changes are inextricably linked. For this reason, we have

emphasized the significance of marine stratigraphic observations (i.e. dynamic topography) at the end of the manuscript which provide key evidence for rapid lithospheric thickness evolution. In that sense, linked temperature and thickness are a key centerpiece of the Manuscript.

(10) “The introduction does not provide a very clear motivation for the study. The first paragraph of the introduction mentions the integration of observations and geodynamic modelling, but this is not something this paper actually does.”

We partly disagree. We believe a wider statement of the overall objective is required to set the scene at the beginning of the manuscript. Whilst we do not undertake specific geodynamic modeling here, we think that the observations we make have direct, and even profound, implications for the geodynamic modeling community. We believe the subsequent sentence provides clear motivation for our study— “It is especially important to quantify the relationship between volcanic activity and mantle convection away from complications associated with active plate boundaries in order to understand the geodynamic interactions between mantle composition, melting and temperature through geologic time”. Thus, the linkage and motivation are clear although in revising this manuscript, we have striven to bring even greater clarity.

(11) “A next paragraph mentions: Recent advances in topographic modelling enable shear wave velocities to be mapped in detail throughout the upper mantle, but most of the tomographic models used are not all that recent. Mainly the motivation is one of opportunity, seismic and geochemical estimates of mantle potential T under continents can only be combined where intraplate volcanism occurs.”

We agree. The years 2011–2016 is not that recent and we have changed the pertinent sentence from “Recent advances in tomographic modelling enable shear wave velocities to be mapped in detail throughout the upper mantle.” to “High resolution global tomographic models map shear wave velocity anomalies, ΔV_s , throughout the upper mantle.”.

(12) “The final conclusion appears to be one on dynamic topography, a topic only introduced in the last few paragraphs of the paper.”

We disagree with the implicit sentiment that dynamic topographic considerations are just afterthought. We are very conscious that the asthenospheric temperature and lithospheric thickness changes discussed in the manuscript have profound implications for the dynamic topographic evolution of the Earth’s surface through space and time. It is brought into the last part of the discussion because independent marine stratigraphic observations have a key role in testing and closure. Nevertheless, we agree that the dynamic topography section should be first presented within the introduction and we have now added a relevant sentence within Paragraph 3 of the introduction. We believe it is important within the discussion of this manuscript to outline the impact of the results within a broader context. This structure is especially true for *Nature Communications*, which has a wide readership covering many disciplines. The dynamic topography section contextualises our results and highlights their implications for geodynamicists, geomorphologists, stratigraphers and climate scientists, amongst other disciplines. We therefore consider our discussion of dynamic topography to be an important part of this manuscript.

(13) “All aspects of the methods need more explanation in the main text. The actual method section only discussed statistical methods and the REE modelling.”

We agree that a more comprehensive description of our methods are required and we have now included details about tomographic modeling and about V_s -to- T parameterization sections within the Methods. However, we have chosen to retain the majority of the methods at the back of the

manuscript in line with *Nature* guidelines

(www.nature.com/nature-research/for-authors/write#how-to-write-a-scientific-paper).

(14) “For the geochemical database, what suite of geochemical indicators is used to remove samples? Supplementary Table 3 only mentions MgO. Is the database to be released with the paper? Will the database contain additional information about how samples have been chosen for inclusion or removal?”

The original full database will indeed be released with the published paper. We have also included a comprehensive reference list within the Supplementary Materials which should help to clarify specific queries. Our screening strategy is clearly explained within the manuscript and as the reviewer points out, MgO weight percentage plays a key role in this regard. This database took considerable effort to curate but our intention is that it will be provided to the community as part of the publication procedure.

We fully agree that we have used a confusing sentence “A suite of geochemical indicators are used to identify and remove samples that are affected by subduction zone processes”. Initially, we attempted to remove samples based on geochemical composition using the same geochemical indicators on a global basis. However, it soon became clear that each subduction region exhibits very different geochemical compositions and using the same geochemical tests globally would include unsuitable samples while excluding suitable samples. We therefore decided to use the conclusions of each individual study. Each time we included data from a specific study within our database, we examined the publication and only included samples which the authors of that study deemed to be generated by intraplate processes. We have therefore amended the relevant sentence in the manuscript to “We have used a combination of review literature and local studies to identify and remove samples pertaining to subduction zone processes”. We hope that this strategy is now clearer. Note that, when all regions close to subduction zones are removed from the database, our conclusions remain robust (Supplementary Figure 4).

(15) “A discussion is missing of whether sampling bias could affect the interpretation of the distribution of intraplate volcanism. I.e., what is the uncertainty and potential bias on the cumulative areal extent that is used for later correlations?”

We agree that it is important to check this issue and have now done so. We have developed and included an additional filtering test where we vary the size of the bins used within our correlation analysis. As bin size is increased from $0.5^\circ \times 0.5^\circ$ to $5^\circ \times 5^\circ$, there is a reduction in correlation coefficient from ~ 0.7 to ~ 0.55 for the SL2013sv model (Rebuttal Figure 2e). We have included this additional figure alongside an explanation of the results it contains within the revised Supplementary Materials.

(16) “The shear velocity models are used for correlation with the distribution of intraplate volcanism without any introduction of the V_s models, their sensitivity, resolution (laterally and with depth) and limitations. There should also be some discussion of what the reference value of V_s means as much significance is attached to where anomalies are positive and negative. Some of these aspects are mentioned in the final section (on geodynamic implications) but without discussing how these might influence the results.”

We agree and we have now added a full description of SL2013sv to the Methods section. We have also stated the spatial and vertical resolution of tomographic models both in the methods and discussed the implications of these limitations within the main text. We have included a specific statement within the main text detailing the reference model used to calculate ΔV_s .

(17) The conversion to thermal structure needs more explanation than currently given, including mentioning uncertainties and possible biases.

We agree and we have therefore added an extensive V_s -to- T parameterization section to the Methods section. We have also revised Section 3 to include a discussion of the uncertainties related to this parameterization.

(18) To what extent are the seismic velocity models able to independently resolve lithospheric thickness and asthenospheric temperature? And how correlated are (continental) lithospheric thickness and asthenospheric? Very correlated if one goes by Fig. 5. This is surprising as one might have expected that under continents, asthenospheric temperature is not the only control on thickness, but that compositional structure of the lithosphere, tectonic history and distribution of heat producing elements might also have an effect.

This issue is an important one and we have taken the following steps to allay the reviewer’s concerns. To assess the degree to which lithospheric thickness and asthenospheric temperature variations can be separated using seismic data, we have included additional analysis within the main text. Thus we have added Figure 2e which shows the correlation coefficient between lithospheric thickness and ΔV_s as a function of depth for four tomographic models. At depths ≤ 100 km ΔV_s and lithospheric thickness are strongly correlated and this correlation decreases with depth. At depths of ≥ 150 km, the correlation between lithospheric thickness and ΔV_s becomes insignificant. Since a clear correlation between La/Sm and ΔV_s is observed at depths ≥ 150 km it is reasonable to assume this correlation is related to asthenospheric temperature variations. In light of this additional analysis we have changed our depth window from 100–200 km to 125–175 km. To demonstrate that these changes make a minimal difference to our overall conclusions, we have now included results for 100–200 km, 125–175 km and 150 km depths within this rebuttal (Figures 3, 4, and 5, respectively). We think that our result is both surprising and significant. We fully agree that lithospheric compositional structure, tectonic history and distribution of heat-producing elements should play a role but our correlations demonstrate that these effects are likely to be secondary compared with temperature and thickness.

In regions where the lithosphere is much greater than 100 km thick, such as beneath cratons, a negative correlation between lithospheric thickness and potential temperature at 100–200 km depths is expected. Since our study is primarily focused on regions where the lithosphere is < 100 km thick, no such correlation is expected between T_p at 150 ± 25 km and lithospheric thickness within our results (Figure 2e of main text). To make this point clearer, we have included an additional statement within our description of Figure 5— “Note that in regions where the lithosphere is > 125 km thick, a correlation between ΔT_p and lithospheric thickness at depth of 150 ± 25 km is expected”.

The lithospheric mantle beneath thick cratonic regions is thought to be depleted and shear waves will travel faster through depleted mantle than through fertile mantle (Schutt and Leshner, 2006). Since our V_s -to- T parameterization does not account for compositional variations, the increase in V_s caused by lithospheric depletion will be attributed to thermal variations. Therefore, temperatures calculated within cratonic regions will be underestimates. These compositional effects act to artificially increase the negative correlation between lithospheric thickness and potential temperature. Despite this additional compositional uncertainty, even in cratonic regions our chosen V_s -to- T parameterization closely reproduces geotherms generated using xenolith thermobarometric data (Hoggard et al., 2020). We have added a discussion of the effects of continental structure and mantle depletion to Section 3 of the main text.

Figure 3: Spatial and geochemical correlations. (a) Percentage cumulative area, ΣA , plotted as function of average value of ΔV_s from SL2013sv tomographic model between depths of 100 and 200 km where globe is sub-divided into 1° bins weighted according to $\cos \phi$ where ϕ = latitude in degrees; black curve = cumulative areal distribution of ΔV_s ; red line = cumulative areal distribution of binned intraplate volcanic samples; black/red numbered dashed lines = percentages of global surface and of intraplate volcanism with $\Delta V_s < 0 \text{ km s}^{-1}$; D = value of Kolmogorov-Smirnov statistic. Probability value of Kolmogorov-Smirnov test is $p = 7 \times 10^{-110}$ (Methods). (b) ΣA plotted as function of lithospheric thickness. Black/red lines = cumulative areal distribution of lithospheric thickness and of binned intraplate volcanic samples, respectively; black/red numbered dashed lines = percentages of global surface and of intraplate volcanism with lithosphere $< 100 \text{ km}$ thick; D as before. Probability is $p = 4 \times 10^{-162}$. (c) La/Sm plotted as function of average value of ΔV_s between depths of 100 and 200 km from SL2013sv tomographic model. Circles and error bars = average value $\pm \sigma$ for each 1° bin; red line = line that best fits values weighted according to $\cos \phi$; pair of dotted red lines = uncertainty envelope for suite of best-fit lines where database is binned using 99 different configurations spaced at 0.1° intervals; R = correlation coefficient and its uncertainty; P = population correlation coefficient; N = number and range of bins. Note that intraplate basalt database is screened to only include samples where $14.5 \geq \text{MgO wt\%} \geq 9$, $< 10 \text{ Ma}$, and number of samples per bin > 5 . (d) Value of R , calculated between La/Sm and average value of ΔV_s at center of 100 km depth range, plotted as function of depth for four different tomographic models. Solid/dash-dot/dashed/dotted lines = SL2013sv/CAM2016Vsv/SEMUCB-WM1/S40RTS tomographic models (Schaeffer and Lebedev, 2013; Ho et al., 2016; French and Romanowicz, 2014; Ritsema et al., 2011); $R = 0.28$ is minimum value of R distinguishable from zero at significance level of $= 0.001$. Red line and shaded red box = $150 \pm 50 \text{ km}$ for reference. (e) Value of R calculated between lithospheric thickness and ΔV_s .

Figure 4: **Spatial and geochemical correlations.** Same as Figure 3 but for ΔV_s averaged between 125–175 km depths.

Figure 5: **Spatial and geochemical correlations.** Same as Figure 3 but for ΔV_s at 150 km depth.

(19) “Actually it is may not be appropriate to call the temperature inferred from velocities between 100–200 km depth below continents T_p . Definitely where the continents are thicker than 100 km, these are lithospheric temperatures or an average of lithospheric and asthenospheric temperatures.”

We understand the point. Potential temperature, T_p , describes the temperature that would be achieved by a packet of rock or melt if it adiabatically decompressed to surface pressures. It is a useful tool for comparing temperatures at different pressures within the Earth. We agree with Reviewer 3 that within the continents heat is transferred via conduction and packets of rock or melt do not decompress adiabatically. However, this lack of adiabatic decompression does not detract from the use of the term T_p , since T_p is purely a measure of heat content accounting for pressure. Even within the lithosphere, the mantle is under compression and this pressure effect must be accounted for. Conductive or convective transfer of heat is not implied by the use of T_p . We agree with Reviewer 3 that in areas where the lithosphere is > 125 km the T_p estimated at 150 ± 25 km depths will be an average of lithospheric and asthenospheric potential temperatures. We have included an additional statement to our description of Figure 5 which we hope will satisfy this reviewer— “Note that in regions where the lithosphere is > 125 km thick, a correlation between ΔT_p and lithospheric thickness at depth of 150 ± 25 km is expected”.

(20) “With the choice for base of the lithosphere as the 1175 °C isotherm, even if the estimated thickness is less than 100 km, some of (transient) thermal boundary layer, which extends to temperatures above 1175 °C, will be below this depth and hence these temperatures will average into the “sublithospheric” temperature. Furthermore, the minimum vertical resolution achieved with the seismic models is generally about 50 km, adding to the uncertainties in the seismic temperature estimates. These arguments together with what appears to be a strong correlation between thickness and ΔV_s in 100–200 km depth range indicate that the velocities in this depth range may in part represent variations in lithospheric thickness.”

We fully agree with Reviewer 3 that the 1175 °C isothermal surface represents the transition between solid and convective mantle, rather than the base of the thermal boundary layer. Therefore, despite recording lithospheric thicknesses of < 100 km for $\sim 95\%$ of the database, the extension of the thermal boundary layer into the 100–200 km channel may impact upon the correlation we observe. We also agree that in Figure 1 of the main text there is a visual association between ΔV_s at 100–200 km depths and lithospheric thickness. However, this visual association is dominated by regions where the lithosphere is > 100 km thick where a correlation between ΔV_s between 100–200 km and lithospheric thickness is expected.

To address this important concern, we have included additional analysis to Section 2 of our study. We show the correlation between lithospheric thickness and ΔV_s becomes insignificant at depths ≥ 150 km (Figure 2e). To reduce the influence of lithospheric thickness variations, we have also revised the depth range of our analysis to 125–175 km. After performing this change to minimize the effect of lithospheric thickness variations there is no appreciable decrease in the correlation between La/Sm and ΔV_s and we therefore consider our conclusion to be robust (Figures 3, 4 and 5). We think that presenting these different possible averaging strategies, which demonstrate that our central result does not change, that we have been as neutral and careful as possible in this regard.

(21) “Fig . 4 gives an example, for two locations, for the potential trade-off between depth and temperature in the results from the REE data. It would help if the reader were given a more general insight in this important trade-off, and under which cir-

cumstances the two might be independently resolvable and when not (maybe with an additional figure)."

We agree that Figure 4 clearly shows the trade-off between thickness and temperature but we are unclear what kind of additional figure that the reviewer actually has in mind. The trade off is not excessive in our view. Within Figure 4, we show two geochemical modelling results, one for the Galápagos and one for Haruj. In Figures 4c and f we show the trade-off between T_p and lithospheric thickness for Galápagos and Haruj, respectively. The degree to which this trade-off can be resolved is data dependent. To allow the reader to observe the trade-off between these variables we provide a misfit figure for every single location within our global database in the Supplementary Materials and we therefore believe an additional figure is not necessary since the reader can inspect this comprehensive (if not exhaustive) set of plots themselves. However, we have now included a statement explaining that a larger trade-off between T_p and lithospheric thickness is expected for locations with larger ranges of rare earth element compositions. We think that we have been open and reasonable in presenting all of these trade-off relationships for the complete database.

(22) "It would also be good if it could be illustrated that by combining the geochemical and seismic constraints it is possible to resolve thickness- T_p trade-offs."

We already believe that our revised manuscript provides convincing evidence that ΔV_s at 150 ± 25 km depth represents variations in asthenospheric temperature rather than lithospheric thickness or composition *per se*. We also use a geochemical inversion scheme which solves for both asthenospheric temperature and lithospheric thickness. A positive correlation is observed between geochemically and tomographically derived temperature estimates at asthenospheric depths (Figure 5a of main text). We believe this correlation demonstrates that both techniques can reasonably reduce the anticipated trade-off between T_p and lithospheric thickness. For the geochemical temperature estimates, a misfit grid is provided to show to what extent these two variables can be separated.

(23) "It is good to see that the correlation between La/Sm and ΔV_s is investigated before a comparison between the temperature estimates from the two types of observables is made. But could you also provide an indication of how significant the correlation is?"

We think that we have already done so but we have revised this aspect. Within the caption of Figure 2, we state that " $R = 0.28$ is the minimum value of R distinguishable from zero at significance level of $= 0.001$ ". We feel this statement is adequate for the reader to interpret the significance of the correlation but we have also decided to update Figure 2c to include the population correlation coefficient. We have also edited the main text to explicitly refer to this significance limit.

(24) "It is nice that there is also a correlation between T_p from seismic and geochemical analyses. However, again the question is how significant is an R of 0.55?"

We fully agree that there is a correlation as the reviewer states. We have now included the population correlation coefficient and the significance limit within Figure 5a so that readers can interpret the significance of the R value. We have also revised the Figure caption to include the value of R that is distinguishable from zero at a significance level of 0.001.

(25) "And in addition, most of the seismic T_p estimates are higher than the geochemical estimates. This warrants discussion."

We agree with the reviewer and we have expanded the discussion of this trade-off within Section 3.

(26) “If however as argued above the variations in dVs in the 100–200 km depth range is, at least in part, an expression of variable lithospheric thickness than the correlation between La/Sm and dVs is an argument for a strong control by lithospheric thickness rather than by sublithospheric temperatures on intraplate volcanism. This again relates to the question of how independently resolvable the two effects are by La/Sm trends.”

As explained in our reply to Comment 20, we believe that by changing to a depth slice of 150 ± 25 km ΔV_s variations are not actually susceptible to variations in lithospheric thickness (Figure 2e of the main text). The degree to which La/Sm is controlled by lithospheric thickness variations is investigated within our manuscript using principal component analysis as previously described and justified. P_1 accounts for $79_{-0}^{+1}\%$ of the variance and is weighted equally for La and Sm (Figure 3a in main text). P_1 correlates with both La/Sm and V_s and can be thought of as a melt-fraction signal (see Rebuttal Figure 6a,d).

As Reviewer 2 and Reviewer 3 correctly identify, melt fraction and La/Sm can vary as a function of both asthenospheric T_p and lithospheric thickness. However, we can separate these signals using principal component analysis by incorporating additional elements, such as Yb, which are sensitive to depth of melting. Spinel-bearing peridotite converts into garnet-bearing peridotite as depths increase below ~ 60 – 80 km (Klemme and O’Neill, 2000; Jennings and Holland, 2015). Since Yb is far more compatible in garnet than spinel, Yb concentrations within a melt are more sensitive to the depth at which melting occurs than to melt fraction variations. P_2 accounts for 14% of the variance and is dominated by Yb variations (Figure 3b in main text). P_2 can therefore be considered as a proxy for the relative contributions of spinel- and garnet-bearing peridotite. If asthenospheric T_p is kept constant and lithospheric thickness increases, an increase in La/Sm and a decrease in Yb concentrations is expected. Alternatively, if lithospheric thickness is kept constant and asthenospheric temperature increases then more melting will occur within the garnet region. Since P_2 does not correlate with La/Sm or ΔV_s , we can conclude that both T_p variations and lithospheric thickness variations affect La/Sm values within our dataset (see Rebuttal Figure 6b,e).

We can investigate the relationship between these principal components, T_p and lithospheric thickness by analysing synthetic examples. Here, we calculate principal components using elemental concentrations generated using the INVMEL model at a range of T_p s, lithospheric thicknesses and mantle compositions. Random distributions of T_p , lithospheric thickness and ϵNd are generated using Gaussian distributions defined by means and standard deviations of 1350 °C and 40 °C, 55 km and 8 km, and 5 and 2 , respectively. Outer limits for T_p , lithospheric thickness and ϵNd are set at 1250 – 1450 °C, 35 – 75 km, and 0 – 10 , respectively. Finally, resultant T_p , lithospheric thickness and ϵNd values are rounded to the nearest 5 °C, 1 km and 0.25 , respectively. In each case 150 values are generated, and these values are combined to describe a series of input parameters for 150 INVMEL models. The concentrations of Ba, Nb, K, La, Nd, Zr, Sm and Yb calculated within these model runs are then analyzed using principal component analysis and this procedure is repeated 99 times.

The first principal component calculated from the synthetic modeling, M_1 , accounts for $79_{-7}^{+7}\%$ of the variance and has a similar weighting and variance to P_1 (Figure 3d in main text). M_1 strongly correlates with T_p and does not significantly correlate with lithospheric thickness or mantle composition (Rebuttal Figure 7a,d,g). We believe it is therefore reasonable to conclude that both M_1 and P_1 are primarily sensitive to changes in T_p . M_2 accounts for $8_{-4}^{+4}\%$ of the variance, and like P_2 it is dominated by variations in Yb (Figure 3e in main text). M_2 correlates with lithospheric thickness but does not correlate with either T_p or mantle composition (Rebuttal Figure 7b,e,h). It is therefore possible that both M_2 and P_2 are primarily sensitive to changes in lithospheric thickness.

Figure 6: Principal component analysis as function of geologic observations. (a) La/Sm as a function of Principal Component 1 (P_1), where correlation coefficient, R , is displayed at bottom right-hand side. (b) La/Sm as function of P_2 . (c) La/Sm as function of P_3 . (d)–(f) ΔV_s as function of principal components. (g)–(i) ϵNd as function of principal components.

Figure 7: Principal component analysis as function of model parameters. (a) T_p as function of Principal Component 1 (M_1), where correlation coefficient, R , is displayed at bottom right-hand side. Red line = line of best fit. (b) T_p as function of M_2 . (c) T_p as function of M_3 . (d)–(f) Lithospheric thickness as function of principal components. (g)–(i) ϵNd as function of principal components.

Figure 8: Principal component analysis as function of temperature and lithospheric thickness estimates. (a) T_p estimated using geochemical inversion scheme as function of Principal Component 1 (P_1), where correlation coefficient/population correlation coefficient, R/P , are displayed at top right-hand side. (b) Lithospheric thickness estimated using geochemical inversion scheme as function of P_1 . (c) T_p estimated using V_s -to- T parameterization as function of Principal Component 1 (P_1). (d) Lithospheric thickness estimated using V_s -to- T parameterization as function of P_1 . (e–h) Same for P_2 .

A second test is to compare principal components derived from our geochemical database to T_p and lithospheric thickness estimates from geochemical and tomographic modeling. As expected, a correlation between P_1 and both geochemical and tomographic T_p estimates is observed (Figure 8a,c). A correlation is also observed between P_1 and geochemical lithospheric thickness estimates (Figure 8b). However, the majority of lithospheric thickness estimates are within a 10 km range, close to the uncertainty on any individual sample. It is therefore difficult to assess the significance of this correlation. P_2 has low variance and does not correlate with any estimates of asthenospheric temperature or lithospheric thickness.

We have now expanded both the principal component analysis section to fully address the degree to which the trade-off between T_p and lithospheric thickness can be resolved. We have also added a discussion of the degree to which vertical smearing may affect ΔV_s estimated between 100–200 km depths.

(27) “The conclusion from the work appears to be that both thin lithosphere and elevated asthenospheric temperatures contribute to intraplate volcanism. However, the distribution of asthenospheric temperatures is not really discussed. As queried above, are the seismic estimates actually estimates of T_p ? A strong correlation between T_p and lithospheric thickness is not necessarily expected. Hotter asthenosphere could thin the lithosphere, but thinner lithosphere also allows melting to occur for cooler

mantle temperatures.”

We fully agree that both thin lithosphere (or lithosphere that has become thin) and elevated asthenospheric temperature play a significant role in generating intraplate volcanism. Our additional analysis within Section 2 demonstrates that seismic velocities at 150 ± 25 km depth, and therefore the temperatures derived from these velocities, do represent asthenospheric mantle. We agree that hotter asthenosphere and thinner lithosphere appear to be linked which is what would be expected from a fluid dynamical perspective. We are unaware, from mantle convective modeling, of situations where a lithospheric plate could become thin when the asthenosphere remained cool. We believe that the onus in this case lies with those who favor such a model.

(28) “The discussion of dynamic topography comes as a bit of a diversion. The rest of the paper had not mentioned dynamic topography, and the distribution of temperature and thickness obtained was not thoroughly discussed before being used in this next step of interpretation.”

We completely disagree that dynamic topography is a diversion. On the contrary, we believe that observed dynamic topography plays a significant role in testing the thesis that we have elaborated—regions with thin lithosphere and hotter asthenosphere often have elevated marine strata which demonstrates that regional uplift has occurred. Thus we firmly believe the dynamic topography section is necessary both to convey the importance of our results to a wider audience and to ‘close the loop’ by testing our global hypothesis. We have now revised the manuscript to ensure that the dynamic topography section is flagged in the introduction. We have also added several sentences at the start of the dynamic topography section which improve the logical flow of our geodynamic implications section. We agree with the reviewer that the temperatures and thicknesses were not adequately discussed prior to this next step. We have now restructured the manuscript to include a thorough discussion of our temperature estimates within Section 3.

(29) “The abstract lists as a main aim to link intraplate volcanism to mantle flow. However, the analyses do not constrain mantle flow, only temperature. Lateral variations in temperature, which may relate to flow are not systematically discussed.”

We only partly agree. We agree with the reviewer that the abstract should focus on thermal variations as such and we have rewritten the abstract accordingly. However, we disagree with the statement that lateral variations in temperature are not systematically discussed since the focus of this study is to determine whether intraplate volcanism can be used as a proxy for upper-mantle temperature variations. Regarding mantle flow, we think that our global results have critical implications for the ability to satisfactorily identify and map flow. A major difficulty is that observable dynamic topography appear to be dominantly controlled and moderated by a combination of asthenospheric temperature variations and lithospheric thickness changes. This dominance places a severe test upon robust identification of mantle flow, whose reality we do not dispute. It is important to bear in mind that mantle flow is the prediction of theoretical geodynamical models but it is fiendishly difficult to actually observe.

(30) “While V_s is sensitive to pressure and temperature variations, it also depends on anelasticity”. This is poorly worded. The variations of V_s with anelasticity contribute to the sensitivity of V_s to temperature and pressure (and possibly also composition, melts and other fluids).”

We agree with the reviewer that this sentence was poorly constructed. We have now changed “While V_s is sensitive to pressure and temperature variations, it also depends on mantle composition, anelasticity, and the presence of melt.” to “Although it is agreed that shear waves propagate

slower through warm mantle and faster through cold mantle, V_s varies non-linearly with pressure and temperature (Priestley and McKenzie, 2006; Yamauchi and Takei, 2016). It is also sensitive to mantle composition and the presence of melt (Schutt and Leshner, 2006; Hirschmann, 2010).”

(31) “Figure 1a demonstrates that there is a striking visual correlation between intraplate volcanism and average shear wave velocity anomalies This sentence needs clarification. As written it implies that intraplate volcanism occurs where there are Vs anomalies. But Vs anomalies cover most of the globe, while intraplate volcanisms does not.”

We agree with the reviewer and have changed “correlation” to “association” which is clearer with regard to our intent.

(32) “Fig. 1 has quite binary colour scales that draw the attention to a specific contour that correlates reasonably with the boundary between areas with and without continental intraplate volcanism.”

In Figure 1a of the main text, we do indeed use a binary colour scale which highlights the difference between areas with positive and negative V_s anomalies. Areas of faster and slower than average velocities as blue or red, respectively, and the colour saturation scales as a function of velocity difference from zero. We consider this scheme to be a very reasonable and frequently employed choice when displaying tomographic models.

In Figure 1b we use a continuous colour scale for lithospheric thickness which includes a saturated white color for regions with a lithospheric thickness of < 100 km. Uncertainties related to lateral crustal thickness and density variations can cause artificial “bleeding” of slow crustal velocities into the upper-most mantle. These regions of artificially slow velocities are confined to < 100 km depths but can lead to underestimates in lithospheric thickness (Priestley and McKenzie, 2013). We therefore saturate our color scale at 100 km since lithospheric thickness estimates < 100 km carry significant uncertainties.

(33) “Fig. 2d could use a legend”

We agree with the reviewer and have added a legend to Figure 2.

(34) “Why are the reference potential temperatures for seismic and geochemical estimates not consistent? This needs some motivation in the main text. Also these reference values need to be introduced before it is argued that in the two examples in Fig. 4 the T_p inferred is elevated.”

We agree with Reviewer 3 that the reference temperatures should be introduced before we state that temperatures beneath Haruj and Galápagos are elevated. We have now included an explanation of the geochemical reference temperature at an appropriate point within the main text. We have also offered a brief explanation as to the difference between the reference temperatures when the seismological reference temperature is introduced later in the main text.

Additional Changes

(1) Additional Data

We have added > 1000 new data from submerged seamounts and Tibet which were missed in the original submission. Results, figures and supplementary tables have been updated according to the new database.

References

- D. F. Argus, R. G. Gordon, and C. DeMets. Geologically current motion of 56 plates relative to the no-net-rotation reference frame. *Geochemistry, Geophysics, Geosystems*, 12(11), 2011.
- P. Bird. Continental delamination and the Colorado Plateau. *Journal of Geophysical Research: Solid Earth*, 84(B13):7561–7571, 1979.
- P. Bird. An updated digital model of plate boundaries. *Geochemistry, Geophysics, Geosystems*, 4(3), 2003.
- J. S. Byrnes, E. E. Hooft, D. R. Toomey, D. R. Villagómez, D. J. Geist, and S. C. Solomon. An upper mantle seismic discontinuity beneath the galápagos archipelago and its implications for studies of the lithosphere-asthenosphere boundary. *Geochemistry, Geophysics, Geosystems*, 16(4):1070–1088, 2015.
- C. A. Dalton, C. H. Langmuir, and A. Gale. Geophysical and geochemical evidence for deep temperature variations beneath mid-ocean ridges. *Science*, 344(6179):80–83, 2014. doi: 10.1126/science.1249466.
- S. W. French and B. A. Romanowicz. Whole-mantle radially anisotropic shear velocity structure from spectral-element waveform tomography. *Geophysical Journal International*, 199(3):1303–1327, 2014. ISSN 1365246X. doi: 10.1093/gji/ggu334.
- O. H. Göğüş, R. N. Psyklywec, A. M. C. Şengör, and E. Gün. Drip tectonics and the enigmatic uplift of the Central Anatolian Plateau. *Nature Communications*, 8(1538), 2017. doi: 10.1038/s41467-017-01611-3.
- M. M. Hirschmann. Partial melt in the oceanic low velocity zone. *Physics of the Earth and Planetary Interiors*, 179:60–71, 2010. doi: 10.1016/j.pepi.2009.12.003.
- T. Ho, K. Priestley, and E. Debayle. A global horizontal shear velocity model of the upper mantle from multimode Love wave measurements. *Geophysical journal international*, 207(1):542–561, 2016.
- M. J. Hoggard, K. Czarnota, F. D. Richards, D. L. Huston, A. L. Jaques, and S. Ghelichkhan. Global distribution of sediment-hosted metals controlled by craton edge stability. *Nature Geoscience*, 13(7):504–510, 2020.
- X.-L. Huang, Y. Niu, Y.-G. Xu, L.-L. Chen, and Q.-J. Yang. Mineralogical and geochemical constraints on the petrogenesis of post-collisional potassic and ultrapotassic rocks from western Yunnan, SW China. *Journal of Petrology*, 51(8):1617–1654, 2010.
- E. S. Jennings and T. J. B. Holland. A simple thermodynamic model for melting of peridotite in the system NCFMASOCr. *Journal of Petrology*, 56(5):869–892, 2015. ISSN 14602415. doi: 10.1093/petrology/egv020.
- R. F. Katz, M. Spiegelmann, and C. H. Langmuir. A new parameterization of hydrous mantle melting. *Geochemistry, Geophysics, Geosystems*, 4(9), 2003. doi: 10.1029/2002GC000433.
- S. Klemme and H. S. C. O’Neill. The near-solidus transition from garnet lherzolite to spinel lherzolite. *Contributions to Mineralogy and Petrology*, 138(3):237–248, 2000. ISSN 0010-7999. doi: 10.1007/s004100050560.

- C.-T. A. Lee. Compositional variation of density and seismic velocities in natural peridotites at stp conditions: Implications for seismic imaging of compositional heterogeneities in the upper mantle. *Journal of Geophysical Research: Solid Earth*, 108(B9), 2003.
- C. McCarthy, Y. Takei, and T. Hiraga. Experimental study of attenuation and dispersion over a broad frequency range: 2. The universal scaling of polycrystalline materials. *Journal of Geophysical Research: Solid Earth*, 116(B9), 2011.
- D. McKenzie. Some remarks on the development of sedimentary basins. *Earth and Planetary science letters*, 40(1):25–32, 1978.
- D. McKenzie. Constraints on melt generation and transport from u-series activity ratios. *Chemical Geology*, 162(2):81–94, 2000.
- D. McKenzie and M. J. Bickle. The volume and composition of melt generated by extension of the lithosphere. *Journal of Petrology*, 29(3):625–679, 1988.
- D. McKenzie and R. K. O’Nions. The Source Regions of Ocean Island Basalts. *Journal of Petrology*, 36(1):133–159, 1995.
- D. McKenzie, J. Jackson, and K. Priestley. Thermal structure of oceanic and continental lithosphere. *Earth and Planetary Science Letters*, 233(3-4):337–349, 2005. ISSN 0012821X. doi: 10.1016/j.epsl.2005.02.005.
- Y. Niu. The meaning of global ocean ridge basalt major element compositions. *Journal of Petrology*, 57(11-12):2081–2103, 2016.
- Y. Niu, M. Wilson, E. R. Humphreys, and M. J. O’Hara. The origin of intra-plate ocean island basalts (OIB): The lid effect and its geodynamic implications. *Journal of Petrology*, 52(7-8): 1443–1468, 2011. ISSN 00223530. doi: 10.1093/petrology/egr030.
- B. Parsons and J. G. Sclater. An analysis of the variation of ocean floor bathymetry and heat flow with age. *Journal of geophysical research*, 82(5):803–827, 1977.
- S. Pilet, M. B. Baker, O. Müntener, and E. M. Stolper. Monte Carlo simulations of metasomatic enrichment in the lithosphere and implications for the source of alkaline basalts. *Journal of Petrology*, 52(7-8):1415–1442, 2011. ISSN 00223530. doi: 10.1093/petrology/egr007.
- K. Priestley and D. McKenzie. The thermal structure of the lithosphere from shear wave velocities. *Earth and Planetary Science Letters*, 244(1-2):285–301, 2006. ISSN 0012821X. doi: 10.1016/j.epsl.2006.01.008.
- K. Priestley and D. McKenzie. The relationship between shear wave velocity, temperature, attenuation and viscosity in the shallow part of the mantle. *Earth and Planetary Science Letters*, 381: 78–91, 2013. doi: 10.1016/j.epsl.2013.08.022.
- N. Rawlinson, D. Davies, and S. Pilia. The mechanisms underpinning Cenozoic intraplate volcanism in eastern Australia: Insights from seismic tomography and geodynamic modeling. *Geophysical Research Letters*, 44(19):9681–9690, 2017.
- F. D. Richards, M. J. Hoggard, L. Cowton, and N. J. White. Reassessing the Thermal Structure of Oceanic Lithosphere With Revised Global Inventories of Basement Depths and Heat Flow Measurements. *Journal of Geophysical Research: Solid Earth*, 123:9136–9161, 2018.

- F. D. Richards, M. J. Hoggard, N. White, and S. Ghelichkhan. Quantifying the relationship between short-wavelength dynamic topography and thermomechanical structure of the upper mantle using calibrated parameterization of anelasticity. *Journal of Geophysical Research: Solid Earth*, page e2019JB019062, 2020.
- J. Ritsema, a. A. Deuss, H. Van Heijst, and J. Woodhouse. S40rts: a degree-40 shear-velocity model for the mantle from new rayleigh wave dispersion, teleseismic travelttime and normal-mode splitting function measurements. *Geophysical Journal International*, 184(3):1223–1236, 2011.
- A. J. Schaeffer and S. Lebedev. Global shear speed structure of the upper mantle and transition zone. *Geophysical Journal International*, 194(1):417–449, 2013. doi: 10.1093/gji/ggt095.
- D. Schutt and C. Leshner. Effects of melt depletion on the density and seismic velocity of garnet and spinel lherzolite. *Journal of Geophysical Research: Solid Earth*, 111(B5), 2006.
- A. Späth, A. P. Le Roex, and N. Opiyo-Akech. plume-lithosphere interaction and the origin of continental rift-related alkaline volcanism - the Chyulu hills volcanic province, southern Kenya. *Journal of Petrology*, 42(4):765–787, 2001.
- L. Stixrude and C. Lithgow-Bertelloni. Mineralogy and elasticity of the oceanic upper mantle: Origin of the low-velocity zone. *Journal of Geophysical Research: Solid Earth*, 110(B3), 2005.
- Y. Takei. Effects of Partial Melting on Seismic Velocity and Attenuation: A New Insight from Experiments. *Annual Review of Earth and Planetary Sciences*, 45(1): 447–470, 2017. ISSN 0084-6597. doi: 10.1146/annurev-earth-063016-015820. URL <http://www.annualreviews.org/doi/10.1146/annurev-earth-063016-015820>.
- F. Volker, R. Altherr, K. P. Jochum, and M. T. McCulloch. Quaternary volcanic activity of the southern Red Sea: New data and assessment of models on magma sources and Afar plume-lithosphere interaction. *Tectonophysics*, 278(1-4):15–29, 1997. ISSN 00401951. doi: 10.1016/S0040-1951(97)00092-9.
- H. Yamauchi and Y. Takei. Polycrystal anelasticity at near-solidus temperatures. *Journal of Geophysical Research: Solid Earth*, 121(11):7790–7820, 2016. ISSN 21699356. doi: 10.1002/2016JB013316.

REVIEWER COMMENTS

Reviewer #1 (Remarks to the Author):

I now understand that there is a mutual misunderstanding about fundamental concepts. I have always distinguished the asthenosphere from mantle plume conduits. In my view, they are separate entities with independent kinematic and, in some cases, geochemical properties. In contrast Ball & others include mantle plume conduits as part of the asthenosphere, and that intraplate volcanism coincides with low seismic velocities in the upper mantle and encompasses low-velocity regions generated by mantle plumes. And I have always understood that some intraplate volcanics can have nothing to do with mantle plumes.

The problem arises with the statement at the beginning of Section 4: "Our global analysis of volcanic provinces and shear-wave velocity anomalies suggests that a combination of sub-plate asthenospheric temperature anomalies and thin lithosphere controls the spatial distribution of intraplate magmatism." How is the average reader to interpret this statement? The use of the word "control" makes it a cause and effect statement. So, please tell me again how does an asthenospheric temperature anomaly control the location of Hawaii, for example? In the mantle plume model, its location at the surface is controlled by the Pacific LLSVP; the temperature anomaly in the asthenosphere is just a thermal expression. But not everyone thinks that Hawaii formed from a mantle plume anchored to the Pacific LLSVP. For example, Don Anderson and acolytes attribute it to localized hot ambient mantle, an interpretation that Ball and others would reasonably ascribe to intraplate volcanics at Eastern China, Mongolia etc. So, I think it is worthwhile adding a few words of clarification about the origin of these temperature anomalies.

It makes no difference to me that this paper go forward in its present version. However, I know I represent a readership who will find it confusing in places. If I were Ball, White, Maclennan, or Stephenson I would spend one hour of my time by adding 2 or 3 sentences of clarification.

Reviewer #2 (Remarks to the Author):

I have read the revised manuscript by Ball et al and their response to my comments and to the comments by other previous reviewers. I remain very much enthusiastic about this research for its publication in Nature Communication.

I am satisfied with authors' response to my queries except for the following:

(1) I suggested the clarification "Why should there be an inverse correlation between mantle T_p and lithosphere thickness? There is no reason." (page 13) The authors partly disagree and cited the paragraphs of the definition by McKenzie & Bickle (1988). They seem eager to answer without

understanding my question for clarification? Let's give a simple example: Assuming mantle T_p is the same everywhere except for hot thermal mantle plumes, then the mantle T_p beneath ocean ridges (essentially zero lithosphere thickness) must be the same as beneath continental cratons (> 200 km lithosphere thickness). Hence, there is no reason mantle T_p correlates with lithosphere thickness. The correct situation is this: the conductive geotherm is steeper beneath ocean ridges than beneath cratons.

(2) I suggested the clarification "Removal or thinning of lithospheric mantle produces regional uplift - No! Basal removal or lithosphere thinning will NOT cause uplift, but if anything, will lead to subsidence due to Airy isostasy (Niu, 2019)." (page 16). The authors completely disagree (see my paper <http://community.dur.ac.uk/yaoling.niu/MyReprints-pdf/2019Niu-CSB-India-Asia.pdf>). An global observation based and fully quantitative paper is in preparation, but the geological and physical concepts are given in this paper in simple clarity. The original advocates (Graig Houseman, Peter Molnar and Philip England have read this short paper). If the authors ask Dan McKenzie now, he may give the same answer as mine although he was one of the original advocates (Houseman GA, McKenzie DP, Molnar P. Convective instability of a thickened boundary layer and its relevance for the thermal evolution of continental convergent belts. *J Geophys Res* 1981;86:6115–32.)

In brief, this will be an important paper and I look forward to seeing its publication. From other reviewers, I can see there are issues to be debated, but I think it is scientifically constructive for its publication and open discussion.

Please help me upload this short comment onto the system. Thank you, and

Kind regards,

Yaoling Niu

Reviewer #3 (Remarks to the Author):

The authors did a substantial revision which in terms of content appears to address many of my and the other reviewers' comments. I appreciate the amount of effort that went into this, and the careful responses given to all comments. However, the writing remains extremely poor, and certainly does not follow the guidelines that the authors refer to in their response to my review. I would request that the authors first fix the writing properly. Then I would be happy to review the resubmitted paper for content.

At a first glance, the writing may look good, but when one tries to understand what is actually said, the writing turns out to be very imprecise and confused. I give a few examples of poor writing, starting with the abstract. In the abstract every single sentence has a problem. The first paragraphs of the introduction suffer from the same kind of imprecision and incorrect wording. I marked a few other locations on the first two pages of the manuscript (attached to the review), but as a reviewer, I cannot give this level of feedback on the rest of the manuscript. Furthermore, I find it so distracting that it is hard to focus on the content. I strongly urge the senior authors to give the text a proper

critical reading and substantial edit, to ensure that all the writing makes sense, before resubmitting.

(i) The abstract is still worded very poorly and is difficult to follow for both experts and non-experts. In detail:

- a. The paper does not establish how intraplate volcanism connects to mantle convection, as one might expect from reading the first two sentences. Rather it establishes what the relationship between intraplate volcanism and plate-mantle thermo-chemical structure is.
- b. The data base is not one of intraplate volcanism, but of the products of this volcanism.
- c. The sentence “the distribution of intraplate volcanism is associated with thin lithosphere” is oddly worded. What you mean to say is that intraplate volcanism occurs where the lithosphere is thin.
- d. The next sentence is impossible to understand: “There is a positive correlation between velocities at 150 ± 25 km depth and elemental ratios sensitive to the depth and extent of melting: regions with large melt fractions match anomalously slow velocities”. The first part of the sentence reads as if higher velocities correlate with larger depths and higher extents of melting, but then the second part says the opposite, i.e. that low velocities correlate with high melt fractions. Furthermore, in your response to R2, you emphasise that you do not interpret lower velocities as being due to higher extents of melting, as the second part of the sentence implies. The reader is left wondering what the authors are trying to say here
- e. The sentence that follows “Modelling of rare earth ...tomographic models” is similarly hard to understand. Melt models indicate that isentropic melting occurs at variable temperatures? What is this supposed to convey, in particular to non-expert readers? “Globally calibrated tomographic models” is a very obscure description (calibrated against what?). Is this even useful to say in the abstract?.
- f. The sentence “intraplate volcanism is often prevalent where elevated, but undeformed, marine sedimentary rocks show that Cenozoic regional uplift occurred” leaves it to the reader to decide what the potential implication of this correlation is.
- g. The final sentence “We conclude that the distribution and composition of intraplate magmatism through geologic time represent a uniquely powerful window into past mantle conditions “ is a generic rather than specific conclusion.

(ii) To illustrate better what I mean, this could be an alternative formulation of the first part of the abstract: “Thermal structure of the plates and mantle exerts a primary control on the evolution of surface topography and the distribution of volcanism. Only indirect observations of the distribution of temperature and its change through time are available. Here we compile a global data base of geochemical parameters for Neogene-Quaternary rocks produced by volcanism away from plate boundaries. When analysed in conjunction with tomographically-imaged seismic structure, this data provides new insights in the relation between magmatism, thermal structure and surface uplift. We find that intraplate volcanism occurs where the lithosphere is thin (< 70 km) and mantle shear wave velocities between 125 and 175 km depth are relatively slow, indicating that the mantle is warm compared to the global average temperature.....”. I find it hard to suggest how to reword the rest of the abstract as the meaning is too unclear.

(iii) First sentence of the paper “Thermal and chemical heterogeneity of the Earth’s mantle generates and maintains vigorous convection” is poorly worded. Thermal structure is largely the consequence and hence an expression of mantle convection, not the driver of mantle convection.

Similarly, it is likely that the distribution of compositional heterogeneity is largely a consequence and not a driving force of convection. There is no need to enter into a debate about this in the first sentence. You could write instead "Mantle convection redistributes heat, cycles chemical species, drives plate tectonics, and generates dynamic topography." This confusion over cause and consequence affects also the rest of this paragraph which keeps discussing mantle convection as if it is caused by thermal structure. What you probably mean to say in this paragraph is that a better mapping of thermal and chemical structure of the lithosphere and mantle can provide insight in the dynamics of mantle convection and plate tectonics. However, this is not what the paragraph says.

(iv) Third sentence of the paper "It is especially important to quantify the relationship between volcanic activity and mantle convection away from complications associated with active plate boundaries in order to understand the geodynamic interactions between mantle composition, melting and temperature through geologic time." This is a long and convoluted sentence. Understanding that intraplate volcanism occurs where plates are thin does not by itself help understand plate-mantle dynamics. I would say that volcanic activity away from the main plate boundaries provides an expression of the thermal and chemical state of the plates and mantle and as such can provide important constraints that complement those provided by geophysical observations.

(v) Line 35-36 "global tomographic models enable shear-wave velocity anomalies to be mapped...". No, tomographic models are images (maps) of shear-wave velocities. You could say "Global seismic tomography enables mapping of shear-velocity anomalies...". Better might be "Global data sets of surface wave and body-wave travel times and/or waveforms allow mapping of shear-wave velocities..."

(vi) Line 40 "A combined geochemical and geophysical analysis of the distribution and composition of igneous rocks...". Again, this is poorly worded. It sounds like you are doing a combined geochemical and geophysical analysis of the igneous rocks and a combined geochemical and geophysical analysis of shear-wave velocity anomalies. You could write "A combined analysis of the distribution and composition of igneous rocks and shear-wave velocity anomalies."

(vii) please refer to the annotated manuscript for some more examples of problems with the writing

Global Influence of Mantle Temperature and Plate Thickness on Intraplate Volcanism

P. W. Ball, N. J. White, J. Maclennan and S. N. Stephenson

Response to Reviewers' Comments

Reviewer 1

Reviewer 1 states that “it makes no difference to me that this paper go forward in its present version” but that they would like us to “spend one hour of (our) time by adding two or three sentences of clarification”. We interpret these brief statements as a positive endorsement of our revised manuscript. Reviewer 1 does not have any specific comments to make about the way in which we dealt with their original review and we also interpret this lack of further comment as an implicit positive endorsement of how we dealt with their original criticisms and comments.

Reviewer 1 does acknowledge that criticisms of our manuscript within their previous review stemmed from “a mutual misunderstanding about fundamental concepts” between the authors and the reviewer. In fact, we do not think there is a significant mutual understanding about fundamental concepts and, in the rebuttal laid out below, we carefully explain our rationale in this regard. Nevertheless, we wish to ensure that our published manuscript is as clear as possible to the wider readership and so we have taken Reviewer 1’s views very seriously during the revision process.

Reviewer 1 states that they “represents a readership who will find it [the manuscript] confusing in places”. In our view, the misunderstanding does not appear to be rooted in the scientific results *per se* but in the presentation thereof. This misunderstanding is also centered upon the precise fluid dynamical meaning and definition of the term *asthenosphere*. We are very grateful to Reviewer 1 for providing a full description of how they define the term *asthenosphere*. We address this definition of terms and the wider fluid dynamical and convective ramifications below. We do, however, point out that our manuscript is not actually concerned with the much broader issue of the planform and evolution of mantle convective processes: our aims are more modest and, in that sense, more tractable. We simply wish to explore the quantitative relationship between intraplate volcanism, tomographic models and thermal structure in a strictly observationally based way. Finally, it is important to note that Reviewer 1 states they are happy for “this paper go forward in its present version” but that they have also requested “2 or 3 sentences of clarification”. Here, we address the putative mutual misunderstanding and we clarify our use of the term *asthenosphere*, both in this rebuttal and within the revised manuscript itself.

Response to specific comments

(1) “In my view, they [the asthenosphere and mantle plumes] are separate entities with independent kinematic and, in some cases, geochemical properties. In contrast Ball & others include mantle plume conduits as part of the asthenosphere, and that intraplate volcanism coincides with low seismic velocities in the upper mantle and encompasses low-velocity regions generated by mantle plumes.”

We are confused by this statement since it does not appear to be in accordance with the general, although arguably partial, fluid dynamical understanding of mantle convection at high Rayleigh number for temperature-dependent viscosity. This understanding, which is gleaned from standard textbooks and classic publications on the topic, is that the lithospheric plate represents the upper

thermal boundary layer of the mantle convective system. Immediately beneath the lithospheric plate, there is the well-known a Low Velocity Zone (LVZ) which, based upon thermal arguments, is the asthenospheric layer of low viscosity. Thus the asthenosphere is a mechanically weak layer beneath the lithosphere (Barrell, 1914). Since the asthenosphere is defined by its rheology, this term is sometimes used to describe all mechanically weak material between the base of the lithosphere and the olivine-wadsleyite transition zone at ~ 410 km depth, which includes mantle plumes (Karato, 2012).

A more generally accepted definition of the asthenosphere is that it is a relatively thin layer of variable thickness immediately beneath the lithosphere. Modern surface tomographic models, such as those discussed in this manuscript, demonstrate that the asthenosphere is 100–200 km thick and it is characterized by velocity anomalies that range in horizontal scale from 100s to 1000s of kilometers. It is widely recognized that asthenospheric velocity anomalies have a largely thermal origin. From an observational perspective, these thermal asthenospheric anomalies play a significant role in generating and maintaining anomalous bathymetric and topographic swells (Crough, 1978). Thus the Icelandic and Hawaiian bathymetric swells are underlain by asthenospheric thermal anomalies that are confined to depths of 100–200 km beneath the surface. These asthenospheric anomalies are undoubtedly generated by mantle convection.

The ongoing debate (and therefore any possible mutual misunderstanding) concerns whether these asthenospheric anomalies are isolated (i.e. rootless) blobs of hot material or whether they are connected by conduits to either the 670 km phase transition between the upper and lower mantle or to the core-mantle boundary. The simple answer to this debate is that we do not know and in this manuscript we are completely and deliberately neutral on this as yet unresolved issue. We simply do not know whether asthenospheric anomalies always have mantle conduits or not. At present, the resolving power of seismic tomographic models is such that the resultant imagery is ambiguous to the neutral observer and thus open to different possible interpretations. It is very important to emphasize that the observations and results presented in our manuscript do not depend upon whether or not asthenospheric anomalies are isolated convective blobs or the heads of plumes that are fed by deeper-seated conduits. Notwithstanding this important and ongoing debate, it is generally accepted that thermal anomalies within the asthenospheric channel are indeed an integral component of mantle convection— detailed observational constraints from the Icelandic plume makes that very clear.

In summary, we do not think that there is any “mutual misunderstanding” between our views and those of Reviewer 1. We fully agree that there is a wider debate about the detailed nature of mantle convection and it is correct to state that seismic tomographic models play a significant role in illuminating the problem. Fortunately, our central thesis and conclusions do not specifically depend upon the outcome of this debate. For clarification, we have added a sentence at the start of the manuscript which states that the upper mantle is subdivided into a mechanically strong lithosphere and a mechanically weak asthenosphere which is 100–200 km thick.

(2) “The problem arises with the statement at the beginning of Section 4: “Our global analysis of volcanic provinces and shear-wave velocity anomalies suggests that a combination of sub-plate asthenospheric temperature anomalies and thin lithosphere controls the spatial distribution of intraplate magmatism.” How is the average reader to interpret this statement? The use of the word “control” makes it a cause and effect statement. So, please tell me again how does an asthenospheric temperature anomaly control the location of Hawaii, for example? In the mantle plume model, its location at the surface is controlled by the Pacific LLSVP; the temperature anomaly in the asthenosphere is just a thermal expression.

We agree that the word “control” is a somewhat loaded one that implies cause and effect. This implication was not our intention and so we have replaced this word and softened the relevant statement to say “...helps to determine the spatial distribution of intraplate magmatism. This inference...” We also agree that the asthenospheric thermal anomaly is merely the near-surface expression of a convective phenomenon that may well be rooted at depth and controlled by, say, an LLSVP. In this study, we are interested in surface manifestations of mantle processes, specifically intraplate volcanic activity and dynamic topography. Therefore, as one example, Hawai’i can either be thought of as a series of volcanic islands or as a long-wavelength topographic swell. We conclude that the location of both the Hawai’ian volcanic islands and the Hawai’ian swell are in that sense “controlled” by temperature variations in the upper mantle. Elevated temperatures beneath a thin lithosphere *cause* the solidus to be exceeded and melting to occur. This melt rises to the surface *affecting* the generation of the Hawai’ian islands. The emplacement of low-density, hot, mantle beneath the plate *causes* an isostatic imbalance between the area directly beneath the Hawai’ian islands and the distal sea-floor. This imbalance *affects* the formation of the Hawai’ian swell as the sea-floor rises until isostatic balance is restored. For the record, we do not necessarily think that it is possible to infer that the formation of the Hawai’ian Islands can be causally linked to the edge of the Pacific LLSVP since there are numerous locations above the edge of the Pacific LLSVP where the equivalent of the Hawai’ian Islands have not formed. Reviewer 1 correctly points out, there is a large body of literature which shows that the presence of the Hawai’ian mantle plume ultimately controls the formation of Hawai’i. However, the presence of mantle plumes alone do not control the distribution of intraplate magmatism. If a mantle plume is rising through the mantle and is yet to reach the base of the lithosphere, or if a mantle plume rises beneath thick lithosphere, it will not produce intraplate volcanism. It is only when thermal anomaly is emplaced within the asthenospheric mantle that melting actually occurs— for us this point is the crucial one.

(3) “In the mantle plume model, its [the Hawai’ian Island’s] location at the surface is controlled by the Pacific LLSVP; the temperature anomaly in the asthenosphere is just a thermal expression. But not everyone thinks that Hawaii formed from a mantle plume anchored to the Pacific LLSVP. For example, Don Anderson and acolytes attribute it to localized hot ambient mantle, an interpretation that Ball and others would reasonably ascribe to intraplate volcanics at Eastern China, Mongolia etc. So, I think it is worthwhile adding a few words of clarification about the origin of these temperature anomalies.”

We neither agree nor disagree. It is important to point out that our central thesis and the principal conclusions drawn at the end of our manuscript are not predicated upon exactly how and where plumes are rooted at depth. Our analysis is strictly concerned with how upper mantle temperature variations affect the location and composition of intraplate magmatism. In this instance, Reviewer 1 has provided a perfect example of why we have chosen not to comment on the specific origin of these temperature anomalies. Whilst we agree with Reviewer 1 that Hawai’i is the surface manifestation of a mantle plume, our analysis cannot distinguish between, say, a localized Andersonian model and a more deeply rooted model. Both of these models seek to account for the presence of a thermal anomaly within the upper mantle. For clarity, we have added the following to the introduction “In this study, we are interested in exploring the quantitative relationship between the composition of volcanic rocks and calibrated seismic tomographic models with a view to constraining the size, extent and surface expression of putative thermal anomalies. We are less concerned with the more difficult problem of how these anomalies are generated in the first place”.

Reviewer 2 - Yaoling Niu

Professor Niu has written a very positive and helpful review. He remains “very much enthusiastic about this research for its publication in *Nature Communications*”. Whilst Professor Niu may disagree with certain aspects of our conclusions, he believes that “this will be an important paper” which he “looks forward to seeing its publication”. We are grateful to Professor Niu for his recognition of the fundamental and global importance of our manuscript and for his highly positive endorsement. He has identified a few key points within the manuscript and within the previous rebuttal where clarification is required. We thank Professor Niu for his careful review and we have endeavoured to address the points that he has raised as carefully as possible below.

Response to specific comments

(1) “I suggested the clarification “Why should there be an inverse correlation between mantle T_p and lithosphere thickness? There is no reason.”. (page 13) The authors partly disagree and cited the paragraphs of the definition by McKenzie and Bickle (1988). They seem eager to answer without understanding my question for clarification? Let’s give a simple example: Assuming mantle T_p is the same everywhere except for hot thermal mantle plumes, then the mantle T_p beneath ocean ridges (essentially zero lithosphere thickness) must be the same as beneath continental cratons (> 200 km lithosphere thickness). Hence, there is no reason mantle T_p correlates with lithosphere thickness. The correct situation is this: the conductive geotherm is steeper beneath ocean ridges than beneath cratons.”

We partly agree with the reviewer’s response to our original rebuttal. However, while he is correct to state that, in terms of global plate tectonics, lithospheric thickness changes are not necessarily coupled with changes in potential temperature, regional correlations are expected from a general fluid dynamical standpoint. The lithosphere represents the upper thermal boundary layer of the mantle convective system. Consequently, temperature changes within the convecting interior caused by the development and circulation of mantle plumes and blobs will indeed result in changes of the thickness of the upper boundary layer, *viz.* the lithosphere. Therefore, on regional scales there may be some degree of correlation between potential temperature and lithospheric thickness. For example, newer regional full-waveform tomographic models such as those published by Andreas Fichtner’s group in Zurich do show that lithospheric thickness changes do indeed show a positive correlation with shear-wave velocity anomalies (and so a negative correlation with inferred temperature anomalies).

Nevertheless, Professor Niu is correct in his assertion that these features are unlikely to correlate globally. It takes time for temperature variations in the asthenosphere to significantly affect the thickness of the thermal boundary layer. High asthenospheric temperatures will take time to thin the lithosphere, similarly when an asthenospheric anomaly is removed the lithosphere will take a long time to rethicken. These timescales are longer than those associated with the planform of mantle convection and plate motions. Consequently, we do not expect to see a global correlation between T_p and lithospheric thickness. In terms of the manuscript, we do not see a significant or clear correlation between asthenospheric T_p and lithospheric thickness. This lack of apparent correlation can be inferred from Figure 2e of the main text where there is no correlation between ΔV_s and lithospheric thickness beneath intraplate volcanic provinces at depths ≥ 150 km.

Professor Niu’s statement within the original rebuttal refers to a correlation between T_p at 150 ± 25 km depth and lithospheric thickness in Figure 5 of the main text. In regions of thin lithosphere, such as beneath intraplate volcanic provinces, we do not see a significant correlation between asthenospheric T_p and lithospheric thickness. However, T_p at 150 ± 25 km depth sharply decreases

in regions such as the North Sea and Tanzania where the lithosphere is > 125 km thick. This sharp decrease in T_p is expected since, in these particular regions, T_p is being calculated within the conductive lithosphere. Figure 1 of this rebuttal document shows how potential temperature at 125 km depth decreases as a function of lithospheric thickness if the lithospheric thickness is > 125 km, even if asthenospheric T_p is constant. To address the point raised by Reviewer 2, we have now added the following to the end of Section 3 “Note that in regions where the lithosphere is > 125 km thick, a correlation between ΔT_p and lithospheric thickness at depth of 150 ± 25 km is expected since these values of T_p sit within the lithosphere”.

(2) “ I suggested the clarification ”Removal or thinning of lithospheric mantle produces regional uplift - No! Basal removal or lithosphere thinning will NOT cause uplift, but if anything, will lead to subsidence due to Airy isostasy (Niu, 2019).” (page 16). The authors completely disagree (see my paper <http://community.dur.ac.uk/yaoling.niu/MyReprints-pdf/2019Niu-CSB-India-Asia.pdf>). An global observation based and fully quantitative paper is in preparation, but the geological and physical concepts are given in this paper in simple clarity. The original advocates (Graig Houseman, Peter Molnar and Philip England have read this short paper). If the authors ask Dan McKenzie now, he may give the same answer as mine although he was one of the original advocates (Houseman GA, McKenzie DP, Molnar P. Convective instability of a thickened boundary layer and its relevance for the thermal evolution of continental convergent belts. *J Geophys Res* 1981;86:611532.)”

We partly agree, and only in the case of cratonic lithosphere, which is not considered within this manuscript. The analysis presented by Niu (2019) focuses on the topographic response to lithospheric thinning in the Himalayas. He states that “... continental mantle of Precambrian age as a result of prior melt extraction is compositionally depleted and physically buoyant relative to the asthenosphere”. Consequently, when depleted lithosphere is thinned, subsidence will occur. In contrast, undepleted lithosphere is denser than asthenospheric mantle. An example is given by Niu (2019) of undepleted Tethyan oceanic lithosphere sinking into the asthenosphere. Removal of undepleted lithosphere by lithospheric mantle thinning will therefore generate topographic uplift as described within our manuscript. Since the vast majority of intraplate volcanism occurs along continental margins and within oceanic regions, where the lithosphere is not expected to be significantly depleted, we feel that our assertion that lithospheric thinning can trigger uplift is a valid one. To clarify this point within the manuscript we have changed “lithospheric thinning to “thinning of undepleted lithospheric mantle. The key issue concerns the degree of depletion which itself is a poorly understood process that it is difficult to assign precise values to. We conclude that Reviewer 2 has a point but it is one that is not directly relevant to our manuscript.

Reviewer 3

Reviewer 3 acknowledges that we have carried out “a substantial revision which in terms of content appears to address many of my and the other reviewers comments”. They also “appreciates the amount of effort that went into this, and the careful responses given to all comments”. We are very grateful for this highly positive endorsement of the work that we have carried out and for Reviewer 3’s implicit enthusiasm for our manuscript. However, Reviewer 3 clearly has misgivings about the writing style that we use in the manuscript. We agree that in both the abstract and the introductory paragraphs that our writing style is not as clear as it could be. In short papers, it is always difficult to write clear and cogent introductory material and we have paid close attention to improving this material during the revision process. Reviewer 3 has provided a series of specific typographic suggestions that pertain to the abstract and, more generally, to the first two pages of the manuscript. They have also requested that we carefully review the presentation of the rest of the manuscript. It is in our own interests to ensure that the published paper has as wide an impact as possible. We are therefore very grateful to Reviewer 3 for pointing out a variety of infelicities and obscurities that we have endeavoured to fix. Nevertheless, we do not agree with every single suggestion. In particular, we disagree with the reworded version of the abstract that Reviewer 3 has provided since we think that this version does not fully honour our intent. We have carefully rewritten the abstract in the light of Reviewer 3’s suggestions and we now feel that our reworded abstract is much improved. In the rest of the manuscript, we have tried to honour all of Reviewer 3’s suggestions and, where we do not agree, we have explained why and provided an alternative version.

Response to specific comments

(1) “The abstract is still worded very poorly and is difficult to follow for both experts and non-experts.”

We agree. We have completely rewritten the abstract following the suggestions outlined in Reviewer 3’s comments (2)–(9).

(2) “The paper does not establish how intraplate volcanism connects to mantle convection, as one might expect from reading the first two sentences. Rather it establishes what the relationship between intraplate volcanism and plate-mantle thermo-chemical structure is.”

We agree. We have changed the aim to constrain the thermal structure of the tectonic plates and asthenosphere..

(3) “The database is not one of intraplate volcanism, but of the products of this volcanism.”

We agree and we have changed volcanism to volcanic rocks.

(4) “The sentence the distribution of intraplate volcanism is associated with thin lithosphere is oddly worded. What you mean to say is that intraplate volcanism occurs where the lithosphere is thin.”

We agree. We have changed this sentence to say “intraplate volcanism is principally concentrated in regions characterized by slow upper mantle shear-wave velocities and by thin lithosphere”.

(5) “The next sentence is impossible to understand: There is a positive correlation between velocities at 150 ± 25 km depth and elemental ratios sensitive to the depth and

extent of melting: regions with large melt fractions match anomalously slow velocities. The first part of the sentence reads as if higher velocities correlate with larger depths and higher extents of melting, but then the second part says the opposite, i.e. that low velocities correlate with high melt fractions. Furthermore, in your response to R2, you emphasise that you do not interpret lower velocities as being due to higher extents of melting, as the second part of the sentence implies. The reader is left wondering what the authors are trying to say here”

Although we agree that this sentence is clumsily and obscurely worded, we do not agree that our statement is incorrect. There is indeed a positive correlation between the sign of shear-wave velocity anomalies and elemental ratios sensitive to melting. It is just that La/Sm decreases as melt fraction increases. So what we say is entirely correct but badly worded. We have simplified this sentence by removing reference to elemental ratios. Instead, we just state that “We observe a negative correlation between mantle seismic velocities at 125–175 km depths and melt fractions inferred from volcanic rock composition”. Simpler, clearer and to the point!

(6) “The sentence that follows “Modelling of rare earth tomographic models” is similarly hard to understand. Melt models indicate that isentropic melting occurs at variable temperatures? What is this supposed to convey, in particular to non-expert readers? “Globally calibrated tomographic models” is a very obscure description (calibrated against what?). Is this even useful to say in the abstract?.”

We agree. We have reworded this sentence to improve its clarity and to make it accessible to the general reader as requested.

(7) “The sentence “intraplate volcanism is often prevalent where elevated, but undeformed, marine sedimentary rocks show that Cenozoic regional uplift occurred” leaves it to the reader to decide what the potential implication of this correlation is.”

We partly agree. We have added a qualifying sentence which describes the relationship between these observations more clearly. “Regional elevation of these rocks can be generated by a combination of hotter asthenospheric mantle and lithospheric thinning”.

(8) “The final sentence We conclude that the distribution and composition of intraplate magmatism through geologic time represent a uniquely powerful window into past mantle conditions is a generic rather than specific conclusion.”

We disagree. It is very important that the abstract concludes on a general note. Our generic conclusion provides an appropriate ending to this abstract.

(9) “To illustrate better what I mean, this could be an alternative formulation of the first part of the abstract: “Thermal structure of the plates and mantle exerts a primary control on the evolution of surface topography and the distribution of volcanism. Only indirect observations of the distribution of temperature and its change through time are available. Here we compile a global data base of geochemical parameters for Neogene-Quaternary rocks produced by volcanism away from plate boundaries. When analysed in conjunction with tomographically-imaged seismic structure, this data provides new insights in the relation between magmatism, thermal structure and surface uplift. We find that intraplate volcanism occurs where the lithosphere is thin (< 70 km) and mantle shear wave velocities between 125 and 175 km depth are relatively slow, indicating that the mantle is warm compared to the global average temperature”. I find it hard to suggest how to reword the rest of the abstract as the meaning is too unclear.”

We disagree with Reviewer 3's reworded abstract since we do not believe that it fully conveys the work that we have carried out. In the end, writing style is a matter of taste but we agree that we need to be clearer in our exposition. The abstract must be less than 180 words to be accepted by *Nature Communications*. The alternative abstract beginning suggested by Reviewer 3 is 111 words— $\sim 50\%$ longer than the original abstract. Although elongating the text may improve clarity, it would come at the expense of the rest of the abstract. We have completely rewritten the abstract to try to satisfy Reviewer 3 but in order to fully explain the results of our analysis we must be concise in places.

(10) “First sentence of the paper “Thermal and chemical heterogeneity of the Earths mantle generates and maintains vigorous convection” is poorly worded. Thermal structure is largely the consequence and hence an expression of mantle convection, not the driver of mantle convection. Similarly, it is likely that the distribution of compositional heterogeneity is largely a consequence and not a driving force of convection. There is no need to enter into a debate about this in the first sentence. You could write instead “Mantle convection redistributes heat, cycles chemical species, drives plate tectonics, and generates dynamic topography.” This confusion over cause and consequence affects also the rest of this paragraph which keeps discussing mantle convection as if it is caused by thermal structure. What you probably mean to say in this paragraph is that a better mapping of thermal and chemical structure of the lithosphere and mantle can provide insight in the dynamics of mantle convection and plate tectonics. However, this is not what the paragraph says.”

We neither agree nor disagree but we take Reviewer 3's point. We disagree with this reviewer's assertion that thermochemical heterogeneity is a consequence and not the driver of mantle convection. For example, in the well-known textbook *Solid Earth* by Fowler “convection in liquids occurs when the density distribution deviates from equilibrium. When this occurs, buoyancy forces cause the liquid to flow until it returns to equilibrium”. It is hotter at the core mantle boundary than at the surface. Since the density of mantle material is inversely proportional to temperature, this thermal heterogeneity generates mantle convection as hot buoyant material rises from below and cold dense material sinks from above. These thermal density variations are superimposed by those generated by compositional heterogeneity, which provides a secondary, but important, forcer. Rather than generating thermal heterogeneity, mantle convection acts to move the mantle into a state of thermal equilibrium. These issues are all very interesting but our intention is not be to provocative right at the start of the manuscript so we think that it is simpler to remove this sentence and start the manuscript in a less contentious way!

(11) “Third sentence of the paper It is especially important to quantify the relationship between volcanic activity and mantle convection away from complications associated with active plate boundaries in order to understand the geodynamic interactions between mantle composition, melting and temperature through geologic time. This is a long and convoluted sentence. Understanding that intraplate volcanism occurs where plates are thin does not by itself help understand plate-mantle dynamics. I would say that volcanic activity away from the main plate boundaries provides an expression of the thermal and chemical state of the plates and mantle and as such can provide important constraints that complement those provided by geophysical observations.”

We agree that the offending sentence is far too long and rambling but we also disagree with this suggested rewording by Reviewer 3. Their alternative version does not provide information as to why we are looking at areas away from plate boundaries. Furthermore, we aim to quantify the relationship between intraplate volcanism and mantle convection using both geophysical and

geochemical approaches. In this alternative version it reads as if we will be providing information which complements rather than incorporates geophysical observations. We have therefore chosen to keep but revise the original sentence.

(12) Line 31 – “Mantle potential temperature, T_p , plays a fundamental role in generating melt.” say what the role is that T_p plays in generating melt.

We have removed this sentence and we have added “For a given mantle source composition, the depth and degree of melting are principally controlled by a combination of asthenospheric temperature and lithospheric thickness (McKenzie and Bickle, 1988). The extent of melting increases with increasing potential temperature, T_p , and decreasing pressure so that elevated asthenospheric temperature and/or thinner lithosphere produce greater volumes of melt”.

(13) Line 32 – “composition of volcanic eruptions” not the composition of volcanic eruptions, but of erupted volcanic rocks.

We agree and we have changed “volcanic eruptions” to “volcanic rocks”.

(14) Line 33 – “In addition to temperature, the distribution and composition of volcanic rocks are significantly influenced by mantle composition” In addition to being influence by temperature, the distribution and composition

We have changed the wording of this sentence.

(15) “Line 35-36 – global tomographic models enable shear-wave velocity anomalies to be mapped. No, tomographic models are images (maps) of shear-wave velocities. You could say Global seismic tomography enables mapping of shear-velocity anomalies. Better might be Global data sets of surface wave and body-wave travel times and/or waveforms allow mapping of shear-wave velocities”

We agree. Since seismic tomography is an act of mapping we have changed “Global seismic tomographic models enable shear-wave velocity anomalies, ΔV_s , to be mapped throughout the upper mantle.” to “Global seismic tomographic models map shear-velocity anomalies, ΔV_s , throughout the upper mantle.”

(16) Line 38 – “It is also sensitive” what does it refer to?

We agree and we have changed “It is” to “Shear-wave speeds are”.

(17) “Line 40 – “A combined geochemical and geophysical analysis of the distribution and composition of igneous rocks”. Again, this is poorly worded. It sounds like you are doing a combined geochemical and geophysical analysis of the igneous rocks and a combined geochemical and geophysical analysis of shear-wave velocity anomalies. You could write “A combined analysis of the distribution and composition of igneous rocks and shear-wave velocity anomalies.””

We agree and we have changed “A combined geochemical and geophysical analysis of the distribution and composition of igneous rocks and shear wave velocity anomalies” to “Analysing the distribution and composition of igneous rocks in conjunction with shear wave velocity anomalies”

(18) Line 42 – “This approach has been successfully used to analyze temperature variation along mid-oceanic ridges.” say what this analysis found rather than just saying an analysis has been done on MOR.

We disagree since we provide references so that interested parties can read the full study if they wish to. We feel expansion of this point is not necessary for the general reader and have chosen to keep it as is.

(19) Line 43 – “Here, we are interested in investigating a more general problem, namely the global relationship between intraplate volcanism and upper mantle shear-wave velocity anomalies” I think your aim is to constrain temperatures and composition, not to relate intraplate volcanism to shear wave velocity anomalies.

We agree and we have altered this sentence to say “we are interested in investigating a more general problem, namely the global relationships between intraplate volcanism, upper mantle shear-wave velocity anomalies and the thermochemical structure of the upper mantle.”

(20) Line 50 – “Finally, we scrutinize the link between intraplate volcanism and the distribution of elevated marine sedimentary deposits that manifest dynamic topographic uplift.” deposits that are a manifestation of dynamic uplift. This needs a better explanation. Why are elevated marine deposits an expression of dynamic uplift? Why do you care about dynamic uplift?

We partly agree but we have changed “Finally, we scrutinize the link between intraplate volcanism and the distribution of elevated marine sedimentary deposits that manifest dynamic topographic uplift.” to “ Finally, we scrutinize the link between intraplate volcanism and the distribution of emergent marine sedimentary deposits, which are a tangible manifestation of dynamic topographic uplift”.

(21) Line 60 – “The majority of analyses are mafic samples” analyses do not equal samples.

We agree and we have changed “analyses are mafic samples” to “analyses are of mafic samples”

(22) Line 63 – “We have used a combination of review literature and local studies” what do you mean to say here? you have done a review of the literature and performed local studies to identify samples?

We agree and we have changed “We have used a combination of review literature and local studies to identify and remove samples pertaining to subduction zone processes (Supplementary Table 3; e.g., Klöcking et al., 2018; McNab et al., 2018)” to “We have carried out a literature review to identify and remove samples pertaining to subduction zone processes (see Supplementary Materials).”

(23) Line 65 – “database shows that most intraplate volcanism is concentrated within bands positioned on continental lithosphere away from thicker cratons” the data base compiles the location and chemistry of intraplate volcanic rocks, i.e. by itself can not show that these occur away from cratons. thicker cratons - thicker than what? is intraplate volcanism found on thinner cratons?

We fully agree and we have reworded this sentence accordingly.

(24) Line 68 – “The lower number of database entries from the oceanic realm” lower than what?

We agree and we have altered this statement to read “The smaller number of database entries from the oceanic realm compared with the continents”

(25) Line 70 – “Figure 1a demonstrates that there is a striking visual association” what kind of association is there between intraplate volcanism and shear wave velocities?

Why not write something like: “When we compare the distribution of intraplate volcanism with several global tomographic models of shear-wave velocities, we find that there is a striking association between regions where intraplate volcanism is concentrated and low velocity anomalies. Figure 1 illustrates this association for model SL2013sv. Model SL2013sv is a global tomographic model based on Rayleigh wave dispersion curves for periods X to Y seconds. Other global models based on combined surface and body wave data sets or inversions of full waveforms, as well as regional models yield the same type of correlation (Supplementary Fig YY)”.

We only partly agree. We have adapted the opening of this paragraph in accordance to some of the suggestions listed here. However, we have decided keep all description of the tomographic model within the Methods section for clarity and conciseness. “If the distribution of volcanic rocks is compared with the pattern of upper mantle shear-wave velocities, it is clear that intraplate volcanism is concentrated within regions where negative shear-wave velocity anomalies occur at depths of 150 ± 25 km. Figure 1a shows this striking visual association for the SL2013sv surface wave tomographic model, where ΔV_s is calculated relative to a three-dimensional reference model (see Methods; 21)”

(26) Line 71 – “Intraplate volcanism and shear wave velocity anomalies, ΔV_s , at 150 ± 25 km depth” this is the first time shear-wave velocity anomalies are introduced, yet you already discuss a comparison above.

This statement is incorrect. We introduce shear wave velocities on Line 29, and shear wave velocity anomalies on Line 35.

(27) Line 73 – “calculated relative to an adapted version of Preliminary Reference Earth Model” if the velocity anomalies are relative to an adapted version of PREM, then a reference to PREM is not helpful. Furthermore, this statement is not meaningful unless you add a comment about what this reference model may mean (in terms of physical state). This is probably too much detail for this part of the manuscript, so better to discuss this in the methods.

We agree and we have changed “an adapted version of Preliminary Reference Earth Model” to “a three-dimensional reference model”. We then describe the reference model within the methods. We link to the Methods at the end of the sentence.

(28) Line 75 – “which show that mafic geochemistry and ΔV_s correlate” be precise. correlate in what way?

We agree and we have changed “mafic geochemistry and ΔV_s correlate within intraplate settings” to “ ΔV_s correlates with geochemical indicators of melt fraction variations within intraplate settings”

(29) Line 77 – “There is a similarly compelling relationship between intraplate volcanism and lithospheric thickness” what type of relationship?

We agree that this sentence is obscurely worded and we have altered it. In the next few sentences, we describe the relationship between intraplate volcanism and lithospheric thickness.

(30) Line 78 – “SL2013sv model” please properly introduce the shear wave model before moving onto the conversion.

We agree and we have added “see Methods” at the end of this sentence.

(31) Line 79 – “global calibration between Vs and T, based upon the oceanic plate cooling model” please word this more clearly “a calibration where Vs structure under the oceans is assumed to satisfy the plate cooling model”

We disagree. We do not believe this suggested change is required. If the reader is interested in a full description of the model we refer them to the methods at the end of the next sentence.

(32) Line 80 – “revised and modified Vs-to-T parameterization described by ref. (25), which is based upon an empirical anelastic parametrization” this is vague, what is significant about this new parameterisation? why is it both revised and modified?

We are unsure what the problem is here. We link to the methods at the end of the sentence for more information. We do not believe any more information is required within the main text itself.

(33) Line 81 – “Lithospheric thickness is calculated by” not calculated, but mapped

We agree and we have changed “calculated” to “estimated”.

(34) Line 83 – “distribution of intraplate volcanism is closely associated” you mean to say that intraplate volcanism is concentrated where the lithosphere is thinner than 100 km.

We agree and we have changed “the distribution of intraplate volcanism is closely associated with lithosphere that is thinner than 100 km” to “intraplate volcanic rocks are concentrated within regions where the lithosphere is < 100 km thick”.

Figure 1: Cartoon depiction of potential temperature, T_p . (A) cross-section showing lithospheric thickness as a function of distance. Solid black line = base of lithosphere; dashed black lines = loci of geotherms depicted in panels B–D; dashed red line = 125 km depth; red circles = intersection between geotherms and 125 km depth. (B)–(D) Geotherms at locations marked in panel A. Solid black line = geotherm; dashed black line = adiabatic decompression from 125 km depth; dashed red line = 125 km depth; red circle = intersection between geotherm and 125 km depth. Asthenospheric T_p assumed to be 1330 °C and the adiabatic temperature gradient is assumed to be 0.4 °C km⁻¹.

References

- J. Barrell. The strength of the Earth’s crust. *The Journal of Geology*, 22(7):655–683, 1914.
- S. T. Crough. Thermal origin of mid-plate hot-spot swells. *Geophysical Journal International*, 55(2):451–469, 1978.
- S.-i. Karato. On the origin of the asthenosphere. *Earth and Planetary Science Letters*, 321:95–103, 2012.
- M. Klöcking, N. J. White, J. Maclennan, D. McKenzie, and J. G. Fitton. Quantitative relationships between basalt geochemistry, shear wave velocity, and asthenospheric temperature beneath western North America. *Geochemistry, Geophysics, Geosystems*, 19:3376–3404, 2018.
- D. McKenzie and M. J. Bickle. The volume and composition of melt generated by extension of the lithosphere. *Journal of Petrology*, 29(3):625–679, 1988.
- F. McNab, P. W. Ball, M. J. Hoggard, and N. J. White. Neogene uplift and magmatism of Anatolia: Insights from drainage analysis and basaltic geochemistry. *Geochemistry, Geophysics, Geosystems*, 19(1):175–213, 2018.

Reviewer #3 (Remarks to the Author):

The revised manuscript has significantly improved and is a lot more readable. With this revision, I can now review the changes that were made in the previous revision round.

There are still a few sections of the text that are confusing or incomplete. I do not agree with the authors that fixing the writing in this manner is just a matter of writing style. It is really a matter of clarity and precision. I do hope the authors can bear with this for one more round, because I do think some more corrections are necessary so that readers can properly understand and evaluate the analyses.

However, after that, I would recommend this paper for publication in Nature Communications

Main comments

(1) Description tomographic model SL2013sv: Paragraph starting line 81 - please give here the main characteristics of the tomographic model used. SL2013sv is not purely a surface-wave model; it is actually based on waveform modelling including body and surface waves (as said in the methods). It is also worth noting that this is an SV model, not isotropic S velocity. And the statement that this is relative to a 3-D reference model will just cause confusion at this point. The reference model is only 3-D in the crust, i.e. not at the depths you are analysing here. Showing velocity anomalies relative to a 3-D reference model would not be meaningful unless the readers know what the 3-D reference is. I would suggest to not mention the reference model here. But rather in the method section, where this is more properly discussed, please do not only refer to a publication but say what reference crustal model is used (e.g. is this CRUST1.0, a model that many readers will be familiar with)?

(2) Motivation of definition of lithospheric thickness: Line 94 - using the 1175°C isotherm as the base of the thermal lithosphere requires more motivation. There is a lot of discussion in the literature about how to define the base of the lithosphere. The choice made here is fine, but a motivation for the particular isotherm is warranted. It is also important to emphasise that this is a thermal definition of lithospheric thickness. It also needs to be motivated somewhere in the manuscript how the thus defined thermal base of the lithosphere would relate to the top of the melting column which is used in the modelling of REE concentrations. Would this be the same isotherm and why?

(3) Seismic lithospheric thickness: Line 113 - It is a key point for the whole analysis that intraplate volcanism appears to mainly occur in lithosphere with thermal thickness, as inferred from DVs, of < 100 km. It would help the rest of the paper to include a (supplementary) figure that shows the distribution of seismically inferred lithospheric thicknesses in the locations of intraplate volcanism. The map shown in Fig. 1 only shows that lithosphere in these locations is thin, but not how thin. Such a figure would help motivate the range of lithospheric thicknesses used in the later REE modelling. Some of the thicknesses tested are close to the base of the Moho, meaning that you infer there is essentially no mantle lithosphere below parts of the continent. This would warrant at least a comment.

(4) Motivate data base filtering Line 114 and rest of this paragraph - It is a jump for the less expert reader why you suddenly start talking about mineral loss or accumulation and why you are concerned about MgO. I think you need to first tell the reader that you are filtering the data base to ensure you analyse only those samples that are most likely to represent primary mantle melts rather than those that are affected by significant mineral loss or accumulation during their rise through the lithosphere.

(5) Asthenospheric velocity anomalies. Line 136 and following. Another important point for the analysis is the assumption that DVs at a depth of 150 ± 25 km represents asthenosphere below the regions of intraplate volcanism. Note that a 25-50 km vertical resolution is a rather optimistic estimate. And also, as mentioned in the text, the thermal boundary layer extends deeper than the chosen definition of thermal thickness, and may well reach depths that are 50 km or more below the lithospheric thickness mapped. Hence it would be useful to further convince the reader that the seismic structure at 150 km depth does indeed not reflect a signature of lithospheric thickness variations. This could be done for example by including a (supplementary) figure that shows that the correlation of velocity structure above 100 km depth with velocity structure below this depth is relatively low in the areas of intraplate volcanism. Or it could be done by showing that the depth of a higher temperature isotherm (e.g. 1250 or 1300°C, closer to but still below reference mantle potential temperature) does not extend to 150 ± 25 km below the regions of intraplate volcanism.

(6) Principal component analysis Line 153 and following. This paragraph is quite confusing to read, because the principal component analysis and subsequent correlation with La/Sm, DVs and eNd are all introduced together. It would help the reader to first explain the PCA, and what this yields, before introducing the correlation analysis. You do a PCA for your global data base of volcanic rock compositions using a subset of the concentrations of eight incompatible elements. You need to tell the reader what the subset is taken from not just say you do a PCA on a subset of eight incompatible elements. It may help to give your data base a name, so that you can reference it more easily throughout the text.

(7) Correlation of principal components and other parameters: Line 165 and following. You need to explain why if P1 were controlled by mantle depletion, P1 would negatively correlate with DVs. It is not obvious to the reader what positive and negative P1 means. Source depletion would lead to relatively high Vs, relatively high eNd and low degrees of melting i.e., high La/Sm. As I understand it, low degrees of melting would also mean high P1 (high concentrations of most incompatible elements). This would mean a positive correlation between P1, La/Sm and DVs in the case of depletion. High temperature would similarly lead to a positive correlation of these three parameters. So to my understanding it is solely the negative correlation between P1 and eNd that indicates that source depletion is not the main influence on P1.

(8) Synthetic principal component analysis: Line 209 and following. This analysis really strengthens the interpretation of the principal components. It would help if it were discussed before the interpretation of P1, P2 and P3 is proposed.

(9) REE modelling - Line 243-344 - a hydrous melting model is used. Please mention in the main text where water content is taken from, as this is a key parameter, just like you mention that source depletion is estimated from eNd.

(10) Misfit measure REE modelling: What motivates accepting only models where misfits are < 1.5 minimum misfit? In principle, all models that fit within the uncertainties of the data are acceptable. What is the justification for excluding models that fit within the data uncertainties but have misfits > 1.5 times the minimum misfit?

(11) Differences geochemical and geophysical T_p : Line 283 and following. One might expect that damping applied in tomographic models would reduce the range of inferred T_p , yet the seismic range of T_p is larger than the geochemical range of T_p . This paragraph mentions many of the uncertainties in interpreting tomographically-imaged seismic velocities in terms of temperature, except the possible influence of the presence of melt. If melt contributes, and it would contribute more to the lowest seismic velocities, this might explain the larger range of seismic T_p than geochemical T_p .

(12) Correlation geochemical and seismic T_p : Note that the reasonable correlation between T_p from geochemical and seismic estimates in Fig. 5 is quite strongly influenced by the highest T_p points. It would be good to see what the correlations look like when done with the T_p from the other seismic tomography models shown in supplementary Fig. 2, i.e. add another supplementary figure with these correlations.

(13) Dynamic topography: line 350 and onwards. The term dynamic topography is used here to refer to changes in isostatic topography due to changes in local mantle thermal structure and lithospheric thickness rather than in the sensu-stricto meaning of the term dynamic topography as topography in response to the upward or downward pressures from mantle flow. It would be good to clarify the use of this terminology at the start, or preferably use a different terminology, e.g. adjustment of isostatic topography.

Small comments

Line 31 and 37 - “upper mantle” should be “uppermost mantle”, as the upper mantle extends down to 700 km depth, i.e. much deeper than lithosphere + 100-200 km thick asthenosphere

Line 42 - “calibrated seismic tomographic models”. In what way are these calibrated? The conversion to temperature is calibrated, but the tomographic models themselves are not or?

Line 55 - “seismic tomography” should be “seismic velocities”

Line 70 - “sub-surface observations” should be “sub-surface structure”. The observations are taken at the surface

Line 121 - the reader could use some additional help with understanding that “positive correlation” means low degree of melting correlates with high seismic

velocities

Line 151 - "melting occurs to a lesser degree within a depleted source region since depleted mantle has lower initial La/Sm". This sentence does not make sense.

Line 187 - "elevated asthenospheric temperature acts to depress the solidus" This is not true. The solidus does not change as a function of asthenospheric temperature. The depth where melting starts changes if asthenospheric temperature changes.

Line 254-256 - last sentence of this paragraph "Volcanic province.. combinations" just seems to make the point that if the spread of data is larger more models can fit. This seems trivial.

Line 285 - Why the emphasis on "great-circle ray path coverage"? Why not just "ray path coverage"

Line 339 - "This inference is manifest by a range of geochemical and geophysical observations" This sentence does not add any information. It could be cut.

Line 347 - "major element compositions" - do you mean "major element compositions of mantle melts"? Or are you referring to composition of the source (as discussed in the following sentence)?

Line 348 - "these correlations could also be partially modified by". It is unclear what "also" refers to. You need to say that the correlation was modelled with variations in mantle temperature. "partially modified by" - do you mean "partially explained by"?

Line 352 - "rapid regional uplift" The rapidity of uplift would depend on the speed with which lithospheric thinning or change in asthenospheric temperature occurs?

Line 379-380 - "vertical motions... superimposed on motions generated by mantle convection". It is the latter motions (by mantle convection) that are usually referred to as "dynamic topography" whereas lithospheric thinning leads to isostatic adjustments of topography.

Fig. 4. The caption of this figure needs to refer to the equation used to calculate RMS (in the methods). Different definitions of RMS are possible. It would also be good if panel c and d could outline the range of accepted solutions to show how this range may differ from the range of models that fall within 1 standard deviation (RMS=1)

Line 413 - What has been recomputed? Has a new 1-D reference model been derived? What crustal model is used as reference?

Line 418-429 - First paragraph Section 8.2. The description of the conversion method starts with secondary details that the method is "pioneering" and based on data from borehole experiments. None of this gives the reader any preview of what the method actually involves. It would be much more useful if you could give a brief summary of the method before explaining the details, i.e. the conversion method is based on a calibration of parameters to maximise the agreement between seismic

velocities imaged below the oceans and a plate cooling model overlying an adiabatic mantle of constant potential temperature. The calibration is used to set 6 material parameters and the adiabatic temperature-depth gradient. The anharmonic shear modulus is assumed to vary linearly with P and T and attenuation is adjusted by varying the reference steady-state viscosity and an activation energy and volume. It is a decision rather than a necessity to adjust these 7 parameters in the fitting.

Second paragraph Section 8.2. It would help to introduce H_i at the end of the first line “compared to four sets of observational constraints H_i .” The next line would be clearer if it read “The total misfit H , between calculated and observed V_s , is quantified...”

Line 440: Why is the isentrope for pyrope used? Why not for a peridotite?

Line 440: What data base was used with the PerPle-X software package?

Line 447: What bulk viscosity was assumed? And why was bulk viscosity used rather than shear viscosity?

Line 481: “the global data base”, please be specific about which global data base. I assume this is your own new data base. You may want to give it a name so that you can refer to it here.

Line 523: Please give references for the PM and DMM models used, either here or in the supplement

Line 525: What X_{H_2O} was used?

Line 580: Please give a reference for the choice of $Fe^{3+}/total\ Fe$

Supplementary Fig. 5 through 8 - please make clear in the captions which of these are PCA of the data base and which from the synthetic models

Supplementary Fig. 8 labels (a), (b) etc missing on the figure panels but referred to in the caption.

Global Influence of Mantle Temperature and Plate Thickness on Intraplate Volcanism

P. W. Ball, N. J. White, J. Maclennan and S. N. Stephenson

Response to Reviewers' Comments

Reviewer 1

Reviewer 1 states that they are “reasonably satisfied with the final revisions and recommend this paper be published in its present form”. We are grateful to this reviewer for their positive endorsement and for the insightful comments that they provided in previous reviews which enabled us to greatly improve this manuscript and its potential impact.

Reviewer 2 - Yaoling Niu

Professor Niu states that he is “happy with this new version that has largely answered (his) queries” and that he is “enthusiastic about this work and looks forward to its publication for continued discussion on this important global problem”. In his review, Professor Niu suggests that we “soften” our conclusions (regarding the significance of plate thickness) since they are “inconsistent with the observations”. We carefully addressed a similar statement by Professor Niu (see his Comment 16 from the first set of reviews). We take this fresh comment equally seriously. It is most important to emphasize in the first instance that our results quantitatively support the importance of thin lithosphere (see our Figures 1b and 4, for example). Although we stand by our original assertion of the importance of temperature, for which we provide observational support, we have further amended the manuscript to reflect that there are previous studies which reached (slightly) different conclusions. Furthermore, we have also highlighted that this field is a subject of ongoing debate. Moreover, we now readily concede that major elemental compositions are likely to be affected to a greater extent by lithospheric thickness as opposed to potential temperature, T_p , alone since these elements can become re-equilibrated at the base of the lithosphere. In summary, we do not believe that Professor Niu and ourselves have any fundamental differences with respect to the moderating influence of lithospheric thickness which we have always acknowledged. The point that he is making is essentially a nuanced one and, in terms of the global results that we present, it is at the edge of what is possible to resolve. In his review, he has raised a suite of more specific comments which we have addressed in the following way.

Response to specific comments

(1) “In terms of basic physical and petrological concepts, mantle melt compositions and the inferred extent of melting must be controlled by [1] fertile mantle compositional variation/heterogeneity, [2] mantle potential temperature variation at which decompression melting begins, and [3] and the lithospheric lid thickness variation where decompression melting ceases to continue (capped by the lid). [1] can be averaged out using large sample sets on large local/regional scales with the means reflecting [2] and [3]; [2] cannot be seen in any straightforward way; [3] is much easier to see because it is geologically and petrologically straightforward without the need of any complex data manipulation or sophisticated interpretation.

We mostly agree with the logical framework proposed here by Reviewer 2. In our study, we have endeavored to compile the largest global geochemical dataset possible in order to limit the effect of mantle heterogeneity [1]. However, we do not believe it is quite so easy to separate out the effects of potential temperature [2], and lithospheric thickness [3], using trace elemental concentrations. Inevitably, both processes contribute to the depth range and extent of melting so that separating their contributions in regard to melt composition is difficult. Hence, in two predominately data-driven studies (i.e. this Manuscript and Niu et al., 2011), it is possible to arrive at different conclusions depending upon which datasets are exploited.

Our conclusion that lithospheric thickness has a secondary influence upon composition is being rather overstated but it is, nonetheless, consistent with our observations. However, we recognise that the precise wording of this conclusion might not be shared by everybody and that it is important to highlight previous studies which reach nuanced alternative conclusions. Therefore, we have included additional discussion to the revised manuscript which suggests that the lid effect can be more important in some instances, particularly for major element compositions which we do not consider in our contribution but which comprise the bulk of the analysis presented by Niu et al. (2011). Melts may re-equilibrate with their surroundings at the base of the plate. This re-equilibration will affect major element compositions but, crucially, not trace elements. Therefore major elemental compositions may be more sensitive to the lid effect than to T_p as such. In making these changes, we believe we have now softened our conclusions to an appropriate degree that satisfies Reviewer 2's final concerns. We also emphasize that our Figures 1b and 4 plus associated commentary make it very clear that we do indeed regard thin lithosphere as a significant part of our story.

(2) For example, [3-a] The lid effect is conspicuous and well preserved in the global ocean islands - This is demonstrated by Niu et al. (2011), which is cited in the Ball et al. manuscript. We demonstrated that the lid effect is readily seen. But, objectively, we do not see the effect of mantle potential temperature variation. Hence, the lid effect is the primary control in terms of OIB compositions and the inferred extent of melting. Because we do not see the latter, we can state objectively that the effect of mantle potential temperature is secondary. Please note: [2] is observed and [3] is interpreted (guessed). See Figure 1 and interpretation of Niu et al. (2011); also see concise summary in this paper: Niu and Green (2018).

We essentially agree: we greatly admire the work presented by Niu et al. (2011) together with the data-driven global approach that it employs. However, as with all scientific contributions, the observations and results are inevitably subject to the underpinning assumptions of that particular study. For example, Niu et al. (2011) assume that lithospheric thickness within the oceanic realm follows a plate cooling trend. They compare their assumed lithospheric thicknesses to geochemical data in order to demonstrate the importance of the lid effect. Whilst this assumption is reasonable and logical, it may not always hold in locations where mantle plumes interact with the bottom of the plate. In the decade since publication of Niu et al. (2011), new lithospheric thickness maps generated by converting tomographic models into temperature have become readily available (e.g., Priestley and McKenzie, 2013). In future, these maps may offer better constraints for lithospheric thickness than plate age as such.

Here, we offer two examples of possible problems for the plate-age assumption:

- Since Iceland sits above the mid-oceanic ridge, it is given a lithospheric thickness of 0 km by Niu et al. (2011). However, crust up to ~ 45 km thick has been imaged by receiver function analysis beneath Iceland, and so it is very unlikely that the mantle decompresses to a depth

of 0 km (i.e. the surface) at this location (Jenkins et al., 2018). In contrast, Iceland has a lithospheric thickness of ~ 50 km in the model we exploit.

- Hawaii is on old oceanic floor which should have cooled to a lithospheric thickness of ~ 90 km (Niu et al., 2011). In the tomographic lithospheric thickness model we use, Hawaii has a lithospheric thickness of ~ 60 km (Hoggard et al., 2020). This thinner lithosphere is in keeping with flexural and gravimetric observations, as well as with xenolith-based thermobarometric results which indicate that temperatures of 1000–1100 °C occur at depths of 45–55 km beneath Hawaii (Guest et al., 2020; Pleus et al., 2020).

Given that Iceland and Hawaii probably represent the largest datasets with the Niu et al. (2011) database and given that these provinces occur at the minimum and maximum lithospheric thicknesses in their analysis, the correlations and results generated using the methodology of Niu et al. (2011) could well change if a modern plate thickness map is applied to their dataset.

We hasten to add that we do not provide these examples as any criticism of Niu et al. (2011). We merely point out that circumstances can change when newly acquired observations are brought to bear upon a given problem. The same conditionality applies to our own analysis and, for this reason, we now provide an extra discussion by referring to work such as Niu et al. (2011) so that the general reader can make up their own mind by evaluating studies which reach alternative viewpoints. Moreover, since our study only analyzes trace element compositions, we cannot easily comment upon the relative importance of T_p and lithospheric thickness with respect to major element concentrations which make up the bulk of the analyses exploited by Niu et al. (2011).

(3) [3-b] In continental settings, the lid effect on basalt compositions and the extent of melting is conspicuous on spatial scale of ~ 250 km, where mantle potential temperature must be the same. Hence, we see [3], but we do not see [2] and [2] is actually has no effect: See this paper: Guo, P.Y., Niu, Y.L., Sun, P., Gong, H.M., Wang, X.H., 2019. Lithosphere thickness controls the continental basalt compositions: An illustration using the Cenozoic basalts from eastern China. *Geology* 48, 128-133. This demonstrates [3] in simple clarity and that [2] has no effect.

[3c] [3-c] This is another simple demonstration on the lid effect of continental intraplate basaltic magmatism away from any plate boundaries: Sun, P., Niu, Y.L., Guo, P.Y., Duan, M., Wang, X.H., Gong, H.M., Xiao, Y.Y., 2020. The lithospheric thickness control on the compositional variation of continental intraplate basalts: A demonstration using the Cenozoic basalts and clinopyroxene megacrysts from eastern China. *Journal of Geophysical Research* 125, pp e2019JB019315.”

We wholeheartedly agree that on a local scale in some locations, lithospheric thickness variations can have more of an effect on melt composition than T_p variations. However, we are sure that the Reviewer would agree that the opposite can also be true. For example, the alkali-tholeiite-alkali cycle recorded on the Hawaiian islands is thought to result from the hottest part of the mantle plume passing beneath each island in turn. Furthermore, the compositional differences between Iceland and other locations along the Atlantic mid-oceanic ridge are caused by the presence of the Icelandic hotspot rather than being a consequence of differences in plate thickness. The conclusions of these important and interesting local studies do not necessarily have over-arching implications for the results of our global analysis, where we consider large-scale trends and patterns.

Reviewer 3

Reviewer 3 now believes that our “revised manuscript has significantly improved and is a lot more readable”. However, they state that “there are still a few sections of text that are confusing or incomplete” and so “some more corrections are necessary”. As a result, Reviewer 3 recommends that our paper for publication in *Nature Communications* if we update the text in light of their suggestions. We take this careful reviewer’s comments and criticism very seriously indeed and we have endeavored to carefully address all the points outlined by them. We hope that they will now agree that this study is worthy of publication.

Response to specific comments

(1) “Description tomographic model SL2013sv: Paragraph starting line 81 - please give here the main characteristics of the tomographic model used. SL2013sv is not purely a surface-wave model; it is actually based on waveform modelling including body and surface waves (as said in the methods). It is also worth noting that this is an SV model, not isotropic S velocity.”

We fully agree and we have now included the following within the main text— “This vertical shear wave model uses body and surface waves that include both fundamental and higher modes with periods of 11–450 s”.

(2) “ The statement that this [the SL2013sv tomographic model] is relative to a 3-D reference model will just cause confusion at this point. The reference model is only 3-D in the crust, i.e. not at the depths you are analysing here. Showing velocity anomalies relative to a 3-D reference model would not be meaningful unless the readers know what the 3-D reference is. I would suggest to not mention the reference model here. But rather in the method section, where this is more properly discussed, please do not only refer to a publication but say what reference crustal model is used (e.g. is this CRUST1.0, a model that many readers will be familiar with)?”

We fully agree and we have removed the statement about the reference model from the main text. We have now added a reference to the starting crustal model within the Methods section where we have also added further clarity (Crust2.0; Bassin et al., 2000).

(3) “Line 94 - using the 1175 °C isotherm as the base of the thermal lithosphere requires more motivation. There is a lot of discussion in the literature about how to define the base of the lithosphere. The choice made here is fine, but a motivation for the particular isotherm is warranted. It is also important to emphasise that this is a thermal definition of lithospheric thickness.”

We agree with this nuanced point. We have now added a brief explanation alongside appropriate references for our choice of lithosphere-asthenosphere boundary. We have also emphasised it is thermally defined.

(4) “It also needs to be motivated somewhere in the manuscript how the thus defined thermal base of the lithosphere would relate to the top of the melting column which is used in the modelling of REE concentrations. Would this be the same isotherm and why?”

We mostly agree. Although the base of the lithosphere presented by global thickness maps employed throughout this study is thermally defined, it is constrained by mechanical observations. 1175 °C was chosen as the base of the lithosphere by Richards et al. (2018) in their revised oceanic plate

cooling model. This particular isothermal surface was chosen since it coincides with the peak change in orientation of azimuthal anisotropy within the Pacific Ocean. The change in orientation is thought to demarcate the transition between rigid lithosphere, where the olivine grains have “locked into” a lattice preferred orientation parallel to plate spreading, and the convecting mantle, where they have not been locked in. This nuanced interpretation makes good mechanical sense. We have now added a comment about the extent to which we can compare seismically and geochemically estimated lithospheric thicknesses.

(5) “Line 113 - It is a key point for the whole analysis that intraplate volcanism appears to mainly occur in lithosphere with thermal thickness, as inferred from DVs, of < 100 km. It would help the rest of the paper to include a (supplementary) figure that shows the distribution of seismically inferred lithospheric thicknesses in the locations of intraplate volcanism. The map shown in Fig. 1 only shows that lithosphere in these locations is thin, but not how thin. Such a figure would help motivate the range of lithospheric thicknesses used in the later REE modelling.”

We only partly agree since we believe this information is already adequately displayed in Figure 2b of the main text. In Figure 2b, the red line describes the cumulative distribution of lithospheric thickness beneath intraplate volcanic provinces. We describe this distribution in Lines 181–182 of the main text. Note that shear wave velocity models can be prone to “crustal bleeding” where inaccuracies in the seismic velocity structure of the crust can produce anomalously slow seismic velocities at shallow mantle depths. Therefore, many studies which generate lithospheric thickness maps from seismic tomographic models advise caution when interpreting results for regions with lithospheric thicknesses less than ~ 75 km (Priestley and McKenzie, 2013; Hoggard et al., 2020). We have now made this important point clear in the revised discussion.

(6) “Some of the thicknesses tested are close to the base of the Moho, meaning that you infer there is essentially no mantle lithosphere below parts of the continent. This would warrant at least a comment.”

We fully agree. Interestingly, beneath both Anatolia and western North America there is practically no lithospheric mantle. Within our geochemical methodology, we do vary lithospheric thickness between 30–80 km. Therefore, the reviewer is completely correct that in many continental locations we test lithospheric thicknesses that require there to be no mantle lithosphere beneath parts of the continent. A lower bound of 30 km was chosen because a number of volcanic provinces included within our analysis occur close to, or above, areas of active rifting (e.g., Iceland). The upper bound of 80 km was set because at depths where melting occurs purely within the garnet stability field (i.e. > 73 km), we no longer have any sensitivity to depth within our model. For consistency, we test the same range of lithospheric thicknesses for each volcanic province. We think that our strategy is a reasonable one that has the advantage of objectivity and neutrality.

In many locations of intraplate volcanism, continental crust is 20–40 km thick and so our results predict a lithospheric mantle of ~ 10 –50 km beneath these regions (Bassin et al., 2000). These thicknesses are consistent with the thicknesses predicted from tomographic models. Moreover, it can be argued that this result is unsurprising when you consider the depth range of the peridotite solidus and average melt fractions commonly predicted at intraplate volcanic provinces (i.e. ~ 1 –10%). Thus we are completely at one with the reviewer that this result is worthy of comment and we have now added a sentence highlighting this conclusion in the revised discussion.

(7) “Line 114 and rest of this paragraph - It is a jump for the less expert reader why you suddenly start talking about mineral loss or accumulation and why you are concerned about MgO. I think you need to first tell the reader that you are filtering the

data base to ensure you analyse only those samples that are most likely to represent primary mantle melts rather than those that are affected by significant mineral loss or accumulation during their rise through the lithosphere.”

We agree that this section is a bit confusing and we have now added two qualifiers to this section to help the less expert reader.

(8) “Line 136 and following. Another important point for the analysis is the assumption that ΔV_s at a depth of 150 ± 25 km represents asthenosphere below the regions of intraplate volcanism. Note that a 25–50 km vertical resolution is a rather optimistic estimate. And also, as mentioned in the text, the thermal boundary layer extends deeper than the chosen definition of thermal thickness, and may well reach depths that are 50 km or more below the lithospheric thickness mapped. Hence it would be useful to further convince the reader that the seismic structure at 150 km depth does indeed not reflect a signature of lithospheric thickness variations. This could be done for example by including a (supplementary) figure that shows that the correlation of velocity structure above 100 km depth with velocity structure below this depth is relatively low in the areas of intraplate volcanism. Or it could be done by showing that the depth of a higher temperature isotherm (e.g. 1250 or 1300 °C, closer to but still below reference mantle potential temperature) does not extend to 150 ± 25 km below the regions of intraplate volcanism.”

We understand the point. Reviewer 3 has requested a figure showing that the velocity structure above 100 km depth does not correlate with the velocity structure below this depth in areas of intraplate volcanism. While we believe these points by Reviewer 3 are important, they are actually addressed in the subsequent paragraph and, more significantly, by Figure 2e in the main text. Figure 2e shows the correlation between lithospheric thickness and ΔV_s as a function of depth. As expected, the correlation between lithospheric thickness and ΔV_s is strong at depths ≤ 100 km ($R = 0.8$). This correlation rapidly decreases and becomes indistinguishable from zero at a depth of 150 km. Therefore, we are confident that ΔV_s observed beneath intraplate volcanic provinces at a depth of 150 ± 25 km represents temperature variations in the asthenosphere rather than changes in thermal boundary layer thickness.

(9) “Line 153 and following. This paragraph is quite confusing to read, because the principal component analysis and subsequent correlation with La/Sm, ΔV_s and ϵNd are all introduced together. It would help the reader to first explain the PCA, and what this yields, before introducing the correlation analysis.”

We understand the viewpoint of Reviewer 3. However, we have decided to retain our current structure since we prefer to fully introduce the methodology (i.e. to carry out principal component analysis and then compare these components to observations) prior to describing the significance of the results. To help the reader, we have now moved a proportion of the explanation of the correlation analysis to a location that follows the description of P_1 in order to improve clarity.

(10) “You do a PCA for your global data base of volcanic rock compositions using a subset of the concentrations of eight incompatible elements. You need to tell the reader what the subset is taken from not just say you do a PCA on a subset of eight incompatible elements. It may help to give your data base a name, so that you can reference it more easily throughout the text.”

We fully agree. We do not wish to confuse the reader and we have now named our database “Database 1” throughout the manuscript. By subset, we meant a subset of elements rather than

a subset of geochemical data. We acknowledge that the original wording was confusing and so we have removed the word subset and improved the clarity of this sentence.

(11) “Line 165 and following. You need to explain why if P1 were controlled by mantle depletion, P1 would negatively correlate with ΔV_s . It is not obvious to the reader what positive and negative P1 means. Source depletion would lead to relatively high V_s , relatively high ϵNd and low degrees of melting i.e., high La/Sm. As I understand it, low degrees of melting would also mean high P1 (high concentrations of most incompatible elements). This would mean a positive correlation between P1, La/Sm and ΔV_s in the case of depletion. High temperature would similarly lead to a positive correlation of these three parameters. So to my understanding it is solely the negative correlation between P1 and ϵNd that indicates that source depletion is not the main influence on P1.”

We only somewhat agree since the Reviewer’s interpretation of what we have written in this paragraph is partially correct. We believe that the key observation is ϵNd and ΔV_s do not positively correlate rather than that there is negative correlation between P_1 and ϵNd . We have now added two qualifiers so that readers will know how positive and negative P_1 relate to element composition, melt fraction and depletion.

(12) “Line 209 and following. This analysis really strengthens the interpretation of the principal components. It would help if it were discussed before the interpretation of P1, P2 and P3 is proposed.”

We agree with the first point. We are glad that the Reviewer believes that the synthetic analyses which we added in response to the first round of reviews really strengthens our interpretation of the data-generated principal components. However, we have decided that we prefer to lead with a fully data-driven description and then follow with other tests of our interpretation using synthetic data. We believe it would potentially be more confusing to lead with the synthetic tests or to include them as an integral part of the analysis.

(13) “Line 243-344 - a hydrous melting model is used. Please mention in the main text where water content is taken from, as this is a key parameter, just like you mention that source depletion is estimated from ϵNd .”

We agree and we now state that both mantle depletion and water content are estimated from the value of ϵNd .

(14) “What motivates accepting only models where misfits are < 1.5 minimum misfit? In principle, all models that fit within the uncertainties of the data are acceptable. What is the justification for excluding models that fit within the data uncertainties but have misfits > 1.5 times the minimum misfit?”

We partly agree but we point out that the key feature of our approach is a joint analysis of disparate datasets which narrows the options. The metric of data uncertainty and the point at which any fit becomes unacceptable are necessarily a subjective choice on the part of the observer/modeler. Generally, there are three commonly used approaches as described below. Of these choices, Approaches 2 and 3 exploit a calculation of average misfit scaled to the standard deviations of the datapoints as a metric from quality of fit (RMS misfit). It should be noted that $RMS = 1$ does not mean the misfit is within 1 standard deviation for all data points. Rather, it means that the misfit is on average within one standard deviation so there could be a 50–50 split between data that are well or badly fitted. For example, by referring to Figure 4 in the main text, the reviewer will see

that the dashed red lines which define the accepted models with the largest rms misfit only just lie within the range for all elements despite having RMS values < 1 .

- Approach 1 is the simplest method whereby all models are accepted that fit every data point within error. We have not chosen this option because it does not yield a ‘best-fit’ model, and it means that the spread of the results will be dominated by the most anomalous data point.
- In Approach 2, all models are accepted that fall beneath a misfit threshold, for example RMS < 1 , whose minimum and maximum values of T_p and a define uncertainties. However, by imposing a constant threshold, a poor best-fit model (e.g., RMS = 0.9) may end up with much smaller error bars than a well fit model (e.g. RMS = 0.4).
- In Approach 3, we choose to accept all models within a percentage of the best-fit RMS misfit. Here, poorly fitted models with broad misfit wells are penalised with larger error bars than well-fitting models. We choose $1.5\times$ as our percentage of minimum misfit as it provides a broad range of errors but rarely accepts models which poorly fit the data. Additionally, we exclude all intraplate volcanic bins where this geochemical modeling approach yields a global minimum RMS misfit > 1 in order to avoid including results which cannot be fitted by the inversion algorithm.

Our preferred approach is used extensively throughout the Earth Sciences, Engineering and Material Sciences. As such, we do not believe it needs to be discussed extensively within the main text. In this study, we have been as transparent as possible about our modeling approach. For readers who prefer Approach 1, we provide a visual fit between the data and the model in the left-hand panels of Figure 4. For readers who prefer Approaches 2 or 3, we also provide a visual interpretation of the misfit well in the right-hand panel of Figure 4 and in the Supplementary Material. It is important to emphasize that usage of these different approach does not materially affect the principal conclusions that we have reached.

(15) “Line 283 and following. One might expect that damping applied in tomographic models would reduce the range of inferred T_p , yet the seismic range of T_p is larger than the geochemical range of T_p .”

We agree. Here, the Reviewer raises a significant and interesting point. We concur with Reviewer 3 that damping will remove the extreme values of V_s from the tomographic model. If there is an isolated short wavelength region of extremely low velocity the the damping applied to the tomographic model will act to greatly reduce the amplitude of this anomaly. We believe that damping is an important aspect of tomographic modeling which is often ignored or side-stepped when V_s -to- T calibrations are considered.

There are a number of different V_s -to- T calibrations that are generated by directly fitting mineral physical constraints (e.g., Stixrude and Lithgow-Bertelloni, 2011). The advantage of these mineral physics calibrations is that they can be easily applied to any given tomographic model. However, as highlighted by the reviewer, damping acts to reduce variations in V_s . As a result, V_s -to- T calibrations generated directly from mineral physics data will tend to under-predict mantle temperatures when applied to a damped tomographic model. We believe the affect of damping on these calibrations is a severe limitation of this approach.

Instead of directly fitting experimental constraints, we have exploited a model which follows the approach pioneered by Priestley and McKenzie (2013). Here, parameters in the equations that define the relationship between V_s and T are calibrated to match an assumed mantle temperature

structure. As a result, this calibration must be performed for each individual tomographic model. Significantly, the advantage of this approach is that the range of observed T_p values should match that which is expected for the upper mantle. In this case, the effect of damping is greatly reduced and it only becomes an issue for the largest amplitude anomalies at the shortest wavelengths.

We believe the fact that the range of seismic T_p values is greater than the range of geochemical T_p values is, instead, the result of limitations in the geochemical and tomographic models. At the highest melt fractions, all incompatible elements, such as the REEs, will have entered the melt phase. Therefore, these models have minimal sensitivity beyond $T_p \sim 1450$ °C and so we imposed an upper limit of $T_p = 1500$ °C on our geochemical models, which is lower than the highest T_p estimated by the tomographic model. However, we concede that the range of T_p values investigated by the geochemical model should equal that predicted by models generated by converting V_s into T_p . Therefore, we have expanded this range to 1250–1550 °C. At the lower end, in some cases the tomographic calibration predicts potential temperatures and lithospheric thicknesses at which no melting should occur. In these cases, smearing of nearby thick cold mantle probably affects the T_p and lithospheric thickness values predicted by the tomographic model. Furthermore, a greater compositional or mineralogical range between primitive and depleted mantle end-members exploited during our geochemical modeling would increase the range of T_p estimated. In general, it is difficult to identify and quantify a single uncertainty, such as the effect of smoothing, as the reason for the difference in seismological and geochemical T_p . In the revised manuscript, we have drawn the attention of the reader to this observation.

(16) “This paragraph mentions many of the uncertainties in interpreting tomographically imaged seismic velocities in terms of temperature, except the possible influence of the presence of melt. If melt contributes, and it would contribute more to the lowest seismic velocities, this might explain the larger range of seismic T_p than geochemical T_p .”

We agree and the reviewer is correct to state that some reference to the possible presence of melt is required within the discussion. We have added several sentences which discuss the implications of incipient melting at a depth of 150 ± 25 km on measurements of ΔV_s .

(17) “Note that the reasonable correlation between T_p from geochemical and seismic estimates in Fig. 5 is quite strongly influenced by the highest T_p points. It would be good to see what the correlations look like when done with the T_p from the other seismic tomography models shown in supplementary Fig. 2, i.e. add another supplementary figure with these correlations.”

We appreciate the Reviewer’s desire to see the T_p distribution constructed from other seismic tomographic models. However, each tomographic model must be individually calibrated when using the V_s -to- T method of Hoggard et al. (2020). As an authorship, we were not involved in this previous study and we simply exploit the temperature grids provided within Hoggard et al. (2020). We do not have access to the codes used to generate these V_s -to- T models and so we cannot calibrate them for additional tomographic models. Furthermore, we believe it is a lengthy process to perform these calibrations and therefore it is beyond the scope of this particular manuscript. Furthermore, we strongly believe that the SL2013sv tomographic model is superior, at least for our purposes, to other published tomographic models. This assertion is based upon the high quality of correlation between the SL2013sv model and a variety of other geologic and geophysical observations.

We agree that the visual correlation between T_p from geochemical and seismic estimates in Fig. 5 is quite strongly influenced by the highest T_p points. However, it should be noted that these points do not strongly influence the calculated correlation coefficient, R . These high- T_p points are

located within Iceland and so the weighting of these points is low since bin area reduces towards the poles. This down-weighted importance can be demonstrated since R has not changed even though the highest T_p points have increased from 1500 to 1550 °C as a result of the changes we have implemented in response to Comment 15.

(18) “Line 350 and onwards. The term dynamic topography is used here to refer to changes in isostatic topography due to changes in local mantle thermal structure and lithospheric thickness rather than in the sensu-stricto meaning of the term dynamic topography as topography in response to the upward or downward pressures from mantle flow. It would be good to clarify the use of this terminology at the start, or preferably use a different terminology, e.g. adjustment of isostatic topography.”

We partly agree, primarily because the *sensu stricto* definition of dynamic topography as being caused exclusively by the pressure field generated by mantle flow is neither complete nor useful. As was noted in the 1980s, it is in a practical sense almost impossible to disentangle mantle flow field from the closely associated effects of thermal isostasy immediately beneath the plate or from modest plate thickness variations that are caused by mantle convection since the plate is, from a fluid dynamical perspective, a thermal boundary layer. Dynamic topography *sensu lato* includes all three effects which are extremely difficult to parse from an observational point of view. Here, we have chosen to define dynamic topography in the same way as a general review of the subject, which is *in press* in an forthcoming contribution to the *AGU Geophysical Monograph* series, has done. To aid clarity, we have added the following statement— “ We define dynamic topography to embrace long-wavelength topography generated mantle flow, isostatic responses to thermochemical processes within the convecting mantle, as well as regional changes in thickness of the lithospheric mantle (Moucha and Forte, 2011; Hoggard et al., in press).” It is notable that there is little agreement in the literature about the definition of the term but we emphasize that a narrow definition of dynamic topography that only refers to the flow field is both restrictive and impracticable.

(19) “Line 31 and 37 - upper mantle should be uppermost mantle, as the upper mantle extends down to 700 km depth, i.e. much deeper than lithosphere + 100–200 km thick asthenosphere”

We fully agree and we have made these changes in the interests of greater clarity.

(20) “Line 42 - “calibrated seismic tomographic models”. In what way are these calibrated? The conversion to temperature is calibrated, but the tomographic models themselves are not or?”

We have replaced “calibrated seismic tomographic models” with “mantle seismic velocities”.

(21) “Line 55 - “seismic tomography” should be “seismic velocities”.”

We fully agree that this sentence is sloppy. We have changed “seismic tomography” to “seismic velocities”.

(22) “Line 70 - “sub-surface observations” should be “sub-surface structure”. The observations are taken at the surface”

We agree but we feel it is important to say we are observing the sub-surface. Therefore we have changed the phrasing to “observations of sub-surface structure”.

(23) “Line 121 - the reader could use some additional help with understanding that “positive correlation” means low degree of melting correlates with high seismic velocities”

We agree and we have added the following sentence: “Therefore, igneous rocks with lower La/Sm ratios, which are indicative of higher melt fractions, coincide with lower ΔV_s at 150 ± 25 km depths, which is indicative of higher temperatures at these depths.”

(24) “Line 151 - “melting occurs to a lesser degree within a depleted source region since depleted mantle has lower initial La/Sm”. This sentence does not make sense.”

We agree and we have added a missing comma.

(25) “Line 187 - “elevated asthenospheric temperature acts to depress the solidus” This is not true. The solidus does not change as a function of asthenospheric temperature. The depth where melting starts changes if asthenospheric temperature changes.”

We agree and we have changed “to depress the solidus” to “to deepen the onset of melting” which makes more physical sense.

(26) “Line 254-256 - last sentence of this paragraph “Volcanic province.. combinations” just seems to make the point that if the spread of data is larger more models can fit. This seems trivial.”

We mildly disagree in that we added this sentence in response to a particular Reviewer at an earlier stage in this editorial process. We completely agree that it may be easy to infer this relationship between data range and misfit but we feel it is worth making this point explicit, as we were asked to do.

(27) “Line 285 - Why the emphasis on “great-circle ray path coverage”? Why not just “ray path coverage”.”

We used this term because the SL2013sv tomographic model is predominantly constructed from surface wave observations which means that resolution is limited by great circle path coverage. Nonetheless, we have now made the change that the reviewer requests to aid the general reader.

(28) “Line 339 - “This inference is manifest by a range of geochemical and geophysical observations” This sentence does not add any information. It could be cut.”

We disagree. We agree that this sentence is not strictly necessary, but we feel it leads nicely into the list of observations we give in the subsequent sentences. Therefore, we have decided to keep this sentence.

(29) “Line 347 - “major element compositions” - do you mean “major element compositions of mantle melts”? Or are you referring to composition of the source (as discussed in the following sentence)?”

We have added a qualifier which states that these major element compositions specifically refer to sea-floor basalts.

(30) “Line 348 - “these correlations could also be partially modified by”. It is unclear what “also” refers to. You need to say that the correlation was modelled with variations in mantle temperature. “partially modified by” - do you mean “partially explained by”? ”

We agree. We have now restructured this sentence so that we refer explicitly to the role of temperature. We have retained the phrase “partially modified by” instead of “partially explained by” since in our view these processes can act to both enhance and destroy this correlation.

(31) “Line 352 - “rapid regional uplift” The rapidity of uplift would depend on the speed with which lithospheric thinning or change in asthenospheric temperature occurs?”

We agree and we have therefore removed the word “rapid” which is unnecessary.

(32) “Line 379-380 - “vertical motions superimposed on motions generated by mantle convection”. It is the latter motions (by mantle convection) that are usually referred to as “dynamic topography” whereas lithospheric thinning leads to isostatic adjustments of topography.”

We disagree for a number of reasons. First, mantle convection will directly affects the thickness and thermal structure of the lithospheric plate on a regional basis (i.e. other than plate cooling in the oceanic realm). Secondly, it is generally impracticable to separate out mantle flow, thermal anomalies within the uppermost mantle, and small changes in the boundary layer thickness. We have now more carefully defined our use of the term “Dynamic Topography” to also include long-wavelength variations of topography which are not generated through tectonic processes. Under this definition, our sentence here is valid.

(33) “Fig. 4. The caption of this figure needs to refer to the equation used to calculate RMS (in the methods). Different definitions of RMS are possible. It would also be good if panel c and d could outline the range of accepted solutions to show how this range may differ from the range of models that fall within 1 standard deviation (RMS=1).”

We agree and we now refer to the relevant equation within the caption. As we explain in Comment 14, we do not believe that $RMS = 1$ is a better or worse metric of misfit than the one we have chosen. However, we have added appropriate error bars to Figure 4.

(34) “Line 413 - What has been recomputed? Has a new 1-D reference model been derived? What crustal model is used as reference?”

We have elaborated on the crustal model that is used and how it is updated during optimization.

(35) “Line 418-429 - First paragraph Section 8.2. The description of the conversion method starts with secondary details that the method is “pioneering” and based on data from borneol experiments. None of this gives the reader any preview of what the method actually involves. It would be much more useful if you could give a brief summary of the method before explaining the details, i.e. the conversion method is based on a calibration of parameters to maximise the agreement between seismic velocities imaged below the oceans and a plate cooling model overlying an adiabatic mantle of constant potential temperature. The calibration is used to set 6 material parameters and the adiabatic temperature-depth gradient. The anharmonic shear modulus is assumed to vary linearly with P and T and attenuation is adjusted by varying the reference steady-state viscosity and an activation energy and volume. It is a decision rather than a necessity to adjust these 7 parameters in the fitting.”

We agree and we have both rearranged and rewritten this opening paragraph in accordance with these suggestions. The revised version is now more succinct and focusses on the key details.

(36) “Second paragraph Section 8.2. It would help to introduce H_i at the end of the first line “compared to four sets of observational constraints H_i .” The next line

would be clearer if it read **The total misfit H, between calculated and observed Vs, is quantified**”

We agree and we have made both these suggested changes and several changes of our own to make this paragraph clearer and more logical.

(37) “Line 440: Why is the isentrope for pyrope used? Why not for a peridotite?”

We have checked the original publications and we apologize for this silly mistake. We have now corrected pyrope to peridotite.

(38) “Line 440: What data base was used with the PerPle-X software package?”

We have revised and updated this section to state that the constants from Shorttle et al. (2014) were used to calculate a peridotite isentrope for a potential temperature of 1331 °C.

(39) “Line 447: What bulk viscosity was assumed? And why was bulk viscosity used rather than shear viscosity?”

We provided the value of “bulk viscosity” in Table 4 of the Supplementary Materials alongside the other 6 independently constrained parameters. However, we acknowledge that “bulk viscosity” is imprecise and vague. What is referred to as “bulk viscosity”, both in our methods and in previous publications that exploit this method (e.g., Hoggard et al., 2020; Richards et al., 2020), is the average viscosity of the upper mantle. This value therefore represents an average, or “bulk”, shear viscosity. We have now changed this term in the Methods and Supplementary Materials to be “shear viscosity”. We have also added the reference value of shear viscosity to the end of the appropriate sentence.

(40) “Line 481: “the global data base”, please be specific about which global data base. I assume this is your own new data base. You may want to give it a name so that you can refer to it here.”

We fully agree and we now refer to our database as Database 1 throughout the Main Text, Methods and Supplementary Materials.

(41) “Line 523: Please give references for the PM and DMM models used, either here or in the supplement.”

We agree and we have added appropriate references at the end of this sentence and to the caption of Table 5.

(42) “Line 525: What $X_{\text{H}_2\text{O}}^{\text{bulk}}$ was used?”

On lines 518–519, we state that “ $X_{\text{H}_2\text{O}}^{\text{bulk}}$ is approximated from the concentration of Ce within the source region and $X_{\text{Ce}}^{\text{bulk}}$ is constrained by assuming that $X_{\text{H}_2\text{O}}^{\text{bulk}}/X_{\text{Ce}}^{\text{bulk}} = 200$ (Michael, 1995)”. On line 525, we highlight that we assume the $\varepsilon\text{Nd} = 10$ at mid-oceanic ridges. Therefore, $X_{\text{H}_2\text{O}}^{\text{bulk}}$ is 200 times the concentration of Ce in the depleted mantle source. The concentration of Ce in depleted mantle can be found in Table 5 of the Supplementary Materials. Notwithstanding these sentences, we understand why the Reviewer found this section difficult to follow. Therefore, we have added a sentence to the Methods to clarify the relationship between εNd and water content. We have also added the H₂O content of primitive and depleted mantle to Table 5 of the Supplementary Materials.

(43) “Line 580: Please give a reference for the choice of Fe³⁺/total Fe.”

We have added an appropriate reference— (Ball et al., 2019).

(44) “Supplementary Fig. 5 through 8 - please make clear in the captions which of these are PCA of the data base and which from the synthetic models.”

We have now revised these captions in accordance with this suggestion.

(45) “Supplementary Fig. 8 labels (a), (b) etc missing on the figure panels but referred to in the caption.”

We have now added these labels.

References

- P. W. Ball, N. J. White, A. Masoud, S. Nixon, M. Hoggard, J. Maclellan, F. Stuart, C. Oppenheimer, and S. Kröpelin. Quantifying asthenospheric and lithospheric controls on mafic magmatism across North Africa. *Geochemistry, Geophysics, Geosystems*, 20:3520–3555, 2019.
- C. Bassin, G. Laske, and G. Masters. The current limits of resolution for surface wave tomography in North America. *EOS Trans AGU*, 81(F897), 2000.
- I. Guest, G. Ito, M. O. Garcia, and E. Hellebrand. Extensive magmatic heating of the lithosphere beneath the hawaiian islands inferred from salt lake crater mantle xenoliths. *Geochemistry, Geophysics, Geosystems*, page e2020GC009359, 2020.
- M. Hoggard, J. Austermann, C. Randel, and S. Stephenson. Observational estimates of dynamic topography through space and time. *AGU Monograph*, in press.
- M. J. Hoggard, K. Czarnota, F. D. Richards, D. L. Huston, A. L. Jaques, and S. Ghelichkhan. Global distribution of sediment-hosted metals controlled by craton edge stability. *Nature Geoscience*, 13(7):504–510, 2020.
- J. Jenkins, J. Maclellan, R. G. Green, S. Cottaar, A. Deuss, and R. S. White. Crustal formation on a spreading ridge above a mantle plume: receiver function imaging of the Icelandic crust. *Journal of Geophysical Research: Solid Earth*, 123(6):5190–5208, 2018.
- P. Michael. Regionally distinctive sources of depleted MORB: Evidence from trace elements and H₂O. *Earth and Planetary Science Letters*, 131(3–4):301–320, 1995. doi: 10.1016/0012-821X(95)00023-6.
- R. Moucha and A. M. Forte. Changes in African topography driven by mantle convection. *Nature Geoscience*, 4(10):707–712, 2011.
- Y. Niu and D. H. Green. The petrological control on the lithosphere-asthenosphere boundary (LAB) beneath ocean basins. *Earth-Science Reviews*, 185:301–307, 2018.
- Y. Niu, M. Wilson, E. R. Humphreys, and M. J. O’Hara. The origin of intra-plate ocean island basalts (OIB): The lid effect and its geodynamic implications. *Journal of Petrology*, 52(7-8): 1443–1468, 2011.
- A. Pleus, G. Ito, P. Wessel, and L. N. Frazer. Rheology and thermal structure of the lithosphere beneath the hawaiian ridge inferred from gravity data and models of plate flexure. *Geophysical Journal International*, 222(1):207–224, 2020.

- K. Priestley and D. McKenzie. The relationship between shear wave velocity, temperature, attenuation and viscosity in the shallow part of the mantle. *Earth and Planetary Science Letters*, 381: 78–91, 2013. doi: 10.1016/j.epsl.2013.08.022.
- F. D. Richards, M. J. Hoggard, L. Cowton, and N. J. White. Reassessing the Thermal Structure of Oceanic Lithosphere With Revised Global Inventories of Basement Depths and Heat Flow Measurements. *Journal of Geophysical Research: Solid Earth*, 123:9136–9161, 2018.
- F. D. Richards, M. J. Hoggard, N. White, and S. Ghelichkhan. Quantifying the relationship between short-wavelength dynamic topography and thermomechanical structure of the upper mantle using calibrated parameterization of anelasticity. *Journal of Geophysical Research: Solid Earth*, page e2019JB019062, 2020.
- O. Shorttle, J. Maclennan, and S. Lambart. Quantifying lithological variability in the mantle. *Earth and Planetary Science Letters*, 395:24–40, 2014.
- L. Stixrude and C. Lithgow-Bertelloni. Thermodynamics of mantle minerals-II. Phase equilibria. *Geophysical Journal International*, 184(3):1180–1213, 2011.

REVIEWERS' COMMENTS

Reviewer #3 (Remarks to the Author):

Generally the authors have carefully responded to my comments.

I still do not fully agree on all counts with the authors. However, I think the manuscript is sufficiently clarified for publication.

For future work, the authors may want to consider the following points though.

(4) The answer of the authors does not actually address my question for how the mapped lithospheric thickness is related to the top of the melting column in the REE concentration modelling

(8) The correlation between V_s and lithospheric thickness presented in Fig. 2e does not directly answer the question I posed. It makes sense that lithospheric thickness correlates with V_s at the depth of this thickness. But the relationship between V_s anomalies and lithospheric thickness is not a simple one and hence it is not necessarily justified to use lithospheric thickness as a proxy for velocity at the depth around the base of the lithosphere. Hence a correlation between thickness and velocity does not necessarily reflect the correlation between velocity anomalies at different depths.

(17) Calling a tomographic model "superior" because it correlates best with your data is not a strong argument. Such an evaluation should be based on the data and methods used to derive the model.

(18) I disagree that it is not useful and impractical to distinguish between the upward or downward pressures of mantle flow and isostatic adjustments to variations in mantle temperature and lithospheric thickness. While in nature, variations in mantle temperature and flow often occur together, from a forward modelling point of view the two effects are easily distinguished and estimating the effect of flow requires much more involved models than the estimating the isostatic part of the response. Variations in lithospheric thickness could well be inherited from earlier stages in mantle dynamics and hence there is no reason to assume that these have to correlate with current regions of upward/downward flow. ("The 1980s" is hardly an authoritative reference for the view presented by the authors).

Global Influence of Mantle Temperature and Plate Thickness on Intraplate Volcanism

P. W. Ball, N. J. White, J. MacLennan and S. N. Stephenson

Response to Reviewers' Comments

Reviewer 3

Reviewer 3 states that since we “carefully responded” to their comments, they now believe that this “manuscript is sufficiently clarified for publication”. We are grateful to this reviewer for their positive endorsement and for the insightful comments that they provided in previous reviews which enabled us to improve this manuscript and to enhance its potential impact. Reviewer 3 includes with four minor follow-up remarks/comments, which we have carefully addressed below. Since these remarks pertain to this reviewer’s previous review, we have in each case included both the original comment and response for reference and for clarity. The numbering of these comments matches that of the reviewer. In our view, none of the four minor issues that have been raised materially affect the principal conclusions reached by us.

(Original Comment 4) “It also needs to be motivated somewhere in the manuscript how the thus defined thermal base of the lithosphere would relate to the top of the melting column which is used in the modelling of REE concentrations. Would this be the same isotherm and why?”

(Original Response) We mostly agree. Although the base of the lithosphere presented by global thickness maps employed throughout this study is thermally defined, it is constrained by mechanical observations. 1175 °C was chosen as the base of the lithosphere by Richards et al. (2018) in their revised oceanic plate cooling model. This particular isothermal surface was chosen since it coincides with the peak change in orientation of azimuthal anisotropy within the Pacific Ocean. The change in orientation is thought to demarcate the transition between rigid lithosphere, where the olivine grains have “locked into” a lattice preferred orientation parallel to plate spreading, and the convecting mantle, where they have not been locked in. This nuanced interpretation makes good mechanical sense. We have now added a comment about the extent to which we can compare seismically and geochemically estimated lithospheric thicknesses.

(Follow-Up of Comment 4) “The answer of the authors does not actually address my question for how the mapped lithospheric thickness is related to the top of the melting column in the REE concentration modelling.”

The simplest interpretation is that the base of the mapped lithosphere effectively represents the top of the melting column subject, of course, to the independent resolution constraints provided by both the calibrated tomographic model(s) and the results of geochemical inverse modeling. Although the implication of the reviewer’s comment is that we have neglected to provide a specific answer to a question originally posed by Reviewer 3 in their previous review, we have, in fact, already revised the manuscript in light of this (previous) comment. In particular, we included the statement “although it is not guaranteed that a seismically defined lithosphere-asthenosphere boundary will

coincide with the depth at which melting ceases, it is likely that the difference between these depths is minimal compared to the uncertainties inherent in both techniques”. This careful statement remains valid for two reasons. First, since, as we state in Original Response 1, we define the base of the lithosphere as the depth at which vigorous convection ceases, it represents a significant barrier to adiabatic decompression melting. Thus, the base of the lithosphere does indeed represent the top of the melt column. Secondly, melting may actually cease at depths that are greater (in resolvable terms) than our thermally defined lithosphere, provided that temperatures are low enough to inhibit melting. Given that temperature increases rapidly as a function of depth close to, and within, the lithosphere, the difference between the base of the lithosphere and the top of the melt column is likely to be negligibly small compared to the uncertainties associated with either the geochemical and tomographic lithospheric thickness estimates. In summary, although Reviewer 3’s further (and original) comment is a reasonable one, we had in fact already addressed it within the previously revised manuscript and, importantly, any residual uncertainty in this comparison of depths does not materially affect our conclusions.

(Original Comment 8) “Line 136 and following. Another important point for the analysis is the assumption that ΔV_s at a depth of 150 ± 25 km represents asthenosphere below the regions of intraplate volcanism. Note that a 25–50 km vertical resolution is a rather optimistic estimate. And also, as mentioned in the text, the thermal boundary layer extends deeper than the chosen definition of thermal thickness, and may well reach depths that are 50 km or more below the lithospheric thickness mapped. Hence it would be useful to further convince the reader that the seismic structure at 150 km depth does indeed not reflect a signature of lithospheric thickness variations. This could be done for example by including a (supplementary) figure that shows that the correlation of velocity structure above 100 km depth with velocity structure below this depth is relatively low in the areas of intraplate volcanism. Or it could be done by showing that the depth of a higher temperature isotherm (e.g. 1250 or 1300 °C, closer to but still below reference mantle potential temperature) does not extend to 150 ± 25 km below the regions of intraplate volcanism.”

(Original Response) We understand the point. Reviewer 3 has requested a figure showing that the velocity structure above 100 km depth does not correlate with the velocity structure below this depth in areas of intraplate volcanism. While we believe these points by Reviewer 3 are important, they are actually addressed in the subsequent paragraph and, more significantly, by Figure 2e in the main text. Figure 2e shows the correlation between lithospheric thickness and ΔV_s as a function of depth. As expected, the correlation between lithospheric thickness and ΔV_s is strong at depths ≤ 100 km ($R = 0.8$). This correlation rapidly decreases and becomes indistinguishable from zero at a depth of 150 km. Therefore, we are confident that ΔV_s observed beneath intraplate volcanic provinces at a depth of 150 ± 25 km represents temperature variations in the asthenosphere rather than changes in thermal boundary layer thickness.

(Follow-Up of Comment 8) “The correlation between V_s and lithospheric thickness presented in Fig. 2e does not directly answer the question I posed. It makes sense that lithospheric thickness correlates with V_s at the depth of this thickness. But the relationship between V_s anomalies and lithospheric thickness is not a simple one and hence it is not necessarily justified to use lithospheric thickness as a proxy for velocity at the depth around the base of the lithosphere. Hence a correlation between thickness and velocity does not necessarily reflect the correlation between velocity anomalies at different depths.”

The reviewer is correct that the relationship between V_s and lithospheric thickness is not a sim-

ple one. Therefore lithospheric thickness cannot generally be used as a proxy for ΔV_s at shallow depths. However, the lithospheric thickness model that we exploit is calculated from the SL2013sv tomographic model. Therefore, in this specific case, lithospheric thickness is indeed closely associated with ΔV_s at lithospheric depths. Furthermore, if ΔV_s at a depth of 75 km strongly correlates with lithospheric thickness ($R \sim 0.8$), and if ΔV_s at a depth of 150 km does not correlate with lithospheric thickness ($R < 0.3$), it is safe to assume that ΔV_s values recorded at 75 and 150 km depth do not correlate with each other. In summary, we stand by the importance of revised Figure 2e in this regard and the conclusions that can be safely drawn from it.

(Original Comment 17) “Note that the reasonable correlation between T_p from geochemical and seismic estimates in Fig. 5 is quite strongly influenced by the highest T_p points. It would be good to see what the correlations look like when done with the T_p from the other seismic tomography models shown in supplementary Fig. 2, i.e. add another supplementary figure with these correlations.”

(Original Response) We appreciate the Reviewer’s desire to see the T_p distribution constructed from other seismic tomographic models. However, each tomographic model must be individually calibrated when using the V_s -to- T method of Hoggard et al. (2020). As an authorship, we were not involved in this previous study and we simply exploit the temperature grids provided within Hoggard et al. (2020). We do not have access to the codes used to generate these V_s -to- T models and so we cannot calibrate them for additional tomographic models. Furthermore, we believe it is a lengthy process to perform these calibrations and therefore it is beyond the scope of this particular manuscript. Furthermore, we strongly believe that the SL2013sv tomographic model is superior, at least for our purposes, to other published tomographic models. This assertion is based upon the high quality of correlation between the SL2013sv model and a variety of other geologic and geophysical observations.

We agree that the visual correlation between T_p from geochemical and seismic estimates in Fig. 5 is quite strongly influenced by the highest T_p points. However, it should be noted that these points do not strongly influence the calculated correlation coefficient, R . These high- T_p points are located within Iceland and so the weighting of these points is low since bin area reduces towards the poles. This down-weighted importance can be demonstrated since R has not changed even though the highest T_p points have increased from 1500 to 1550 °C as a result of the changes we have implemented in response to Comment 15.

(Follow-Up of Comment 17) “Calling a tomographic model “superior” because it correlates best with your data is not a strong argument. Such an evaluation should be based on the data and methods used to derive the model.”

We disagree with the implication that we have been insufficiently rigorous, or overly qualitative, in our approach to selection and use of tomographic models. Crucially, we chose to exploit the SL2013sv tomographic model primarily because of the quality of its correlation with a variety of geologic and geophysical observations that are independent to the results of this study (e.g., Klöcking et al., 2018; Ball et al., 2019; Richards et al., 2020). We strongly disagree with the implication by Reviewer 3 that we simply chose this model because it provides the best results for our purposes. In fact, of the four tomographic models that we analyzed, it provided the third strongest correlation between ΔV_s and La/Sm at 150 ± 25 km depth (Main Text Figure 2d, Supplementary Figure 2). For clarity and transparency, we do indeed present the results for four tomographic models until we reach the “Calculating Asthenospheric Temperatures” section. Here, as we clearly and previously explained, we have focussed upon the SL2013sv tomographic model since it is the only one that has been converted into temperature using the scheme of Hoggard et al. (2020). In summary, the issue that the reviewer raises is rather a minor one since our general approach and conclusions are indeed consistent for a range of global tomographic models.

(Original Comment 18) “Line 350 and onwards. The term dynamic topography is used here to refer to changes in isostatic topography due to changes in local mantle thermal structure and lithospheric thickness rather than in the sensu-stricto meaning of the term dynamic topography as topography in response to the upward or downward pressures from mantle flow. It would be good to clarify the use of this terminology at the start, or preferably use a different terminology, e.g. adjustment of isostatic topography.”

(Original Response) We partly agree, primarily because the sensu stricto definition of dynamic topography as being caused exclusively by the pressure field generated by mantle flow is neither complete nor useful. As was noted in the 1980s, it is in a practical sense almost impossible to disentangle mantle flow field from the closely associated effects of thermal isostasy immediately beneath the plate or from modest plate thickness variations that are caused by mantle convection since the plate is, from a fluid dynamical perspective, a thermal boundary layer. Dynamic topography sensu lato includes all three effects which are extremely difficult to parse from an observational point of view. Here, we have chosen to define dynamic topography in the same way as a general review of the subject, which is in press in an forthcoming contribution to the AGU Geophysical Monograph series, has done. To aid clarity, we have added the following statement— “ We define dynamic topography to embrace long-wavelength topography generated mantle flow, isostatic responses to thermochemical processes within the convecting mantle, as well as regional changes in thickness of the lithospheric mantle (Moucha and Forte, 2011; Hoggard et al., in press).” It is notable that there is little agreement in the literature about the definition of the term but we emphasize that a narrow definition of dynamic topography that only refers to the flow field is both restrictive and impracticable.

(Follow-Up of Comment 18) “I disagree that it is not useful and impractical to distinguish between the upward or downward pressures of mantle flow and isostatic adjustments to variations in mantle temperature and lithospheric thickness. While in nature, variations in mantle temperature and flow often occur together, from a forward modelling point of view the two effects are easily distinguished and estimating the effect of flow requires much more involved models than the estimating the isostatic part of the response. Variations in lithospheric thickness could well be inherited from earlier stages in mantle dynamics and hence there is no reason to assume that these have to correlate with current regions of upward/downward flow. (“The 1980s” is hardly an authoritative reference for the view presented by the authors).”

We disagree with the further elaborated views raised by this follow-up comment. A fundamental and general problem is that, for any given thermal anomaly, the mantle flow field can conceivably be either vertical (i.e. upward or downward pressure), horizontal (i.e. negligible upward or downward pressure) or, indeed, some combination of both. Pertinent examples are large mantle plumes where localized vertical flow is probably confined to the immediate environs of the conduit and elsewhere horizontal flow is likely to prevail. Where flow is vertical, it is correct to state that the thermal isostatic calculation (carried out using appropriate sensitivity kernels as necessary) yields a lower bound for dynamic topography which, for the purposes of this manuscript, is a perfectly satisfactory outcome that in no way affects our principal conclusions. Where flow is horizontal, the thermal isostatic calculation yields an estimate of dynamic topography. We emphasize that it is important to recall that mantle flow and thermal isostasy are both dynamic responses to density variations within the mantle. These responses can act in the same way (i.e. a low density parcel of mantle can flow upward) and thus generate a net positive isostatic response. However, these responses can also be divergent in terms of direction. We completely agree with Reviewer 3 that modeling of mantle flow is a much more involved process which requires, at the very least, knowledge of, or

a prediction of, the density structure of the mantle. If this structure is known, then the isostatic response generated by this structure is straightforward to calculate. When we study observed dynamic topography we are primarily interested in the topographic response to mantle processes, rather than the mantle processes themselves which are poorly known. The bottom line is that forward modeling of mantle convection cannot easily solve the problem of how the observable Earth actually behaves—hence our focus on observational constraints rather than on numerical models of what might conceivably be happening.

With regard to the point raised concerning the longevity of lithospheric thickness variations, we agree in general terms with Reviewer 3 that present-day lithospheric thickness could have persisted for extended periods of time. If so, there may be no reason to assume that these regions have been uplifted as a result of lithospheric thinning. Nevertheless, it is important to emphasize that we include a significant corollary in this manuscript where we state that in the purely isostatic case “variations in ΔT will change uplift, U , by $\sim 3 \text{ m } ^\circ\text{C}^{-1}$ ”. The whole strength of our analysis is that for melting to initiate, and for intraplate volcanic provinces to form, there must be either an increase in asthenospheric temperature, or a decrease in lithospheric thickness, or both. These processes will generate a positive dynamic response immediately prior to, or coincident with, this volcanic activity through a combination of thermal isostatic uplift and, possibly, mantle flow. Crucially, this corollary directly leads to the fact that we observe emergent Cretaceous-Quaternary marine strata in many active volcanic regions. Notwithstanding the reviewer’s comment, by far the simplest inference is that a combination of hotter asthenosphere and a thinner (i.e. thinned) plate has led to these marine stratigraphic anomalies. In summary, we disagree with both of the points re-raised by this follow-on comment and we believe that we have provided sufficient justification in the existing manuscript to support our views.

References

- P. W. Ball, N. J. White, A. Masoud, S. Nixon, M. Hoggard, J. MacLennan, F. Stuart, C. Openheimer, and S. Kröpelin. Quantifying asthenospheric and lithospheric controls on mafic magmatism across North Africa. *Geochemistry, Geophysics, Geosystems*, 20:3520–3555, 2019.
- M. Hoggard, J. Auestermann, C. Randel, and S. Stephenson. Observational estimates of dynamic topography through space and time. *AGU Monograph*, in press.
- M. J. Hoggard, K. Czarnota, F. D. Richards, D. L. Huston, A. L. Jaques, and S. Ghelichkhan. Global distribution of sediment-hosted metals controlled by craton edge stability. *Nature Geoscience*, 13(7):504–510, 2020.
- M. Klöcking, N. J. White, J. MacLennan, D. McKenzie, and J. G. Fitton. Quantitative Relationships Between Basalt Geochemistry, Shear Wave Velocity, and Asthenospheric Temperature Beneath Western North America. *Geochemistry, Geophysics, Geosystems*, 19(9):3376–3404, 2018. ISSN 15252027. doi: 10.1029/2018GC007559.
- R. Moucha and A. M. Forte. Changes in African topography driven by mantle convection. *Nature Geoscience*, 4(10):707–712, 2011.
- F. D. Richards, M. J. Hoggard, L. Cowton, and N. J. White. Reassessing the Thermal Structure of Oceanic Lithosphere With Revised Global Inventories of Basement Depths and Heat Flow Measurements. *Journal of Geophysical Research: Solid Earth*, 123:9136–9161, 2018.

F. D. Richards, M. J. Hoggard, N. White, and S. Ghelichkhan. Quantifying the relationship between short-wavelength dynamic topography and thermomechanical structure of the upper mantle using calibrated parameterization of anelasticity. *Journal of Geophysical Research: Solid Earth*, page e2019JB019062, 2020.